# Sensitivity of atmospheric rivers to aerosol treatment in regional climate simulations: Insights from the AIRA identification algorithm

Eloisa Raluy-López[1], Juan Pedro Montávez[1], and Pedro Jiménez-Guerrero[1,2]

[1]Physics of the Earth, Regional Campus of International Excellence (CEIR) "Campus Mare Nostrum", University of Murcia, Spain.
[2]Biomedical Research Institute of Murcia (IMIB-Arrixaca), Spain.

**Correspondence:** Juan P. Montávez (montavez@um.es); Pedro Jiménez-Guerrero (pedro.jimenezguerrero@um.es)

**Abstract.** This study analyzed the sensitivity of Atmospheric Rivers (ARs) to aerosol treatment in regional climate simulations. Three experiments covering the Iberian Peninsula for the period 1991 to 2010 were examined, each including prescribed aerosols (BASE), direct and semi-direct aerosol effects (ARI), and direct, semi-direct, and indirect aerosol effects (ARCI). A new regional-scale AR identification algorithm, AIRA, was developed and used to identify around 250 ARs in each experiment.
The results showed that spring and autumn ARs were the most frequent, intense, and long-lasting, and that ARs could explain up to a 30 % of the total accumulated precipitation. The inclusion of aerosols was found to redistribute precipitation, with increases in the areas of AR occurrence. The analysis of common AR events showed that the differences between simulations were minimal in the most intense cases, and a negative correlation was found between mean direction and mean latitude differences. This implies that more zonal ARs in ARI or ARCI with respect to BASE could also be linked to northward deviations. The joint analysis and classification of dust and sea salt aerosol distributions allowed clustering the common events into eight main aerosol configurations in ARI and ARCI. The sensitivity of ARs to different aerosol treatments was observed to be relevant, inducing spatial deviations and IVT magnitude reinforcements/attenuations with respect to the BASE simulation depending on the aerosol configuration. The correct inclusion of aerosol effects is thus important for the simulation of AR behavior at both global and regional scales, which is essential for meteorological predictions and climate change projections.

## 1 Introduction

Atmospheric rivers (ARs) are long and narrow structures with high water vapor concentration that transport up to 90 % of moisture from the tropics to mid-latitudes and the poles (Gimeno et al., 2014; Zhu and Newell, 1998). ARs provide a significant source of water in the form of rain or snow, enabling the regeneration of water resources in areas where they make landfall. However, they are also associated with extreme precipitation events (Trigo et al., 2015; Eiras-Barca et al., 2018). ARs are typically situated in the warm conveyor belts of extratropical cyclones and are associated with strong winds at low levels. At any given time, there are usually 4-5 ARs on a global scale (Zhu and Newell, 1998), since each planetary wave is generally linked to an extratropical cyclone on a synoptic scale (Ralph et al., 2004). The number of ARs in the midlatitudes increases during autumn and winter months, as extratropical cyclones are more frequent during these seasons (Gimeno et al., 2014).

With the advent of weather satellites and atmospheric general circulation models, research on ARs has considerably in-creased. From the beginning, the West Coast of the United States (e.g., Lorente-Plazas et al. (2018), Guan et al. (2012)) and the Pacific Ocean (e.g., Ralph et al. (2004, 2011)) have been the most studied regions. However, a multitude of different studies have been carried out more recently trying to shed light on ARs all around the world. The modification of ARs due to climate change is of great research interest (e.g., Lavers et al. (2013), Ramos et al. (2016b), Payne et al. (2020), Algarra et al. (2020), Gröger et al. (2022), O'Brien et al. (2022), Shields et al. (2023)). Some of these authors suggest that an increased atmospheric moisture due to global warming will lead to an intensification of ARs activity, and to a potential enhancement of the AR-related precipitation. Another topic of great interest is the influence of ARs on the Arctic Sea ice, as they have been related with a slowing of the ice seasonal recovery (e.g., Zhang et al. (2023)). Western Europe has been the focus of several studies in the last decade. These studies have demonstrated a connection between ARs and their Mediterranean variant (Lorente-Plazas et al., 2020) with some of the heaviest rainfall recorded in the Iberian Peninsula (IP) (e.g., Lavers and Villarini (2013, 2015), Trigo et al. (2015), Eiras-Barca et al. (2018)). In addition, up to 90 % of anomalous rainfall in some IP areas coincides with the arrival of ARs. This percentage has a maximum in winter and a minimum in mid-spring (Eiras-Barca et al., 2018). A more recent study has characterized the strength and impacts of ARs on the European west coast by adapting and applying the AR Scale (Ralph et al., 2019) to Europe (Eiras-Barca et al., 2021).

The importance of ARs has given rise to numerous identification algorithms (also known as Atmospheric River Detection Tools, ARDTs) with a wide range of methodologies and conclusions. This diversity is, among others, due to the ongoing need of establishing a robust AR definition (Gimeno et al., 2021) and to the vast variety of questions that these ARDTs were developed to answer. The Atmospheric River Tracking Method Intercomparison Project (ARTMIP, Shields et al. (2018)) aims to quantify the uncertainties in AR climatology based on detection algorithms alone and to provide guidance on the most appropriate algorithm for a given science question or study region. ARTMIP also states the need of creating a common software infrastructure and classifying ARDTs to understand the broad uncertainty in AR detection results. The outcomes of ARTMIP Tier 1 phase addressed these topics and were summarized in Rutz et al. (2019). It was found that threshold values were the main contributors to AR uncertainty. For instance, an IVT magnitude greater than 250 kg m$^{-1}$ s$^{-1}$ and a length over 2,000 km would be considered an AR according to some algorithms (Zhu and Newell, 1998) but not to others. Percentiles of the IVT or IWV fields, typically the $85^{th}$ or $90^{th}$ percentile, have also been utilized (Lavers et al., 2012). ARTMIP Tier 2 conducted several AR detection sensitivity analyses to reanalysis products, such as MERRA-2 or ERA5 (Collow et al., 2022), and under climate change scenarios (O'Brien et al., 2022; Shields et al., 2023), including their impacts on AR-related precipitation. They found that the ARDT selection is the main contributor to the uncertainty in projected AR frequency. Therefore, climate change studies should consider using more than a single ARDT and assessing their uncertainties. The Third ARTMIP Workshop (O'Brien et al., 2020) contemplates the existence of different "flavors" of ARs, although most tracking methods have not considered this possibility yet. Future AR researches would also be able to apply machine-learning techniques easily.

These algorithms are mainly applied to the outputs of global climate models (GCMs) and reanalysis (Gimeno et al., 2014). Therefore, most of them consist of detecting the arrival of the potential AR and spatially tracking its elongated 2D structure until it is delimited for that fixed time (e.g., Brands et al. (2017)). However, inaccuracies in the forecasted precipitation intensity,

frequency and spatial variability due to the arrival of an AR on a local scale are encountered. As ARs interact with orography
on a regional scale, GCMs can represent ARs but may not accurately reproduce AR-related precipitation (Lorente-Plazas et al.,
2018). In such cases, the use of regional climate models (RCMs) with higher resolution can provide a better understanding of
ARs. This is the case of the IP (Gröger et al., 2022), characterised by a complex orography. Nevertheless, it should be taken
into account that the spatial tracking given a fixed time step method may not be suitable for data obtained from RCMs whose
spatial limits are very close to the detection area. This is the case for most of the RCM runs, as they are primarily land-focused.

Several researchers have investigated the role of ARs and similar structures in the global transport of atmospheric aerosols
(Chakraborty et al., 2021). However, the isolated impact of these aerosols and their variability on the formation, characteristics
and behavior of ARs has received less attention. One of the most important studies concerning this issue at global scale
was carried out by Baek and Lora (2021). It uncovered opposite influences of industrial aerosols, which weakened ARs, and
greenhouse gases, which strengthened them. Another relevant research, conducted by Naeger (2018), explored the impact of
long-range transported dust aerosols on the precipitation related with a specific AR over the western United States.

Aerosols, both natural and anthropogenic, interact with incoming solar radiation by absorption and scattering processes
(direct aerosol effects). On a global scale, scattering has a net cooling effect on the surface (Jerez et al., 2021; Glassmeier
et al., 2021). However, the regional impacts may differ significantly depending on the type of aerosol (Palacios-Peña et al.,
2019; Palacios-Peña et al., 2020; Li et al., 2022). Additionally, direct effects can derive in thermodynamic alterations of cloud
properties, leading to subsequent changes in radiative forcing (semi-direct effects (Hansen et al., 1997)). Moreover, aerosols
interact with clouds, acting as Cloud Condensation Nuclei (CCN), affecting cloud albedo (Twomey effect (Twomey, 1977), first
indirect effect) and cloud lifetime (Albrecht effect (Albrecht, 1989), second indirect effect), as well as precipitation (López-
Romero et al., 2021; Sun and Zhao, 2021).

RCMs typically introduce aerosol species and their concentrations in a prescribed manner (Forkel et al., 2015), neglecting
changes in their concentration and interactions with radiation and cloud microphysics, thus not taking into consideration some
important feedback processes, like changes in the CCN concentration due to precipitation or the modification of cloud droplets
properties based on the aerosol type acting as CCN. In contrast, a coupled online approach for aerosol calculation allows
the effects of aerosols on radiation and clouds (i.e., direct, semi-direct, and indirect effects) to be quantified from a climate
perspective (López-Romero et al., 2021). This approach can lead to variations between simulated meteorological situations
and those obtained by using a prescribed aerosol configuration, potentially resulting in changes in the frequency, intensity, or
landfall areas of ARs and their consequences across all sectors, both environmental and socio-economic.

The primary objective of this study is to evaluate the influence of atmospheric aerosols and their interactions with solar
radiation and cloud microphysics processes on ARs that impact the IP region. To achieve this goal, a comparative analysis
of regional climate simulations with different levels of interactions between aerosols, radiation, and cloud microphysics is
required.

This study develops and employs a novel AR identification algorithm that processes data from regional-scale simulations
to verify its accuracy for the IP region. The developed identification algorithm is then applied to three experiments: BASE,
ARI, and ARCI. The BASE simulation uses a prescribed aerosol configuration and serves as the reference. The ARI simulation

includes dynamic aerosol-radiation interactions (i.e., direct and semi-direct effects). Finally, the ARCI experiment includes

aerosol interactions with both radiation and cloud microphysics, accounting for direct, semi-direct, and indirect effects. By comparing the results of these experiments, we can evaluate the impact of atmospheric aerosols and their interactions on ARs affecting the IP region.

## 2   Methods

### 2.1   Data

The data used in this study were derived from the REPAIR project, which involved regional climate simulations for Europe spanning the period 1991-2010 at hourly resolution (López-Romero et al., 2021). The WRF-Chem model (v.3.6.1) was used for the simulations, both in a decoupled configuration (WRF alone (Skamarock et al., 2008)) and in a fully coupled configuration with atmospheric chemistry and pollutant transport to account for aerosol-radiation and aerosol-cloud interactions (Grell et al., 2005). The initial and boundary conditions were obtained from the ERA20-C reanalysis.

The spatial configuration included two one-way nested domains, with the inner domain covering Europe with a resolution of 0.44° in latitude and longitude following the Euro-CORDEX recommendations (Jacob et al., 2014). The outer domain had a spatial resolution of about 150 km and extended southwards to a latitude of 20° N to encompass major dust emission areas (the Sahara desert and its surroundings), which were incorporated into the inner domain via boundary conditions as in Palacios-Peña et al. (2019). Nudging was used for the outer domain in order to minimize the internal variability of the model. The

boundary conditions for the outer domain were updated every 6 hours and the model outputs were recorded every hour. The vertical domain comprised 29 non-uniform sigma levels with higher resolution near the surface, subsequently interpolated to pressure levels. The upper boundary was set at the 50 hPa level.

     The physics configuration included the Lin microphysics scheme (Lin et al., 1983), the Noah land surface layer (Tewari et al., 2004), the RRTM radiative scheme for both short- and longwave (Iacono et al., 2008), the Grell 3D ensemble cumulus

scheme (Grell, 1993; Grell and Dévényi, 2002), and the University of Yonsei boundary layer scheme (Hong et al., 2006).

     Three experiments were considered in this study, each of which included different aerosol interactions. The complete description of these three simulations can be found in López-Romero et al. (2021). The BASE experiment served as a reference, with aerosols not treated interactively in the model, but prescribed with an Aerosol Optical Depth (AOD) set to zero and 250 CCN cm$^{-3}$ considered in each domain cell. This experiment did not account for the effects of aerosols on radiation and cloud

microphysics. In the ARI experiment, aerosols were treated online, introduced as an active fully coupled component, and the aerosol-radiation interactions were activated in the model (Fast et al., 2006). The CCN concentration was the same as that in the BASE experiment. Thus, this simulation only accounted for the direct and semi-direct effects of aerosols. In the ARCI experiment, aerosol interactions with cloud microphysics (indirect effects) were also activated.

     In the ARI and ARCI experiments, aerosols were calculated in the WRF-Chem model using a coupled approach, where the

model solved the aerosol dynamics online, allowing it to incorporate its own aerosols based on variables such as soil type, vegetation, and wind at each point of the domain (López-Romero et al., 2021). The gas-phase chemical mechanism RACM-

KPP (Stockwell et al., 1997; Geiger et al., 2003) used in the model was coupled to the GOCART aerosol module (Ginoux et al., 2001; Chin et al., 2002), which considers five aerosol species: sulphates, mineral dust, sea salt, organic matter, and black carbon. The Fast-J module (Fast et al., 2006) was used for photolysis and the Guenther scheme (Guenther et al., 2006) was employed for biogenic emissions. Anthropogenic emissions were derived from the Atmospheric Chemistry and Climate Model Intercomparison Project (ACCMIP) (Lamarque et al., 2010) and did not vary during the simulations. However, natural emissions are dependent on meteorological conditions and thus change over time (Jiménez-Guerrero et al., 2013).

The WRF-Chem model's aerosol-radiation-cloud interactions were explained in detail by Palacios-Peña et al. (2018). To calculate aerosol-radiation interactions, each species was associated with a complex refractive index. Mie's theory was used to obtain the optical properties of the aerosols in each cell by summing up the contributions from all aerosol sizes and species, which were then incorporated into the solar radiation scheme. The ARCI experiment's description and aerosol validation results were reported in Palacios-Peña et al. (2020). The WRF-Chem model facilitated converting the single-momentum Lin parameterization into a double-momentum one, essential for the comprehension of aerosol indirect effects. This microphysics approach involves six species: water vapor, cloud water, rain, cloud ice, snow, and graupel (Ghan et al., 1997). The conversion of cloud droplets into rain droplets depends on the droplet number (Liu et al., 2005). The rates of droplet nucleation and evaporation represent aerosol activation and resuspension rates. Although the experiments did not consider ice nuclei based on forecasted aerosols, a prescribed ice nuclei distribution was used to include ice clouds. Radiation-cloud interactions were included by connecting the number of simulated cloud droplets to the Goddard solar radiation scheme (Chou and Suarez, 1999), representing the first indirect effect (i.e., increased droplet number due to increases in aerosols), and to Lin's microphysics parameterization, representing the second indirect effect (i.e., decreased precipitation efficiency related to increases in aerosols). Consequently, the number of droplets affected both their mean radius and the optical depth of the cloud.

## 2.2 The AR identification algorithm: AIRA

The identification of ARs on a global scale may not apply to regional climate simulations due to the limited spatial domain. Consequently, it would be impossible to determine the complete length of an AR if the regional domain is not sufficiently wide to track the AR structure for a fixed time. Many ARDTs employ this method (e.g., Brands et al. (2017), Ramos et al. (2016a)). To address this issue, a new AR identification algorithm, named AIRA, has been developed. AIRA is designed to work with regional data and utilizes Integrated water Vapor Transport (IVT) as its basis. IVT is a horizontal vector that is defined by Equations 1 and 2, where $q$ represents the specific humidity, $u$ and $v$ refer to the zonal and meridional wind components and $g_0 = 9.81$ m s$^{-2}$ is the gravity acceleration at sea level.

$$IVT_u = \frac{1}{g_0} \int_{1000\,hPa}^{300\,hPa} qu \, dP \tag{1}$$

$$IVT_v = \frac{1}{g_0} \int_{1000\,hPa}^{300\,hPa} qv\,dP \qquad (2)$$

Equations 1 and 2 can be numerically solved by calculating the sum of the product of specific humidity and wind component values at a given pressure level, and the pressure increment between that level and the level below it (Equation 3). The magnitude and direction of the IVT with respect to the East can then be easily obtained. The magnitude of the IVT is expressed in kg m$^{-1}$ s$^{-1}$, while its direction is calculated in degrees East and is positive in the counter-clockwise direction. AIRA primarily utilizes the magnitude and direction of the IVT in its functioning.

$$IVT_u = \frac{1}{g_0} \sum_i q_i\,u_i\,\Delta P_i, \quad IVT_v = \frac{1}{g_0} \sum_i q_i\,v_i\,\Delta P_i \qquad (3)$$

AIRA relies on two fixed longitude detection lines to identify ARs. The main detection line, referred to as line 1 (L1), is located closest to the region of interest and is used to analyze the magnitude of the IVT. The second line (L2) serves as an auxiliary line to study the direction of potential AR candidates that pass through L2 before reaching L1. Figure 1 illustrates the detection lines utilized in this study.

AIRA is structured into two main blocks, with the first block encompassing data preprocessing and initial filtering, while the second block involves the identification and filtering of AR candidates and is subdivided into two stages. The first stage, Part A of the algorithm, determines the time intervals in which the IVT threshold has been consecutively exceeded. In the second stage, Part B, each of these intervals is evaluated against a set of conditions to determine whether an AR has been identified. Both blocks are elaborated upon in detail in the following subsections.

### 2.2.1 Data preprocessing

Figure 2 illustrates the data preprocessing stage of AIRA. First, the magnitude and direction of the IVT are bi-linearly interpolated to the detection lines, L1 and L2, enabling the computation of the geometry and magnitude variables required later in the algorithm. Next, the maximum IVT detected on L1, denoted as $IVT_1$, is identified for each time step $t$, and a first filter is applied to detect potential ARs. This filter applies a threshold value $\Gamma$ to the IVT magnitude. $\Gamma$ is an absolute threshold established by the user. Section 3.1 contains specific information about the AIRA implementation in this study. If $IVT_1(t) \geq \Gamma$, then $t$ is considered as a time step with a potential AR. At this time step $t$, AIRA also determines the latitude of the maximum IVT on L1, denoted as $\phi_1$, and the direction of the IVT at that point, denoted as $D$. Additionally, AIRA computes the latitude of the minimum value of IVT on L1 that still exceeds the threshold, denoted as $\phi_{min}$, and the latitude of the maximum value of IVT on L2, denoted as $\phi_2$. This information is necessary to estimate the direction, $d$, and width, $w$, of the potential AR. A diagram displaying the trigonometric elements utilized to calculate the aforementioned parameters is available in Fig. A1.

The direction of the potential AR at time step $t$, denoted as $d$, can be computed using Equation 4, where $\phi_1$ and $\phi_2$ are the latitudes of the maximum IVT value on L1 and L2, respectively. The distance in kilometres corresponding to one degree of

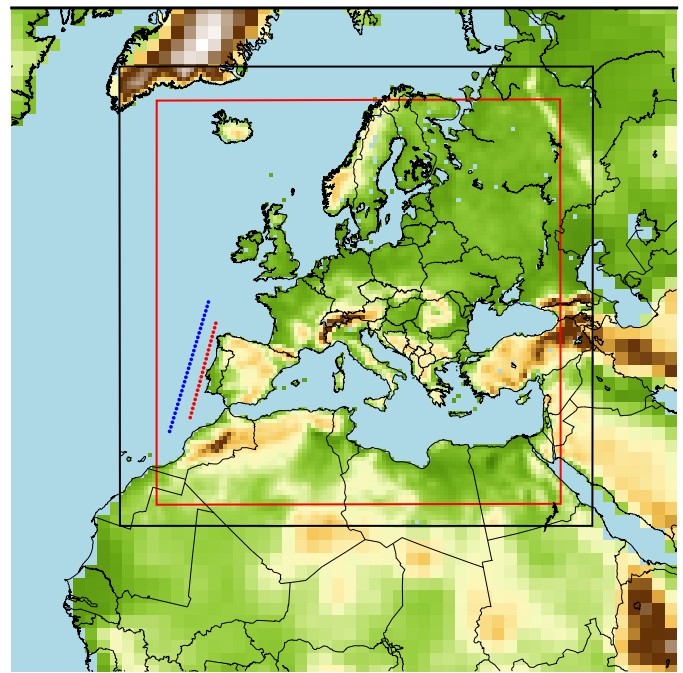

**Figure 1.** Spatial configuration of the three experiments, consisting of two one-way nested domains. The outer domain has a resolution of about 150 km and the inner Euro-CORDEX domain, which is boxed in black, has a resolution of $0.44°$. The area between the black and red boxes approximately represents the blending area of the inner domain. Identification line 1 (red) consists of 22 equidistant points between $34°$ N and $44.5°$ N located at a longitude of $-10°$ E. Identification line 2 (blue) consists of 30 equidistant points between $32°$ N and $46.5°$ N located at a longitude of $-12°$ E.

latitude is assumed to be constant at 111.20 km, while the equivalent distance for a degree of longitude varies with the latitude $\phi$. The Earth radius is represented by $R_T$, and an average value of $R_T = 6371$ km is assumed. $\Delta\ell$ represents the longitude difference between the detection lines. All the trigonometric functions in the following equations are defined in sexagesimal degrees, resulting in $d$ being obtained in degrees East.

$$d = \tan^{-1}\left(\frac{111.20(\phi_1 - \phi_2)}{\frac{\Delta\ell\,\pi}{180} R_T \cos\phi_2}\right) \tag{4}$$

The width of the potential AR is calculated under the assumption of its symmetry in section, meaning that the maximum is located at the center with equal distances to both lateral boundaries where the IVT threshold is still exceeded. Hence, the width detected on L1, $a$, expressed in degrees North, can be determined by using Equation 5. However, ARs usually do not pass through the detection lines with a perfectly perpendicular trajectory. As such, the actual width $w$ of the potential AR that would be detected can be estimated using Equation 6.

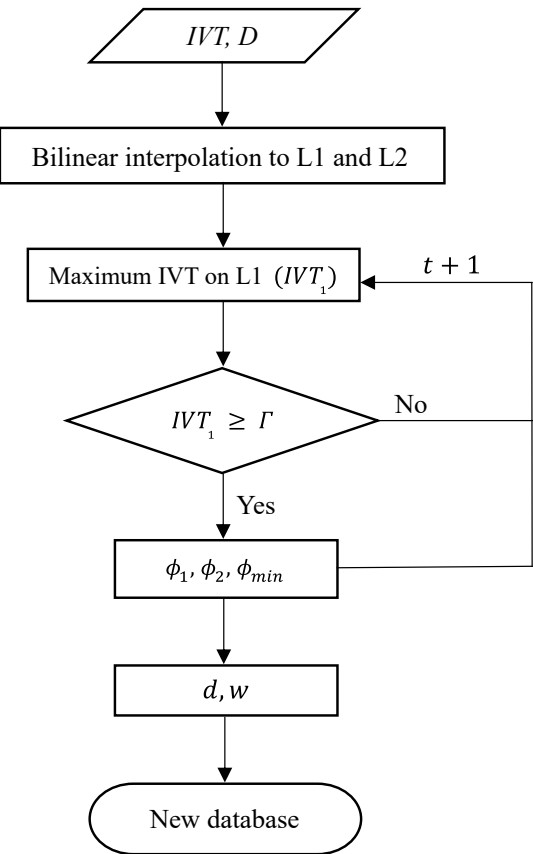

**Figure 2.** Diagram of the data preprocessing. The magnitude and eastward direction of the IVT are bilinearly interpolated to the detection lines. The maximum IVT on L1 is located for each time step and if it exceeds the established IVT threshold $\Gamma$, it is considered as part of an AR candidate. Thus, the maximum IVT, the latitudes of the IVT maximum and minimum, the IVT direction, the AR width and its direction through the lines are recorded in a new database.

$$a = 2|\phi_1 - \phi_{min}| \tag{5}$$

$$w = 111.20\, a \cos d \tag{6}$$

Finally, at each time step $t$, the date and time are associated with the values obtained for the variables of interest, namely $IVT_t$ (the maximum IVT detected on L1), $D$, $\phi_1$, $\phi_2$, $\phi_{min}$, $d$ and $w$. These values are recorded in a new database. The procedure is then repeated for the next time step, and so on until the entire study period is covered. If $IVT_1(t) < \Gamma$, the current time step is skipped as an AR cannot be considered to be passing through L1, and the procedure moves on to the next time step until the entire study period is covered.

### 2.2.2 Identification and filtering: parts A and B

Part A of the algorithm involves delimiting the time intervals in which the maximum IVT magnitude exceeds the threshold value $\Gamma$ consecutively (Fig. 3 (left)). To accomplish this, the data obtained in the data preprocessing stage is augmented with the remaining time steps where $IVT_1(t) < \Gamma$, and all their variables are set to zero except for the date and time.

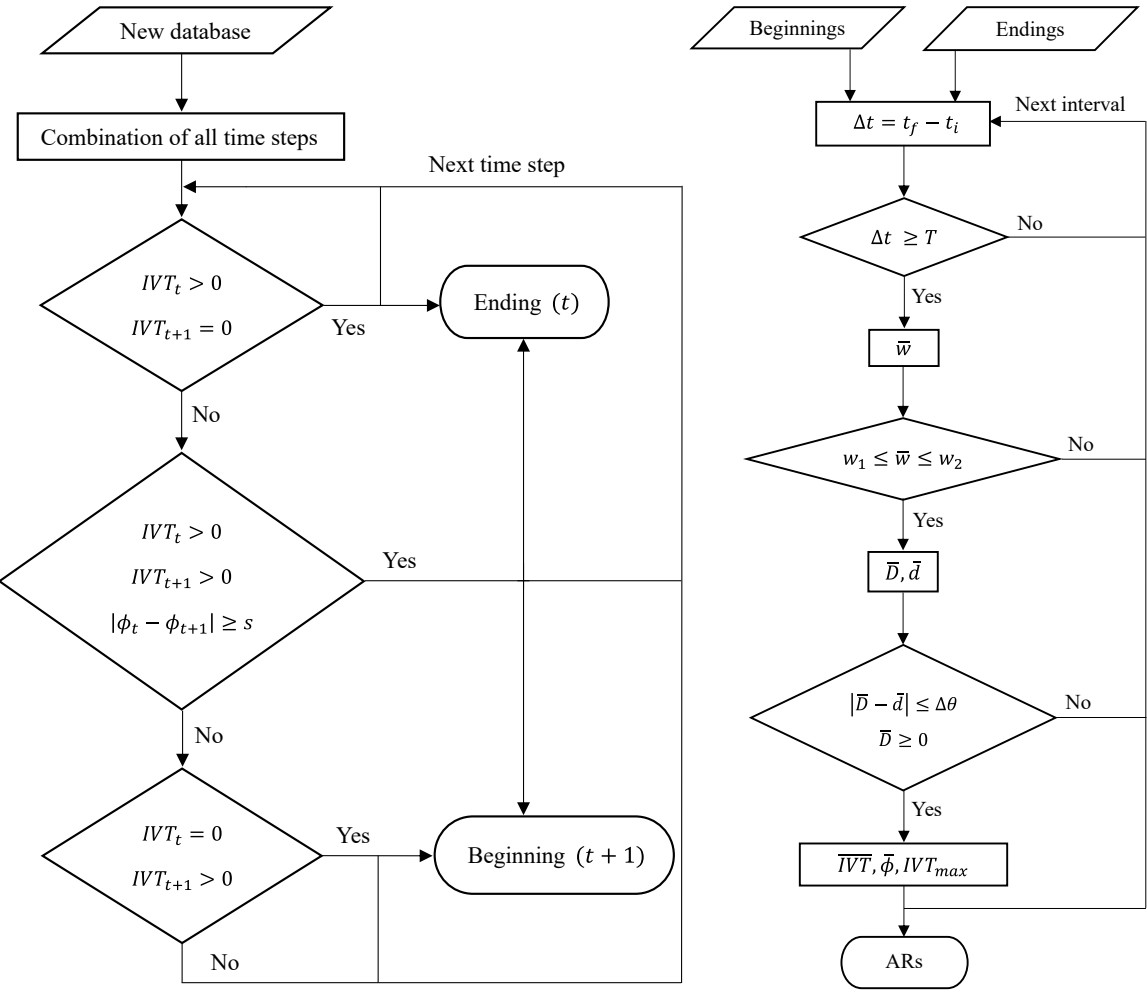

**Figure 3.** Diagram of AIRA. (Left) Part A. The database obtained in the data preprocessing is completed with the remaining time steps and the intervals candidate to AR are delimited with their beginning and ending. (Right) Part B. Each AR candidate interval is analysed separately. An AR is considered to be detected if the interval has at least a minimum duration, its average width is between a lower and upper limits and the direction of the IVT is positive and similar to the direction in which the potential AR passes through the detection lines. In the final database, each identified AR is recorded with 13 variables of interest.

To characterize the different time intervals, two lists are employed, with one recording the beginning time steps and the other recording the ending time steps. Moreover, the possibility of consecutive arrivals of two atmospheric rivers is considered in

the algorithm. If a significant shift ($s$) in latitude is detected between $\phi_1(t)$ and $\phi_1(t+1)$ an interval is considered to end and another to begin.

Each interval obtained in Part A of the algorithm represents a potential AR. Part B of the algorithm (Fig. 3 (right)) aims to determine whether these intervals meet a set of criteria to be considered as ARs. Firstly, the IVT threshold, $\Gamma$, must be exceeded on L1 for at least T consecutive time steps, which allows for the estimation of the AR length. Additionally, the average width of the potential AR, $\bar{w}$, must fall within the limits of $w_1$ and $w_2$ to ensure its filamentary structure. Moreover, the mean direction of the IVT, $\bar{D}$, must be positive and match the mean direction through the identification lines, $\bar{d}$, within a range of $\Delta\theta$, to ensure moisture transport is occurring. For the Southern Hemisphere, the condition for $\bar{D}$ changes to $\bar{D} \leq 0$, but both conditions can be disabled in the AIRA setup parameters. Finally, potential ARs with a mean latitude corresponding exactly to the limits of L1 are excluded since they occur outside the study domain.

The AR database obtained includes 13 variables, providing detailed information about each AR detected. For each AR, the database records the start and end date and time, the initial and final time-step indices, the number of time-steps $\Delta t$ of the AR, its mean impact latitude ($\bar{\phi}$), the mean intensity ($\overline{IVT}$), the maximum intensity ($IVT_{max}$), the mean width ($\bar{w}$), the mean direction of the IVT ($\bar{D}$), and the mean direction of the AR passing through the detection lines ($\bar{d}$).

In the ARTMIP context (Shields et al., 2018), AIRA would be classified as a condition ARDT that imposes an absolute IVT threshold to determine if an AR could be present on the detection lines at a given time slice over the IP region. Then, all the consecutive potential AR time slices are gathered into potential AR intervals with a minimum time stitching $T$ and the geometry requirements (width, direction) are imposed to the mean values of the intervals. Throughout this process, the trigonometric elements employed are derived from just two close points: the locations of maximum IVT over L1 and L2. No spatial tracking is required. This is the main difference between AIRA and other ARDTs that look at ARs over the IP or west Europe. For instance, the IDL ARDT (Ramos et al., 2016a) uses an IVT threshold (relative instead of absolute) to identify the arrival of a potential AR to a detection line. However, once the threshold is exceeded, this ARDT performs an east-west analysis to spatially determine the AR spine and impose a minimum length. A similar approach was employed by Lavers et al. (2012) and Brands et al. (2017). The innovation of the AIRA approach relies on overcoming the RCMs limitations where most of the runs are focused over land, and this may preclude capturing the long AR structure over the ocean. AIRA was designed to work even in regions located very close to the domain borders, as it only employs data over two line grids.

## 3 Results and discussion

### 3.1 AIRA implementation and application

To focus on ARs landfalling on the IP from the west, this study considered identification lines shown in Fig. 1. L1, situated at a longitude of -10° E, is the nearest line to the Iberian west coast, comprising 22 points with latitude increasing from 34° N to 44.5° N in steps of 0.5° N. L2, on the other hand, has 30 points and is positioned at a longitude of -12° E, with the latitude range between 32° N and 46.5° N, also increasing by 0.5° N in steps.

Before implementing the algorithm, it is necessary to determine the values of the parameters involved. The same values were used in the application of AIRA for the three simulations (Table 1). Firstly, the mean of the $99^{th}$ percentile of the IVT magnitude on L1 for all time steps resulted in a value close to 260 kg m$^{-1}$ s$^{-1}$ in all three experiments. The computation of this value using only the data with the 12:00 h time stamp would have resulted in a higher IVT, as seen in Lavers and Villarini (2013) or Ramos et al. (2016a). To ensure the identification of ARs occurring in summer, a higher IVT threshold of $\Gamma$ = 300 kg m$^{-1}$ s$^{-1}$ was selected. Secondly, the minimum time duration for an interval to be classified as an AR was set at T = 10 h. Given that the mean wind speed associated with ARs in the study area is around 30 m s$^{-1}$, this minimum duration would indicate the occurrence of an AR approximately 1,000 km in length. Another condition was established to ensure the filamentary structure of the ARs, which are long and narrow. Considering a minimum length of approximately 1000 km, it was established that the width of the ARs must be between $w_1 = 150$ km and $w_2 = 800$ km.

**Table 1.** Imposed values for the AIRA parameters: IVT threshold, $\Gamma$; latitude shift between two consecutive ARs, $s$; minimum interval duration, T; lower and upper width limits, $w_1$ and $w_2$, and maximum deviation between directions, $\Delta\theta$.

| $\Gamma$ (kg m$^{-1}$ s$^{-1}$) | $s$ (°N) | T (h) | $w_1$ (km) | $w_2$ (km) | $\Delta\theta$ (°) |
|---|---|---|---|---|---|
| 300 | 10 | 10 | 150 | 800 | 25 |

The application of AIRA to the three simulations (BASE, ARI and ARCI) resulted in three databases with the information of the identified ARs during the two decades from 1991 to 2010. In the reference experiment, BASE, a total of 244 ARs were detected. The AIRA algorithm identified 248 ARs in the ARI experiment and 250 ARs in the ARCI simulation.

It was found that most of the ARs identified by AIRA also matched those identified by global-scale algorithms. AIRA's outcomes were compared against the results of Brands et al. (2017) ARDT for ERA20-C data over W Iberia region (Brands, 2023). Specifically, the daily JFMOND performance of both algorithms, i.e., whether an AR was present over western Iberia during a JFMOND day, displayed similar results in 82.1 %, 81.6 % and 80.9 % of the total days for BASE, ARI and ARCI, respectively. Discrepancies could be mainly due to differences in the identification approach. Brands Method 0 employed the $95^{th}$ percentile to detect the AR arrival and the $85^{th}$ percentile to perform the spatial tracking of the AR structure, imposing a minimum AR length of 2000 km. In addition, its detection region for W Iberia did not extend to the most southern latitudes of the IP, as they were considered as a different region. Furthermore, aerosol effects may cause spatial deviations, potentially pushing ARs out of the identification area and lowering the number of coincidences from simulation to simulation.

### 3.1.1 Sensitivity assessment

Using a single ARDT can entail some limitations when studying ARs. ARTMIP has conclusively demonstrated that the thresholds selection constitutes the principal source of variability in AR metrics across different ARDTs, resulting in substantial variations in frequency, depending on the chosen criteria (Rutz et al., 2019). Among the different parameters, the IVT threshold was reported to be the main contributor to the uncertainty. To address this variability, an analysis of the sensitivity to the

IVT threshold given a fixed minimum duration and the sensitivity to the duration threshold given a fixed $\Gamma$ was performed (Tables 2 and 3). The values of the remaining AIRA parameters were identical to those presented in Table 1.

Lowering the IVT threshold decreased the number of ARs but increased their duration due to the possibility of multiple closely timed events being identified as a single, longer event. Conversely, raising the IVT threshold above 300 kg m$^{-1}$ s$^{-1}$ resulted in a decrease in the mean duration of the ARs but had little effect on the number of ARs itself. For example, the selection of an IVT threshold of 400 kg m$^{-1}$ s$^{-1}$ would have led to a decrease in the number of identified ARs in BASE, ARI and ARCI of 2.5 %, 5.6 %, and 6.8 %, respectively. As for the duration threshold, increasing the value of the minimum duration

criteria resulted in a lower number of identified ARs.

**Table 2.** Sensitivity analysis to the IVT threshold, given a fixed minimum duration (T = 10 h), of the number of ARs identified in the three simulations, the number of common (COM) AR events, the percentage of common AR time-steps and the mean intensity and duration of the ARs of the three simulations.

| T = 10 h | $\Gamma$ (kg m$^{-1}$ s$^{-1}$) | | | | | | | | |
|---|---|---|---|---|---|---|---|---|---|
| | 200 | 225 | 250 | 275 | **300** | 325 | 350 | 375 | 400 |
| **ARs BASE (#)** | 194 | 212 | 230 | 236 | **244** | 245 | 252 | 244 | 238 |
| **ARs ARI (#)** | 166 | 195 | 210 | 217 | **248** | 254 | 247 | 230 | 234 |
| **ARs ARCI (#)** | 173 | 205 | 222 | 232 | **250** | 244 | 243 | 230 | 233 |
| **ARs COM (#)** | 39 | 54 | 63 | 73 | **80** | 92 | 94 | 86 | 91 |
| **COM time-steps (%)** | 24.79 | 28.51 | 32.11 | 33.65 | **37.16** | 40.54 | 38.91 | 38.11 | 40.65 |
| $\overline{IVT}$ **BASE (kg m$^{-1}$ s$^{-1}$)** | 344.42 | 380.58 | 407.61 | 435.15 | **469.20** | 495.67 | 523.24 | 549.25 | 579.26 |
| $\overline{IVT}$ **ARI (kg m$^{-1}$ s$^{-1}$)** | 345.14 | 373.10 | 407.43 | 440.88 | **465.47** | 491.42 | 520.35 | 551.15 | 589.44 |
| $\overline{IVT}$ **ARCI (kg m$^{-1}$ s$^{-1}$)** | 347.59 | 377.23 | 404.54 | 434.99 | **459.18** | 490.43 | 517.50 | 550.55 | 574.19 |
| $\overline{d}$ **BASE (h)** | 53.17 | 50.73 | 47.54 | 45.36 | **42.55** | 40.44 | 40.11 | 37.75 | 36.35 |
| $\overline{d}$ **ARI (h)** | 56.76 | 52.44 | 51.24 | 47.47 | **43.13** | 41.26 | 39.00 | 37.69 | 36.46 |
| $\overline{d}$ **ARCI (h)** | 56.61 | 53.45 | 48.71 | 46.72 | **43.79** | 43.10 | 41.82 | 40.03 | 36.76 |

## 3.2 Climatology of the identified ARs

The number of ARs detected using AIRA is consistent across the three experiments, with between 5 and 15 ARs detected per year on average. However, there are exceptions to this pattern, such as in 1994 and 2007-2008, where the number of ARs is slightly lower. Furthermore, the total number of ARs identified per month exhibits a significant decline in July and August, but

the number of detections increases in other seasons, particularly in spring and autumn. Notably, the highest number of ARs is detected in October, with at least 30 ARs identified in all three simulations (Fig. 4 (top)). This result is consistent with the findings of Rutz et al. (2019).

**Table 3.** Sensitivity analysis to the minimum duration threshold, given a fixed IVT threshold ($\Gamma = 300$ kg m$^{-1}$ s$^{-1}$), of the number of ARs identified in the three simulations, the number of common AR events and the percentage of common (COM) AR time-steps.

| $\Gamma$ = 300 kg m$^{-1}$ s$^{-1}$ | T (h) | | | | | | | | |
|---|---|---|---|---|---|---|---|---|---|
| | 4 | 6 | 8 | **10** | 12 | 14 | 16 | 20 | 24 |
| **ARs BASE (#)** | 267 | 262 | 253 | **244** | 233 | 222 | 209 | 183 | 162 |
| **ARs ARI (#)** | 261 | 259 | 254 | **248** | 232 | 225 | 212 | 193 | 170 |
| **ARs ARCI (#)** | 267 | 261 | 254 | **250** | 233 | 226 | 212 | 198 | 171 |
| **ARs COM (#)** | 86 | 85 | 84 | **80** | 74 | 69 | 65 | 58 | 50 |
| **COM time-steps (%)** | 35.73 | 35.83 | 35.94 | **37.16** | 36.19 | 36.12 | 35.97 | 36.41 | 36.75 |

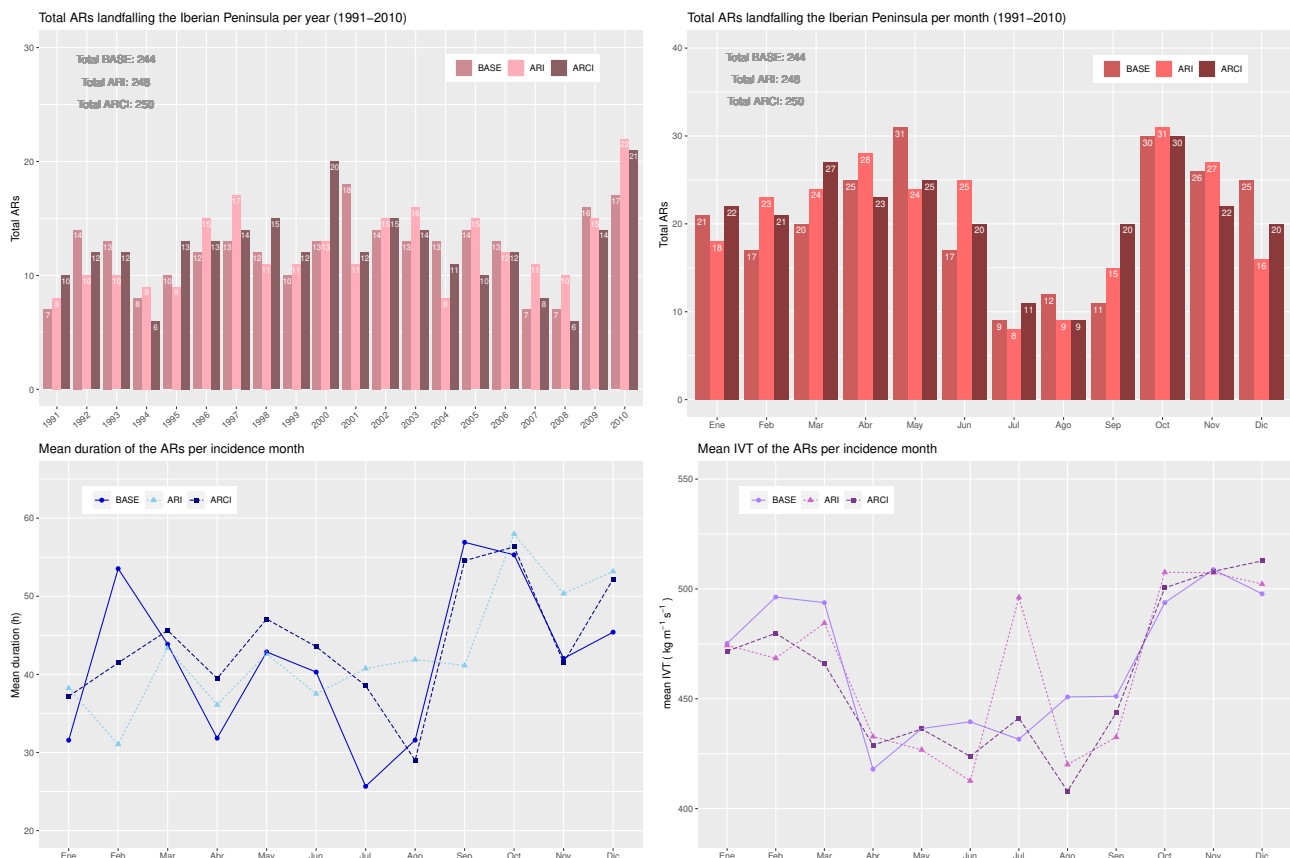

**Figure 4.** (Top) Histograms of the total ARs landfalling the IP per year (1991-2010) and per month in the three experiments. (Bottom) Mean duration and mean IVT magnitude of the ARs per month.

In addition, the mean duration of ARs varies depending on the month of occurrence, and it is observed to be longer than one day, approximately equivalent to 2,600 km, and in some cases, even longer than two days. September and October showed the most persistent ARs while the summer months had the shortest duration for ARCI and BASE. In ARI, the longest events took place in October and December, while the minimum duration occurred in February. The mean intensity of the identified ARs, which is the mean magnitude of the maximum IVT on L1, exhibits similar behavior to that of the duration for all three experiments. The least frequent, shortest, and weakest ARs occurred in the summer, while the most frequent, longest, and most intense ARs occurred in autumn (Fig. 4 (bottom)).

The ARs identified by AIRA had an average width ranging from 200 to 500 km, with the highest frequency close to 200 km in BASE and ARCI, and close to 300 km in the ARI experiment. The majority of the ARs lasted between 10 and 50 hours, with the highest frequency around 20 hours. However, there were a few persistent events that lasted more than 170 hours. The ARs generally had a mean direction between $30°$ and $50°$, with the highest frequency around $40°$, though some cases of inclination less than $10°$ were also identified. The incidence of ARs was minimum above $36°$ N, with no clear maximum. Most of the identified ARs had a mean intensity between 320 and 500 kg m$^{-1}$ s$^{-1}$, although some of them exceeded 700 kg m$^{-1}$ s$^{-1}$ on average. Additionally, the majority had a maximum IVT between 350 and 800 kg m$^{-1}$ s$^{-1}$, with the highest frequency around 500 kg m$^{-1}$ s$^{-1}$. However, some cases exceeding 1,200 kg m$^{-1}$ s$^{-1}$ were identified in all three simulations.

### 3.2.1 Associated ratio of the total precipitation

In order to estimate the percentage of total accumulated precipitation that could be related to the presence of ARs, the precipitation recorded on a given day was considered to be due to an AR if its presence has been detected in at least one hour of that day.

In all three simulations, it is apparent that the maximum percentage of total precipitation attributable to the presence of ARs is close to 30 % and occurs along the western Iberian coast, which is the impact zone of the ARs (Fig. 5). In Galicia, located in the northwest region of the IP, this percentage is slightly lower owing to the greater amount of precipitation that is associated with other phenomena, like cold fronts. These results are similar to those obtained by Gao et al. (2016) and Gröger et al. (2022) at the regional-scale, and consistent with the findings of Baek and Lora (2021) for the IP at global-scale. The ARCI experiment exhibits the highest percentage for the entire domain, which can be attributed to the interactive introduction of various types and concentrations of aerosols acting as CCN in this simulation. Additionally, in maritime regions, aerosol concentrations are lower than over land, and highly hygroscopic aerosols predominate. Consequently, droplets grow more rapidly, resulting in increased precipitation (Pravia-Sarabia et al., 2022). The overall rainfall changes due to aerosols effects using the same set of regional simulations (BASE, ARI and ARCI) were analyzed by López-Romero et al. (2021).

### 3.3 Common AR events

To study the potential differences between the ARs of the three experiments, a one-to-one comparison of their coherent AR events was designed. Each coherent interval reproduces the same forecast period but with three different aerosol treatments. The common AR intervals have been identified by applying AIRA to the common time steps, eliminating coincident intervals

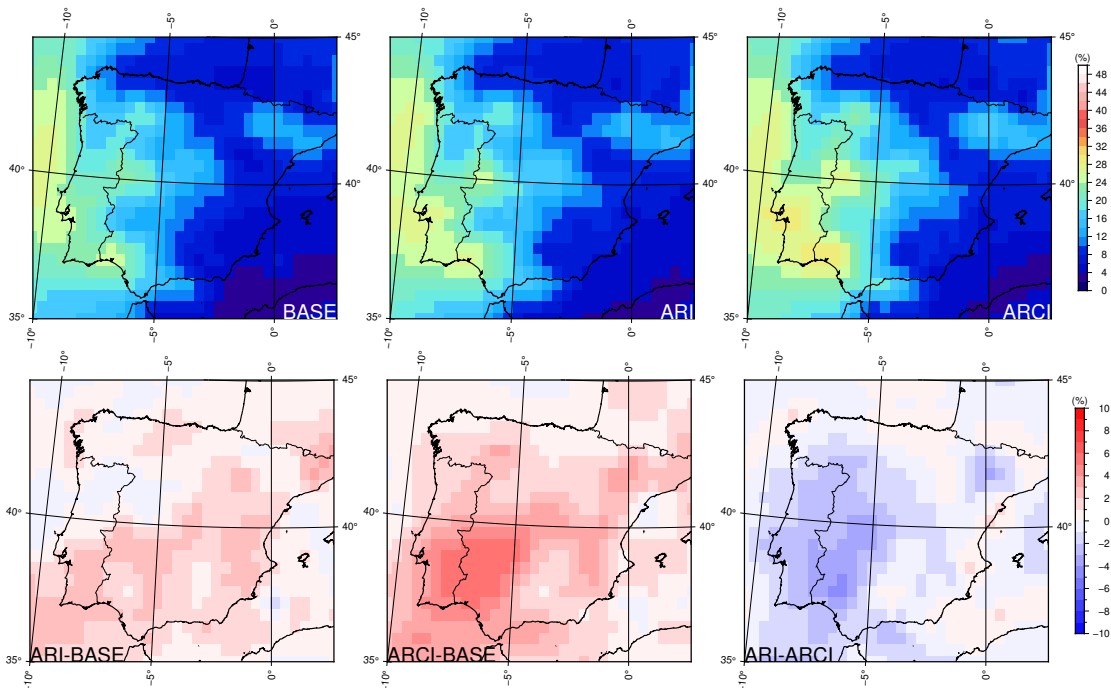

**Figure 5.** (Top) Percentage of total precipitation explained by the influence of ARs in each simulation for the period 1991-2010 and (bottom) differences between simulations.

with a duration of less than 10 hours or not satisfying the other criteria to be deemed proper ARs. As a result, a total of 80 common AR intervals from the three experiments were obtained, and their characteristics were compiled. These common AR events represent only the 37 % of the time steps with ARs concerning the BASE total. This low percentage could be attributed to weak events and the temporal limitations of the identified ARs, where the IVT threshold is exceeded in some simulations but not in others. Furthermore, aerosol effects can cause spatial deviations, as seen in the following sections, potentially pushing ARs out of the study area, decreasing the time steps with AR on the detection lines in some experiments, and thus lowering the coincidence percentage. The fewest number of common events occurred in summer, with only one occurrence in July or August, while the most frequent events occurred in March (13 intervals) and October (11 common AR events).

### 3.3.1 Analysis of the differences

The common AR events from the three simulations show that there are no significant deviations between the ARs in the strongest events. However, some differences are observed in the mean direction and impact latitude of the ARs in other cases. To investigate this further, the ARI-BASE and ARCI-BASE differences of the mean IVT direction, mean impact latitude, and mean IVT magnitude were plotted against the mean maximum IVT (Fig. 6). The mean maximum IVT was obtained by averaging the maximum IVT intensity of each AR event in the three experiments. The spatial deviations (latitude and direction differences) tended to zero, and the ARI-ARCI differences of the three considered variables (latitude, direction and mean IVT)

became minimal in the most intense events. The absence of a clear general signal in the differences prompted the clustering analysis explained in the following section.

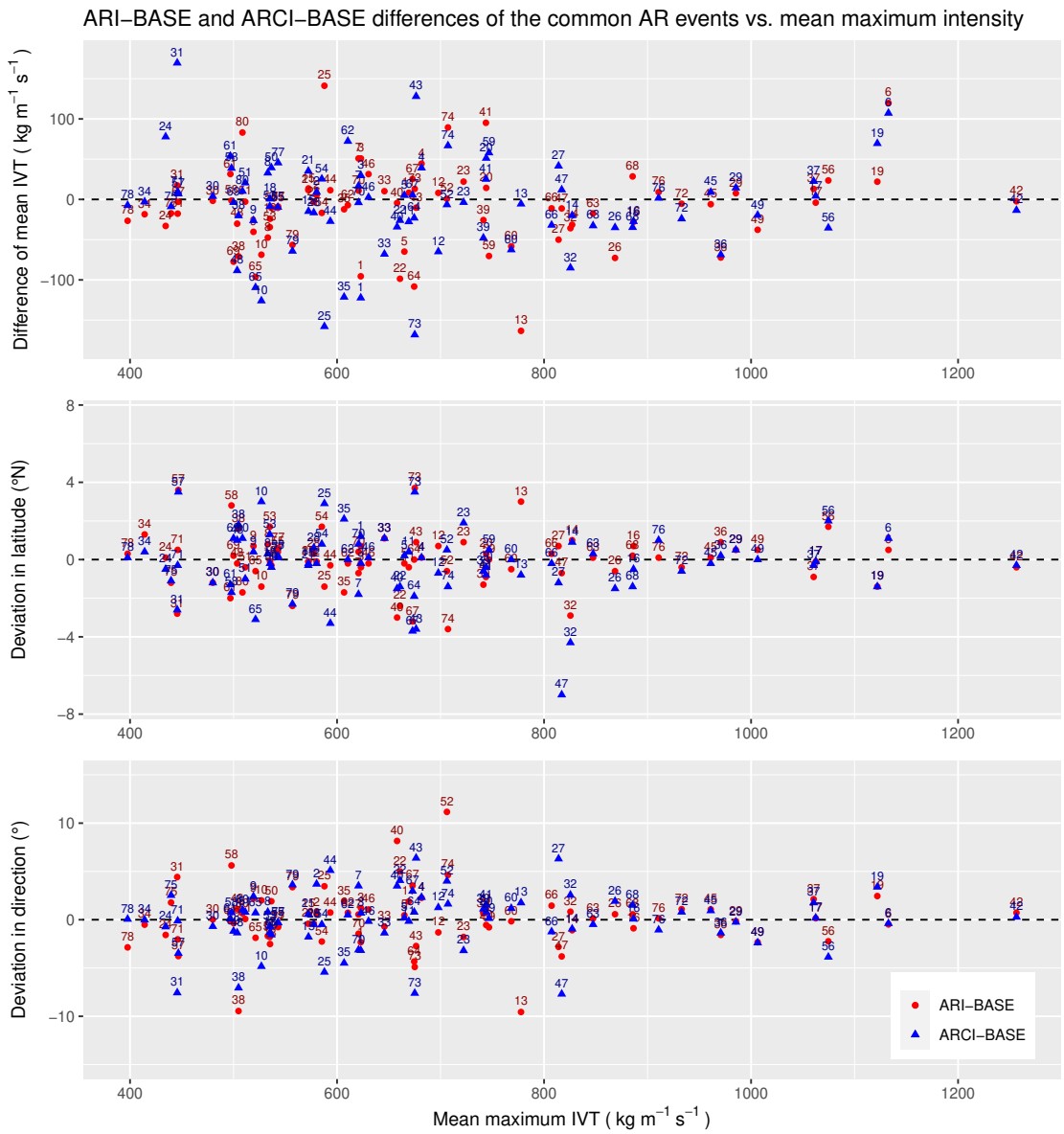

**Figure 6.** ARI-BASE (red) and ARCI-BASE (blue) differences in mean IVT direction, mean impact latitude and mean intensity (IVT magnitude) of the common AR events vs. their maximum intensity.

### 3.3.2 Spatial aerosol loading and ARs modification

To understand the influence of aerosols on the observed spatial deviations and IVT differences of the ARs, a classification and analysis of the common events have been carried out based on the spatial loading of the most relevant aerosols present in the study area, namely dust and sea salt. Initially, an EOF analysis (Principal Component Analysis) has been jointly performed for the sea salt and dust AOD (550 nm) standardized anomalies within the region bounded by -15° E and 4° E longitude and 33° N and 45° N latitude. The ARI and ARCI experiments used five and six retained EOFs to explain at least a 75 % of their total variance, respectively. The three leading EOFs of each experiment are portrayed in Figs. B1 and B2. A clustering classification following the Ward method (Ward, 1963) was then performed on these analyses, which separated the common cases into eight different groups in each experiment. The centroid of each cluster was associated with two centre fields, one per considered aerosol.

Figure 7 displays the two centres (sea salt and dust) of the eight ARI clusters, which were obtained as the mean fields of the events belonging to each group. The first clusters present a higher dust aerosol loading, while the latter ones exhibit a more significant loading of sea salt. Figure 8 is a box and whiskers plot that shows the ARI-BASE differences of mean IVT, mean incidence latitude and mean IVT direction of the common AR events belonging to each ARI cluster. The box length represents the interquartile range (IQR) of the data, thus the bottom (Q1) and top (Q3) edges of the box correspond to the $25^{th}$ and $75^{th}$ percentiles, respectively. The line inside the box is the median, or $50^{th}$ percentile. The whiskers extend to 1.5 times the IQR. The outliers, data points that fall outside the whiskers range, are marked with dots. The p-values of the differences are displayed in black for the clusters with at least 5 members. Focusing on the groups that have more than one event and present the most significant IVT differences, it was observed that an AR weakening (negative ARI-BASE IVT differences) occurs in clusters 2 and 3. Cluster 3, comprising only two AR events, precludes conducting a meaningful statistical significance analysis due to the insufficient sample size. However, cluster 3 could be interpreted as particularly intense dust events of the same nature as in cluster 2. The high concentration of mineral aerosols over the IP may be the reason for this weakening. In contrast, the sea salt aerosols have very low presence in both groups, and their effects are expected to be small and thus negligible.

ARs are commonly associated with a frontal surface, which can be identified by analyzing the thickness field. The thickness field of an atmospheric layer is directly and solely related to its mean virtual temperature given two fixed pressure levels, as depicted in the hypsometric equation (Stull, 2011). Therefore, the thickness field shows the maximum temperature gradient and thus the position of the front, which guides the AR. Moreover, stronger thickness gradients lead to more intense ARs. The mean thickness fields between 1,000 and 850 hPa of the events belonging to ARI clusters 2 and 3 are represented in Fig. 9 for ARI and BASE experiments. The same time steps are included in the representations of both experiments. Each thin arrow represents an AR event, located on its mean latitude and oriented accordingly to its mean direction. The length of the arrow is proportional to its mean IVT. The thickest arrow depicts the mean characteristics of all the ARs belonging to a cluster. As observed in cluster 2, the inclusion of aerosol-radiation interactions (direct effects) of dust aerosols in the ARI experiment results in a cooling of the atmospheric layer. This cooling acting on the warmer zones of the domain derives in weaker thickness gradients when compared to BASE, simulation in which radiation encountered a perfectly clean atmosphere (prescribed AOD

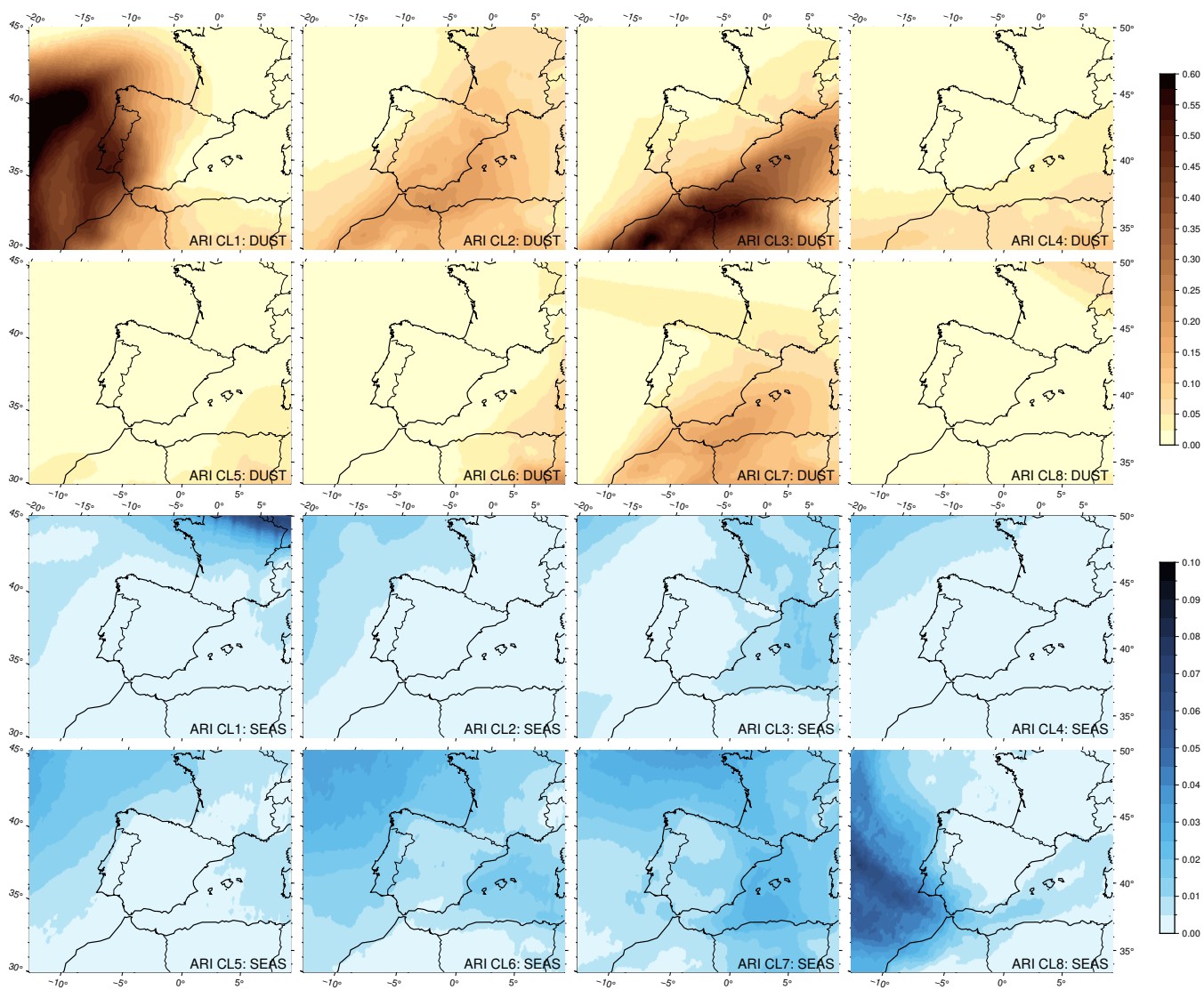

**Figure 7.** Centres of dust (DUST) and sea salt (SEAS) AOD at 550 nm of the different ARI clusters (CL).

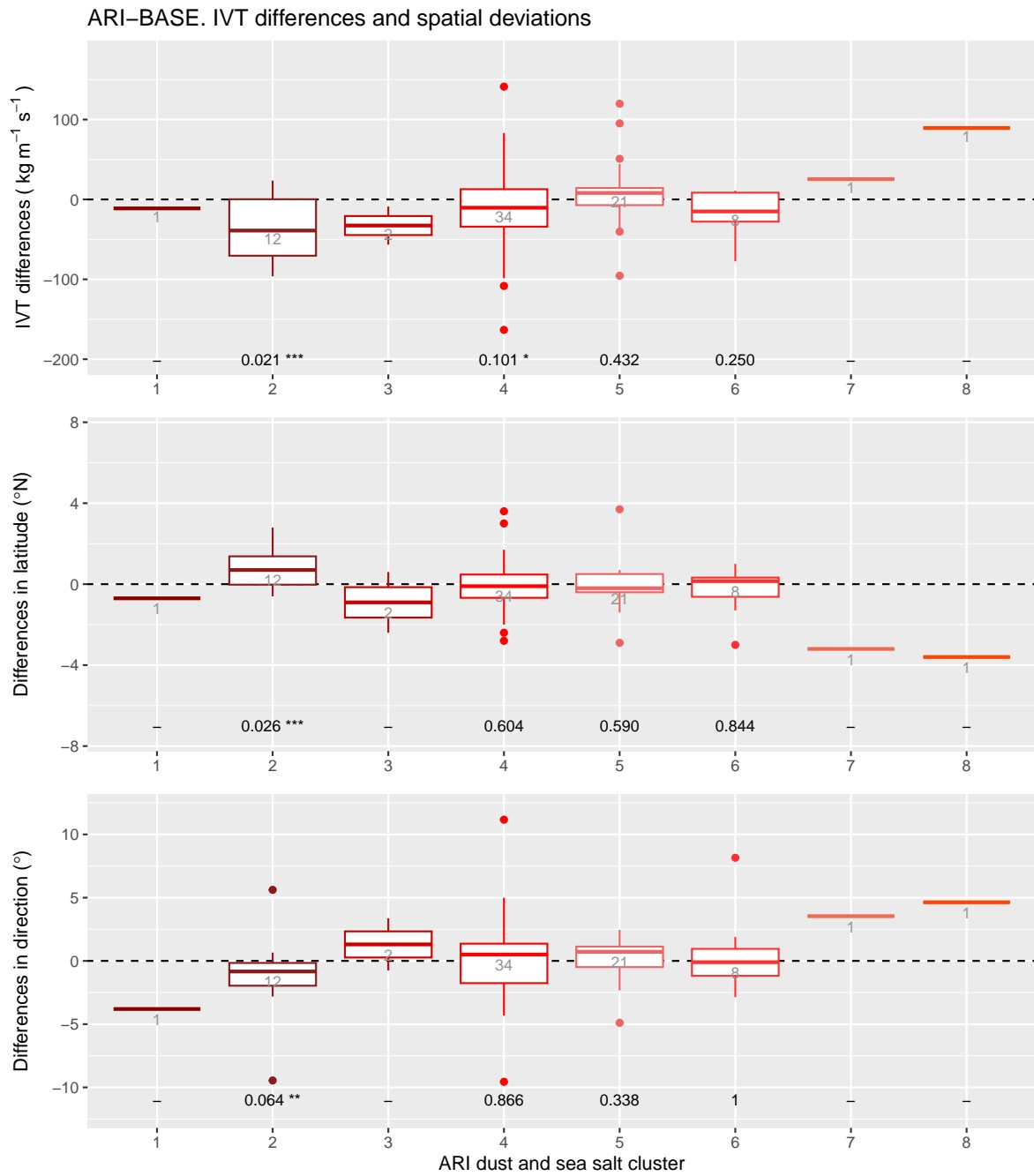

**Figure 8.** ARI-BASE differences of the mean IVT magnitude, mean incidence latitude and mean IVT direction of the common AR intervals grouped by the eight ARI sea salt and dust cluster groups. The number of events belonging to each cluster is indicated in grey. The p-values of the clusters with at least 5 members are included in black ("*": $p \leq 0.20$, "**": $p \leq 0.10$, "***": $p \leq 0.05$).

set to zero). In cluster 3, a wider cooling effect is present, but the more pronounced cooling in the south (over the north of Africa) leads to the observed weakening.

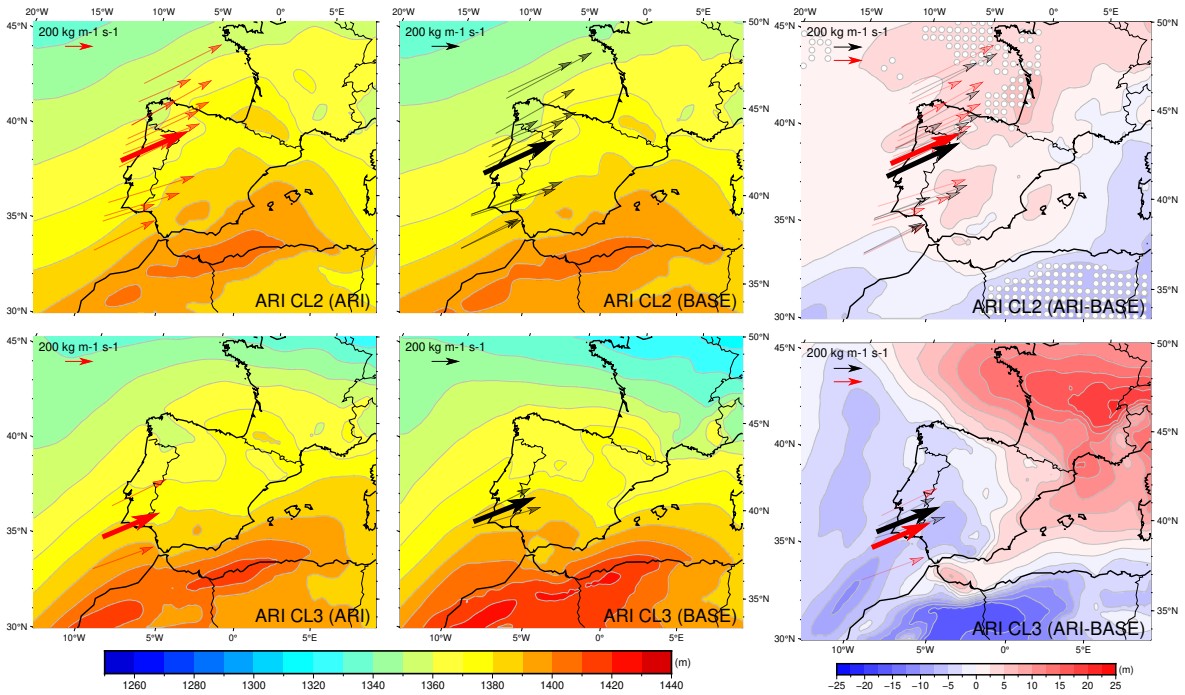

**Figure 9.** ARI and BASE mean thickness fields of the atmospheric layer between 1,000 and 850 hPa of the common AR events belonging to clusters 2 and 3 in the ARI simulation and ARI-BASE thickness differences. The same time steps are included in the representations of both experiments. Each thin arrow represents an AR event in (red) ARI or (black) BASE, located on its mean latitude and oriented accordingly to its mean direction. The length of the arrow is proportional to its mean IVT. The thickest arrow represents the mean characteristics of the cluster. White dots highlight statistically significant differences with a 90 % confidence level.

On the other hand, clusters 2 and 3 of the ARI experiment also exhibit some spatial deviations, as shown in Fig. 8. However, they exhibit opposite behaviors. The comparison of latitude and direction differences of the entire set of common events yields a significant negative correlation factor of -0.62, indicating that the aerosol configurations associated with northward (southward) deviations could also be linked to changes in the mean direction, resulting in more zonal (meridional) ARs relative to the BASE simulation.

An analogous analysis was employed to investigate the ARCI-BASE differences. The center fields of each ARCI dust and sea salt cluster are depicted in Fig. 10, which were computed as the mean fields. As before, the first clusters are characterized by a higher concentration of dust, whereas the last ones exhibit a higher presence of sea salt aerosols. The integration of the direct, semi-direct, and indirect effects of these aerosols causes a significant strengthening of the ARs associated with clusters 2 and 6, and a considerable weakening of the events belonging to clusters 7 and 8, as illustrated in Fig. 11. A meaningful

statistical analysis of cluster 8 is not viable with only three cases. However, it gathers the most intense sea salt events, whose effects can be explained as in cluster 7.

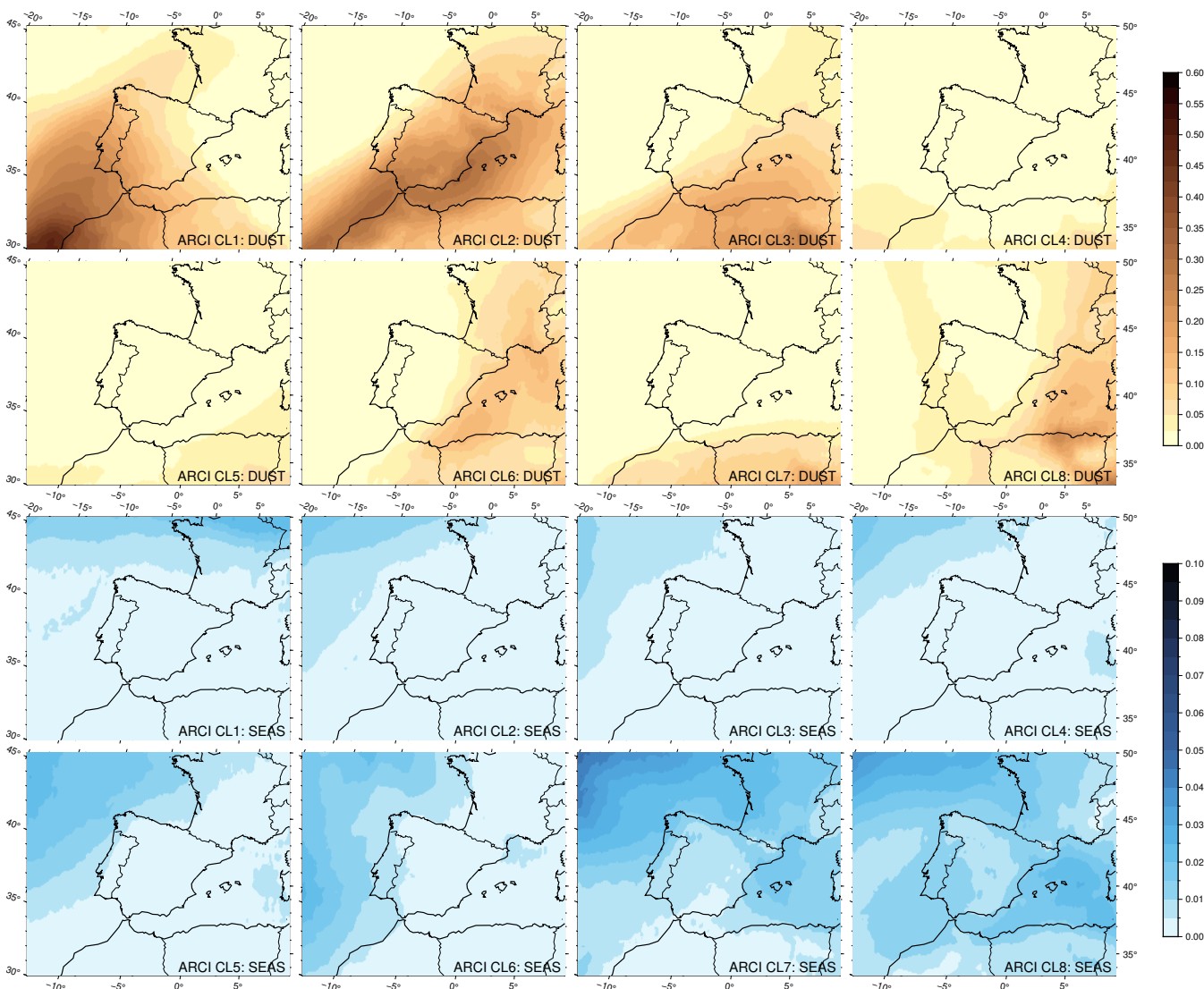

**Figure 10.** Centres of dust (DUST) and sea salt (SEAS) AOD at 550 nm of the different ARCI clusters (CL).

To understand the influence of aerosol interactions on the mean integrated vapor transport (IVT) magnitude of ARs, the thickness field of the clusters that exhibited the greatest ARCI-BASE differences was analyzed, as shown in Fig. 12. Specifically, the presence of dust aerosols was found to be associated with warming of the atmospheric layer compared to the BASE case, primarily due to indirect effects. High concentrations of dust aerosols generate a large number of small droplets that lead to increased cloud lifetime and warming of the atmospheric layer due to the release of latent heat. This effect is evident in

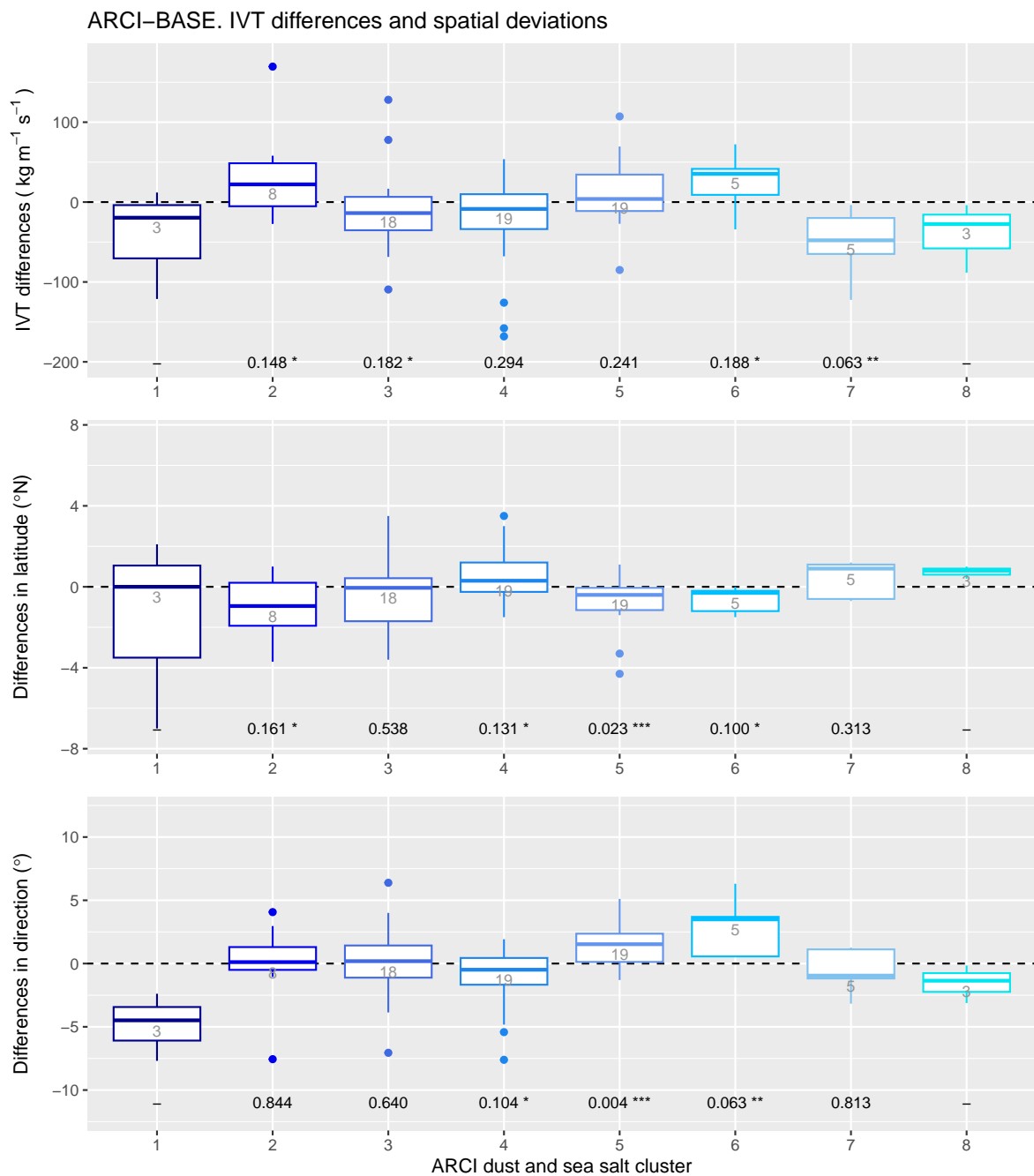

**Figure 11.** ARCI-BASE differences of the mean IVT magnitude, mean incidence latitude and mean IVT direction of the common AR intervals grouped by the eight ARCI sea salt and dust cluster groups. The number of events belonging to each cluster is indicated in grey. The p-values of the clusters with at least 5 members are included in black ("*": p ≤ 0.20, "**": p ≤ 0.10, "***": p ≤ 0.05).

cluster 2, where the positive differences align well with the distribution of dust. Furthermore, this warming of the warm zones leads to the strengthening of the thickness gradient, resulting in increased mean IVT magnitude of the ARs associated with this aerosol configuration.

Clusters 6, 7, and 8 are characterized by higher concentrations of sea salt aerosols, and their interactions with the atmosphere become relevant. Sea salt aerosols are highly hygroscopic and lead to a cooling effect on the atmospheric layer when interacting with it. This is due to enhanced rain droplet formation and early precipitation, which reduces the release of latent heat. When combined with the effects of dust aerosols, the ARCI-BASE thickness differences observed in Fig. 12 emerge. In cluster 6, the more significant cooling effect over the east of the IP than over the southeast results in a strengthening of the thickness gradient due to the higher concentration of sea salt aerosols. The particularly strong cooling observed over the north of the African continent may be attributed to the near-absence of sea salt and dust aerosols, which remarkably reduces the release of latent heat associated with droplet formation when compared to the BASE experiment (fixed concentration of CCN). On the other hand, cluster 7's common events, with a mean incidence latitude to the north, are primarily influenced by sea salt aerosols, which generate a wide but slight cooling in this configuration. Even subtle differences in the strength of this cooling may result in the observed weakening of the thickness gradient that guides the ARs. In a similar vein, weakened thickness gradients and ARs result in cluster 8, due to warming of the cold northern zones and slight cooling of the warm zones of the southeast of the IP.

Furthermore, similar to the ARI-BASE analysis, the comparison between latitude and direction differences of the entire set of common events in the ARCI-BASE experiment (Fig. 11) resulted in a significant negative correlation factor of -0.43.

### 3.3.3 Case studies: 27 October 2005 and 12 January 1998

The application of clustering analysis and mean fields has provided valuable insights into the mechanisms underlying the perturbations of ARs by aerosols. Nonetheless, two case studies that compare the BASE, ARI and ARCI simulations can offer further insights into the role of aerosols in modifying ARs, while avoiding any potential smoothing effect of the mean fields. Specifically, the common AR events of 27 October 2005 (belonging to cluster 2 in both ARI and ARCI experiments) and 12 January 1998 (belonging to cluster 2 in the ARI experiment and cluster 6 in the ARCI experiment) were selected for analysis in these two case studies.

In the first case (Figs. 13 and 14), ARI and ARCI ARs present a mean intensity difference of -70.32 and 58.01 kg m$^{-1}$ s$^{-1}$ with respect to the BASE AR, respectively. The IVT differences observed in the ARI simulation may be attributed to a cooling of the southwestern region of the study domain caused by dust aerosols, resulting in a weakening of the thickness gradient. On the other hand, the opposite occurs in the ARCI simulation, with a strong heating of the warm eastern zones and a slight cooling of the cold northern zones with respect to the BASE, resulting in a strengthening of the thickness gradient of the front that drives the AR. In the ARCI experiment, the indirect effects of dust particles in high concentration result in a larger number of small droplets, and the increased cloud lifetime leads to a greater release of latent heat, thereby causing a heating of the atmospheric layer. Moreover, the ARCI AR exhibits a more zonal direction (and a slight northward displacement, as expected from the results of the previous subsection), with the maximum temperature gradient and the frontal surface that guides the

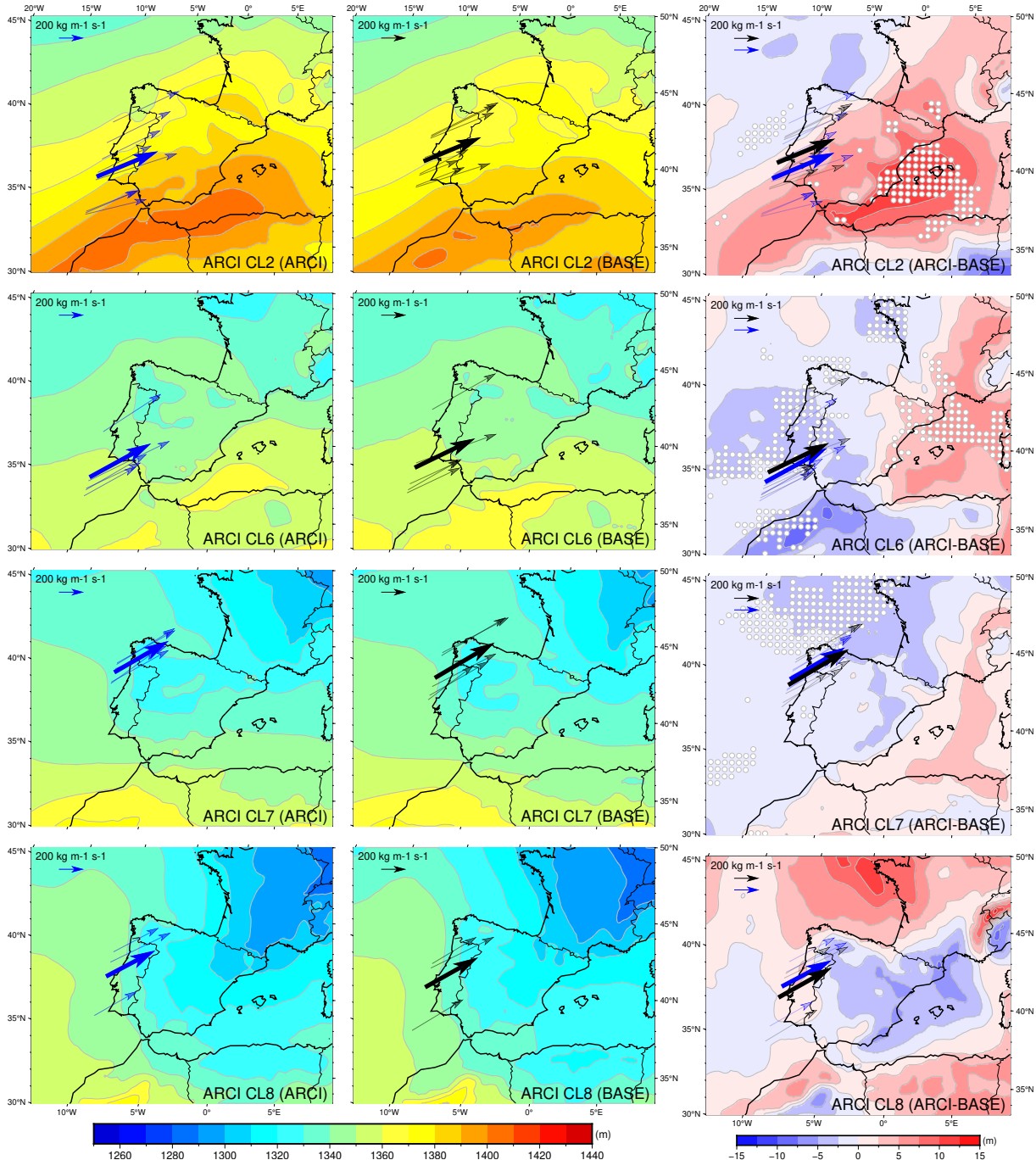

**Figure 12.** ARCI and BASE mean thickness field of the atmospheric layer between 1,000 and 850 hPa of the common AR events belonging to clusters 2, 6, 7 and 8 in the ARCI simulation and ARCI-BASE thickness differences. The same time steps are included in the representations of both experiments. Each thin arrow represents an AR event in (blue) ARCI or (black) BASE, located on its mean latitude with its mean direction. The length of the arrow is proportional to its mean IVT. The thickest arrow represents the mean characteristics of the cluster. White dots highlight statistically significant differences with a 90 % confidence level.

AR coinciding with the area of the highest dust AOD gradient. These temperature variations and circulation changes give rise to the observed IVT differences and the deviation of the detected AR. Furthermore, Fig. 15 displays the total accumulated precipitation distribution of this event. BASE and ARI present a similar magnitude, while the ARCI experiment exhibits a notably higher amount of precipitation on the west coast of the IP. The recorded accumulated rainfall can be found in Fig. D1 (left).

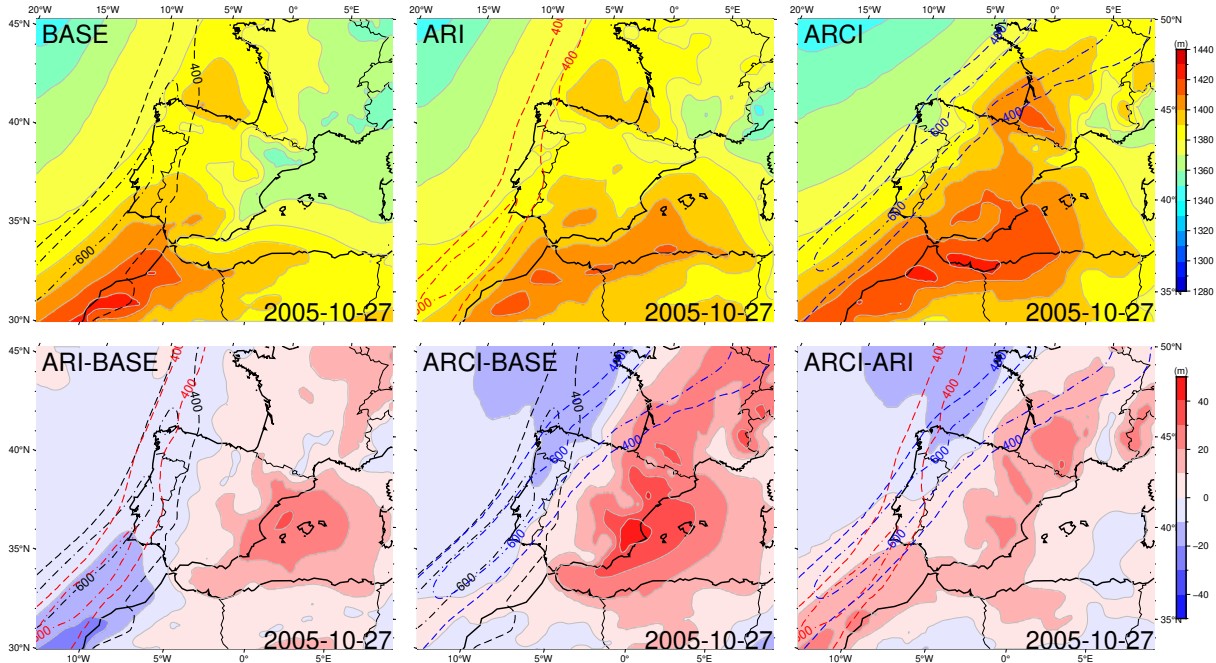

**Figure 13.** 22:00 h, 27 October 2005. Thickness field between 1000 and 850 hPa of the three simulations (top) and thickness differences (bottom). Black, red and blue contours show BASE, ARI and ARCI ARs respectively (400 and 600 kg m$^{-1}$ s$^{-1}$ IVT levels).

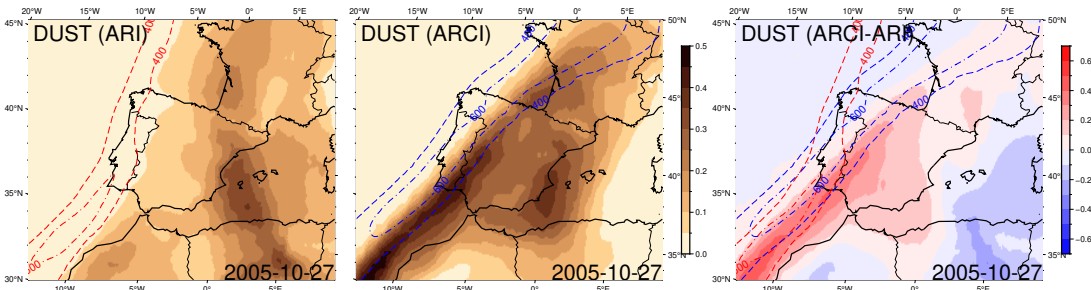

**Figure 14.** 22:00 h, 27 October 2005. ARI and ARCI dust AOD at 550 nm and ARCI-ARI AOD differences. Red and blue contours show ARI and ARCI ARs respectively (400 and 600 kg m$^{-1}$ s$^{-1}$ IVT levels).

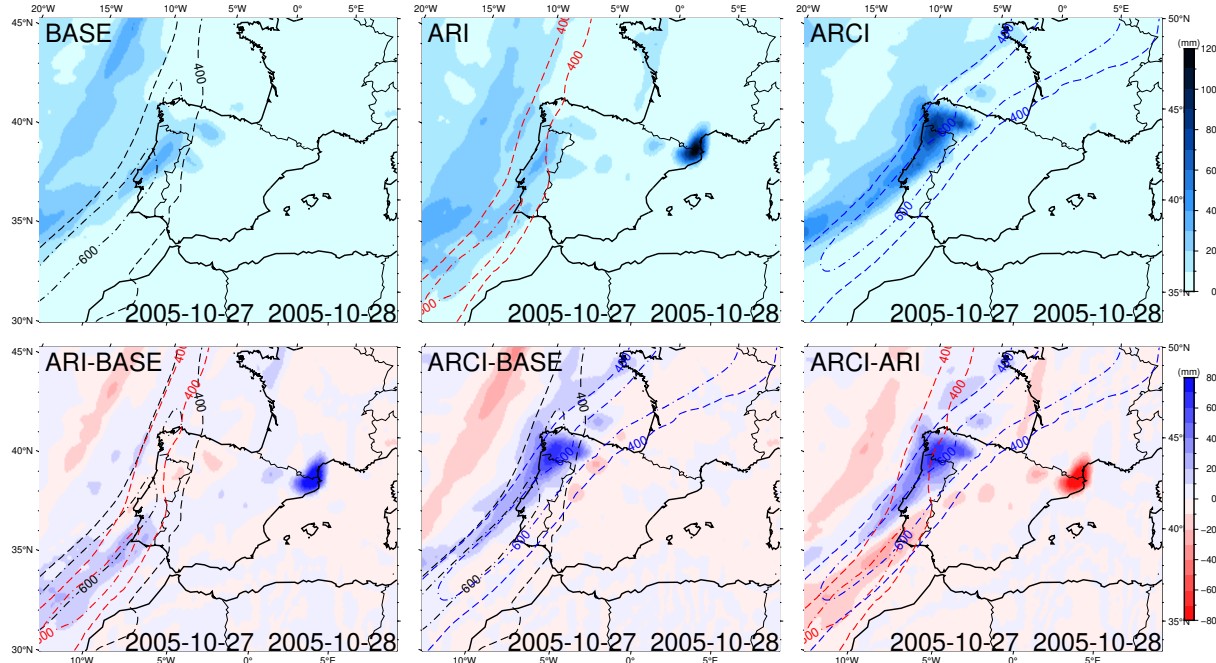

**Figure 15.** Common AR event of the 27 October 2005. Total accumulated precipitation during the entire event (2 days) in the three simulations (top) and precipitation differences (bottom). Black, red and blue contours represent BASE, ARI and ARCI ARs at 22:00 h on October 27, respectively (400 and 600 kg m$^{-1}$ s$^{-1}$ IVT levels).

Regarding the event on 12 January 1998 (Figs. 16 and 17), both ARI and ARCI ARs show a mean intensity difference of -50.04 and 41.54 kg m$^{-1}$ s$^{-1}$ with respect to the BASE AR, respectively. The cooling effect of dust aerosols in the ARI simulation could be responsible for the weakening of the temperature gradient and the observed negative IVT differences. In contrast, the indirect effects of dust aerosols in the ARCI experiment result in a heating effect, and a further southward latitude deviation of the AR trajectory. Although the ARCI AR shows a more zonal direction during the represented time-step, the whole interval has a mean direction 6.3° higher than in the BASE experiment (with an increased relative meridional component). The southward displacement of the sea salt distribution in the ARCI experiment coincides with the deviation of the AR trajectory. As a result of this shift, the ARCI simulation displays the highest values of accumulated precipitation over land (Fig. 18), which aligns with the observed data for this event (Fig. D1 (right)).

In BASE and ARI experiments, the same types and concentrations of CCN are prescribed for both marine and continental areas. However, marine aerosols, primarily sea salt, are more hygroscopic and occur in lower concentrations compared to continental aerosols. As a result, larger and more rapidly precipitating droplets are formed, releasing less latent heat in the ARCI simulation, where aerosol effects on condensation are considered interactively. This decrease in the released latent heat leads to cooling of the cold zones relative to the BASE experiment, moving the maximum temperature gradient further south,

and coinciding with the southern border of the sea salt distribution. The combination of sea salt and dust aerosol indirect effects
gives rise to a strengthening of the ARCI AR.

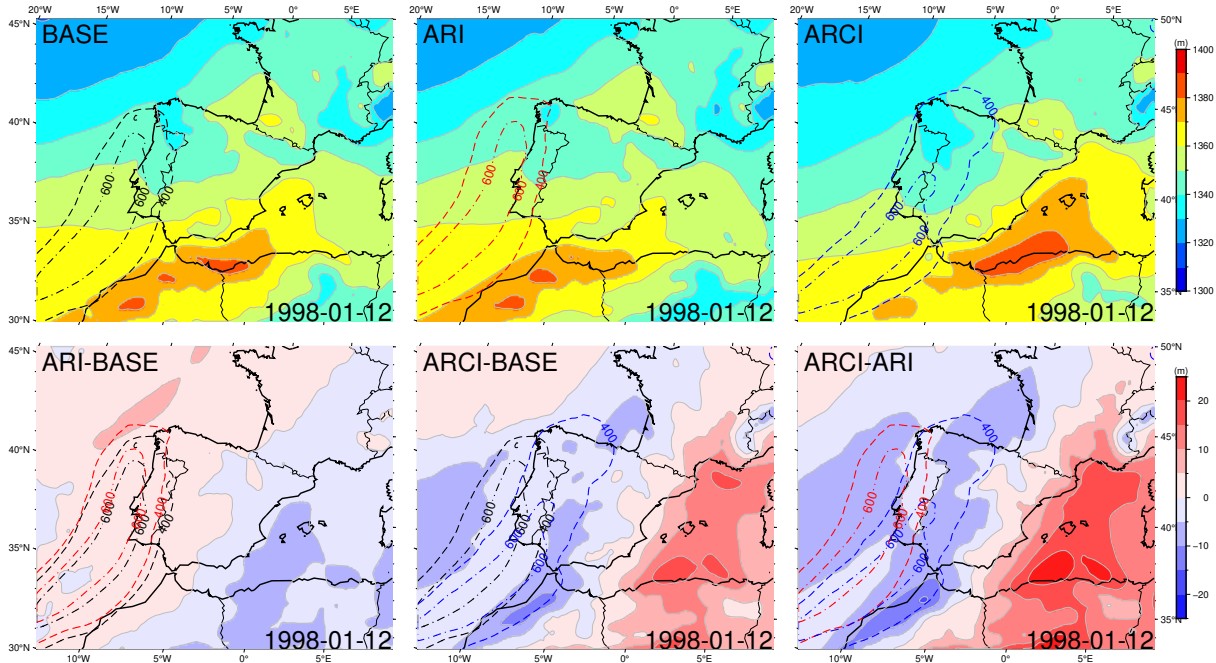

**Figure 16.** 09:00 h, 12 January 1998. Thickness field between 1000 and 850 hPa of the three simulations (top) and thickness differences (bottom). Black, red and blue contours show BASE, ARI and ARCI ARs respectively (400 and 600 kg m$^{-1}$ s$^{-1}$ IVT levels).

## 4 Conclusions

ARs, which are associated with numerous extreme precipitation events in regions influenced by maritime flows, are of critical importance in meteorological predictions and climate change projections, both globally and regionally. The primary objective of this research was to investigate the sensitivity of ARs to aerosol treatment in regional climate simulations for the IP. To achieve this objective, ARs in three experiments that covered Europe during the period of 1991-2010 were analyzed. In the BASE experiment, aerosols were prescribed without considering their interactions with radiation, and the number of CCN was fixed. In contrast, in the ARI and ARCI experiments, the model resolved aerosols dynamically. The ARI experiment considered only the direct and semi-direct effects of aerosols, while the ARCI experiment included indirect effects as well.

A number of AR identification algorithms are available. However, many of them may not be suited for use with regional land-focused models, whose spatial limits are very close to the detection area, and thus preclude capturing the AR structure over the ocean. To address this issue, a novel regional scale AR identification algorithm, called AIRA, has been developed based on IVT, which comprises two stages: 1) preprocessing and 2) filtering/identification. Initially, an IVT threshold is set, and the time intervals where the maximum IVT magnitude exceeds this threshold are identified. Subsequently, each time interval is assessed

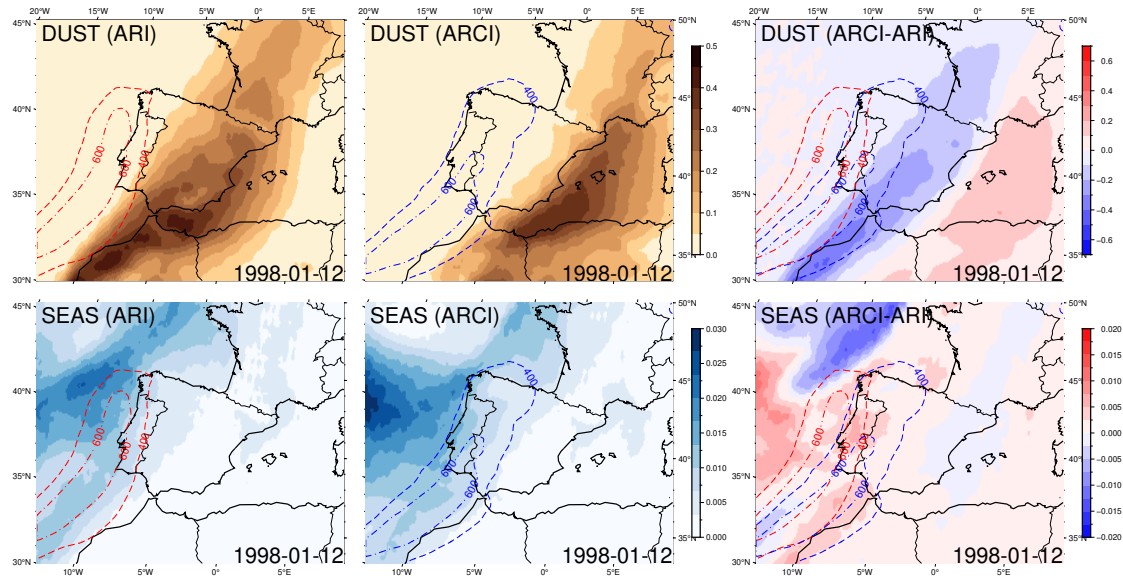

**Figure 17.** 09:00 h, 12 January 1998. ARI and ARCI dust (top) and sea salt (bottom) AOD at 550 nm and ARCI-ARI AOD differences. Red and blue contours show ARI and ARCI ARs respectively (400 and 600 kg m$^{-1}$ s$^{-1}$ IVT levels).

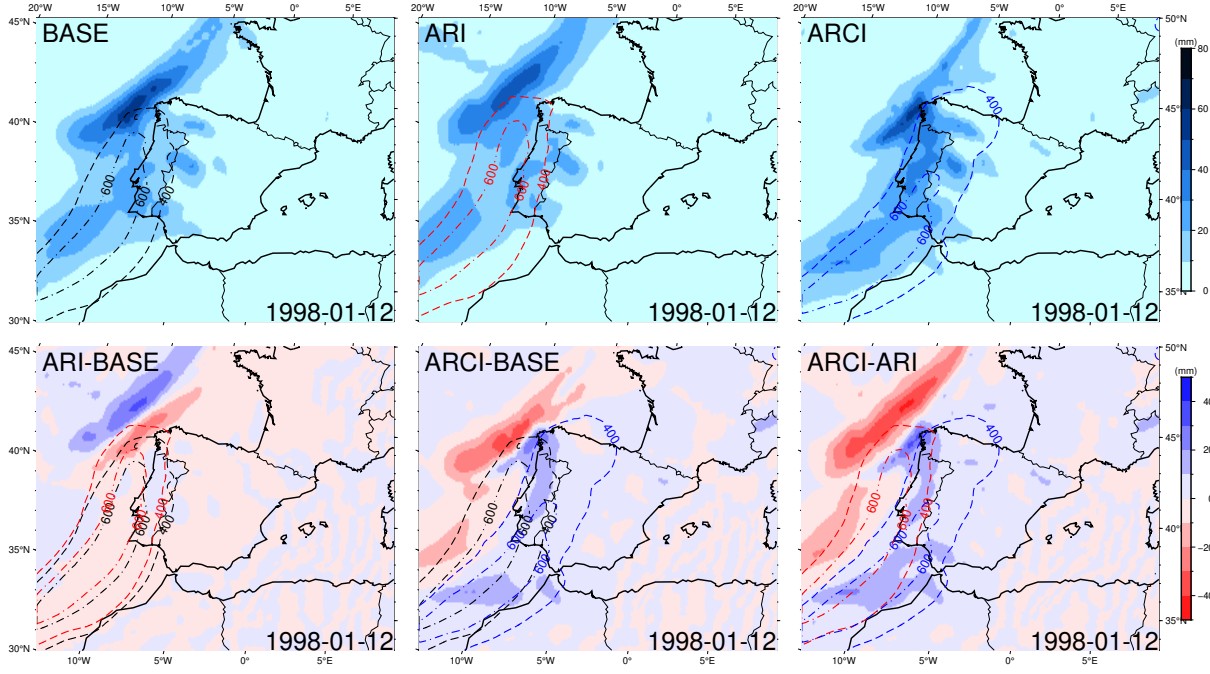

**Figure 18.** Common AR event of the 12 January 1998. Total accumulated precipitation during the entire event (1 day) in the three simulations (top) and precipitation differences (bottom). Black, red and blue contours represent BASE, ARI and ARCI ARs at 09:00 h, respectively (400 and 600 kg m$^{-1}$ s$^{-1}$ IVT levels).

to determine if it meets the necessary geometry conditions (length and width) and direction requirements to be identified as an AR. AIRA's performance was evaluated by comparing the results with a global algorithm archive of identified ARs.

AIRA successfully identified approximately 250 ARs in each experiment spanning the period from 1991-2010. Notably, the most frequent, intense, and long-lasting ARs occurred during spring and autumn, while they were less frequent, shorter, and weaker in the summer season. The direction of the axis of the identified ARs was observed to be distributed around $40°$, indicating their transport from the tropical regions of the Atlantic Ocean. Furthermore, it was found that ARs were responsible for up to 30 % of the total accumulated precipitation, underscoring the extreme nature of the precipitation associated with these events. Interestingly, the inclusion of aerosols, particularly their indirect effects, led to a redistribution of precipitation, with notable increases in the areas of occurrence.

Although the number of identified ARs is comparable among the three simulations, their common AR events were only observed in 37 % of the time steps with AR. This suggests that the inclusion of aerosols at different levels in the simulations may play a crucial role in determining their behavior. Comparison of the ARI-BASE and ARCI-BASE differences revealed that the deviations were minimal in the most intense cases, while the largest differences were observed in weaker events.

The joint analysis and classification of dust and sea salt aerosol distributions enabled the identification of eight main aerosol configurations in both ARI and ARCI simulations. In the ARI experiment, the aerosol-radiation interactions of dust resulted in a cooling effect. In contrast, dust was associated with a warming effect in the ARCI simulation, where the aerosol-cloud interactions of sea salt led to a cooling effect with respect to the BASE experiment. The combined action of dust and sea salt aerosols, through their direct and/or indirect effects, strengthened (weakened) the frontal surfaces guiding ARs by producing a cooling (warming) of the cold zones and/or a warming (cooling) of the warm areas. The physical mechanisms underlying the modification of the dynamical conditions driving ARs were related to differences in the temperature gradient, which led to significant changes in the thickness field. These differences were associated with aerosol-radiation-cloud interactions, including direct, semi-direct and indirect effects, which were found to be relevant. Furthermore, a negative correlation was observed between mean direction and mean latitude differences. Specifically, deviations towards the north (south) were associated with more zonal (meridional) ARs with respect to the reference simulation.

In summary, this study has highlighted the impact of varying aerosol treatments on the simulation of ARs in regional climate models. Future research could examine the sensitivity of the outcomes to alterations in the different parameters employed by AIRA. Moreover, it would be worthwhile to explore whether incorporating more complex and computationally-intensive processes in the models would lead to notable enhancements in the representation of real AR events.

*Code and data availability.* The code developed to build AIRA is fully available as an open-access resource (https://doi.org/10.5281/zenodo.7885383, Raluy-López et al. (2023b)) on the Zenodo archive. The final product consists on a preprocessing bash script meant to be followed by the main part of the algorithm implemented with R functions. Figures have been prepared with R and Generic Mapping Tools (GMT) software. All the WRF-Chem simulations presented in this paper have been carried out in the MAR group of the University of Murcia as a product

of the REPAIR project. The simulation output data used to generate figures presented throughout this paper are available as an open-access resource (https://doi.org/10.5281/zenodo.7898400, Raluy-López et al. (2023a)) on the Zenodo database.

*Author contributions.* ERL wrote the algorithm code, and performed the calculations in this paper. JPM contributed to the design of the simulations and their analysis. He also provided ideas for new approaches in the analysis of the simulations that have been integrated into the
490 final paper. PJG and JPM provided substantial expertise for a deep understanding of the AR concept, which led to a successful conception of the algorithm. The manuscript writing has been led by ERL, with all authors contributed to reviewing the text.

*Competing interests.* The authors declare no conflict of interest.

*Acknowledgements.* The authors acknowledge the ECCE project (PID2020-115693RB-I00) of the Ministerio de Ciencia e Innovación/A-gencia Estatal de Investigación (MCIN/AEI/10.13039/501100011033). ERL thanks her predoctoral contract FPU to the Ministerio de Uni-
495 versidades of Spain.

**Appendix A: Trigonometric elements involved in AIRA**

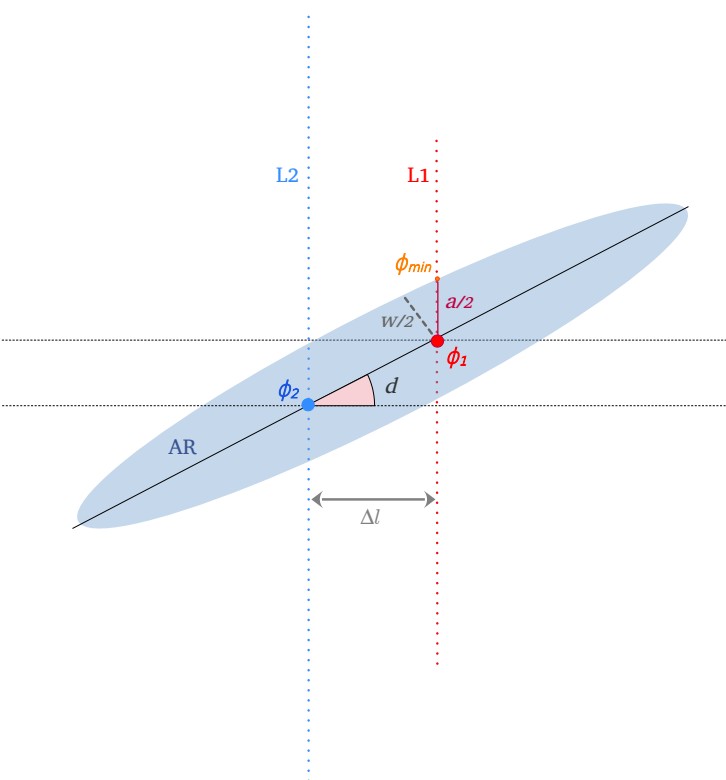

**Figure A1.** Sketch of the trigonometric elements used to derive the relevant parameters in AIRA. The blue shade represents an AR. L1 and L2 are the identification lines. $\phi_1$ and $\phi_2$ denote the latitudes of the maximum IVT registered on L1 and L2, respectively. $\phi_{min}$ corresponds to the latitude of the farthest point from the AR spine whose IVT value over L1 still exceeds the IVT threshold. $\Delta l$ is the distance between the identification lines expressed in degrees of longitude. These parameters are used to calculate the AR direction ($d$), the AR width over L1 ($a$) and the AR width ($w$).

## Appendix B: Dust and sea salt leading EOFs

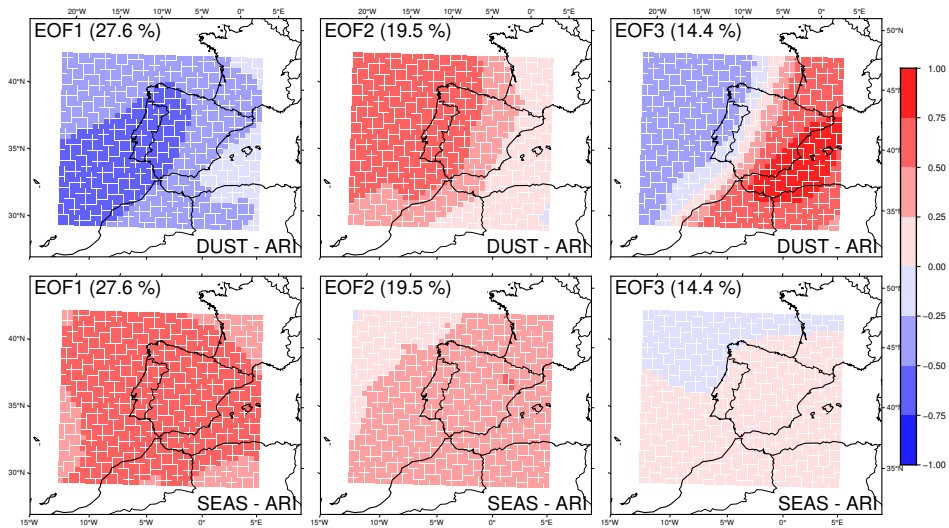

**Figure B1.** Three leading EOFs of the joint analysis of dust and sea salt aerosols of the 80 common AR events in the ARI simulation. Each EOF is associated with two fields, one per considered aerosol: (top) dust and (bottom) sea salt. The percentage of variance explained by each EOF is shown in brackets.

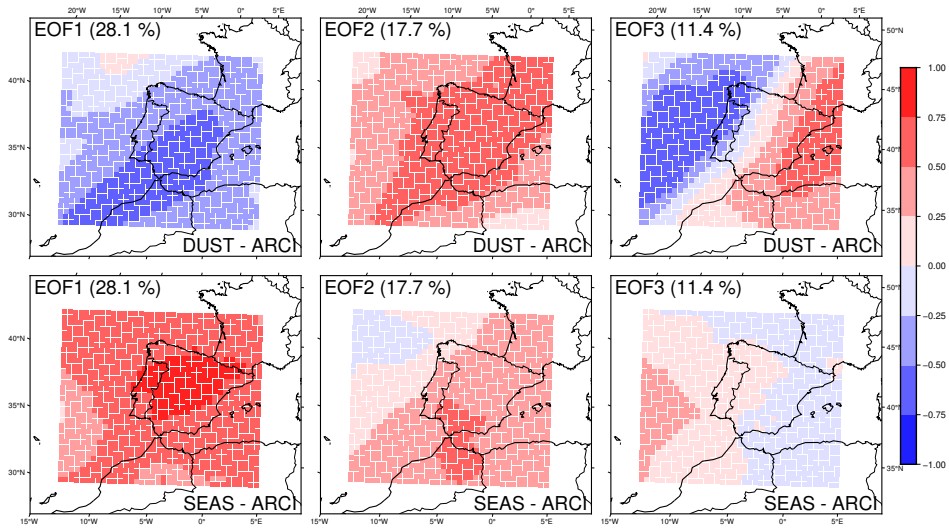

**Figure B2.** Three leading EOFs of the joint analysis of dust and sea salt aerosols of the 80 common AR events in the ARCI simulation. Each EOF is associated with two fields, one per considered aerosol: (top) dust and (bottom) sea salt. The percentage of variance explained by each EOF is shown in brackets.

## Appendix C: Correlation between latitude and direction differences

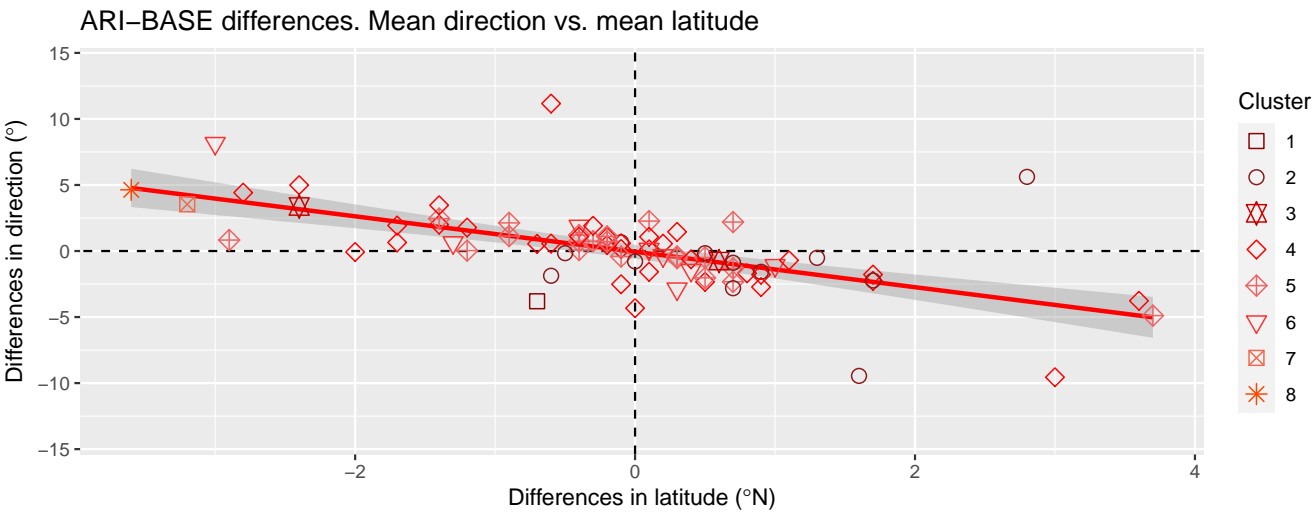

**Figure C1.** ARI-BASE differences in mean direction versus ARI-BASE differences in mean incidence latitude. Each set of shape and colour represents an ARI cluster. The linear regression model is plotted with a red line and the grey area represents the 95 % confidence level interval.

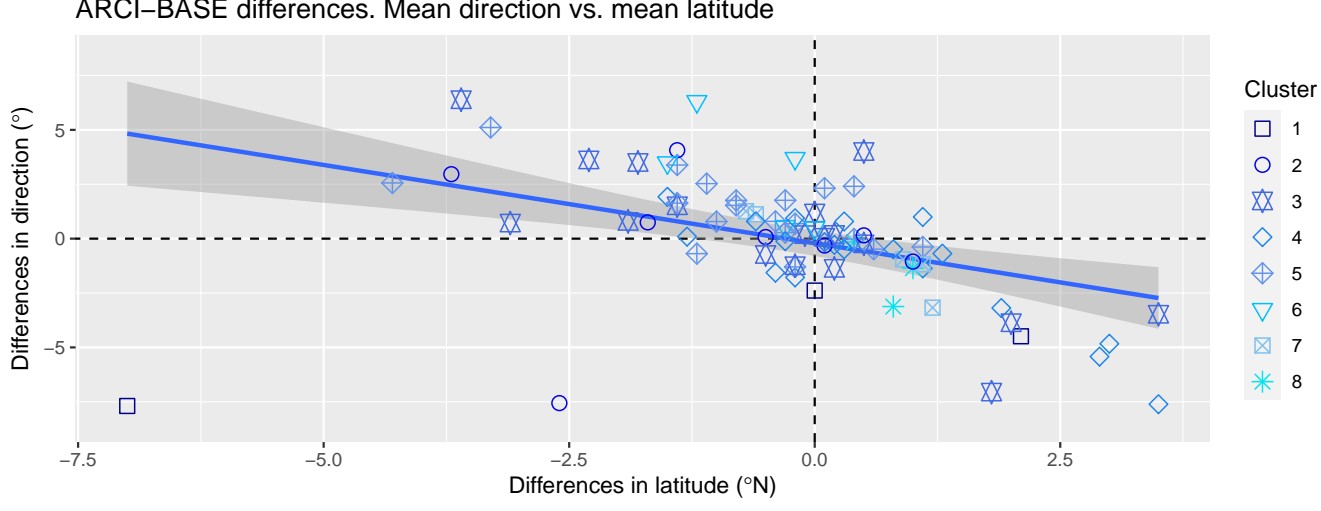

**Figure C2.** ARCI-BASE differences in mean direction versus ARCI-BASE differences in mean incidence latitude. Each set of shape and colour represents an ARCI cluster. The linear regression model is plotted with a blue line and the grey area represents the 95 % confidence level interval.

## Appendix D: Case studies - Observed accumulated precipitation

Observed accumulated precipitation during the two case studies of Section 3.3.3. The employed data was derived from the
Iberia01 daily gridded (0.1° resolution) dataset of precipitation over the Iberian Peninsula (Gutiérrez et al., 2019).

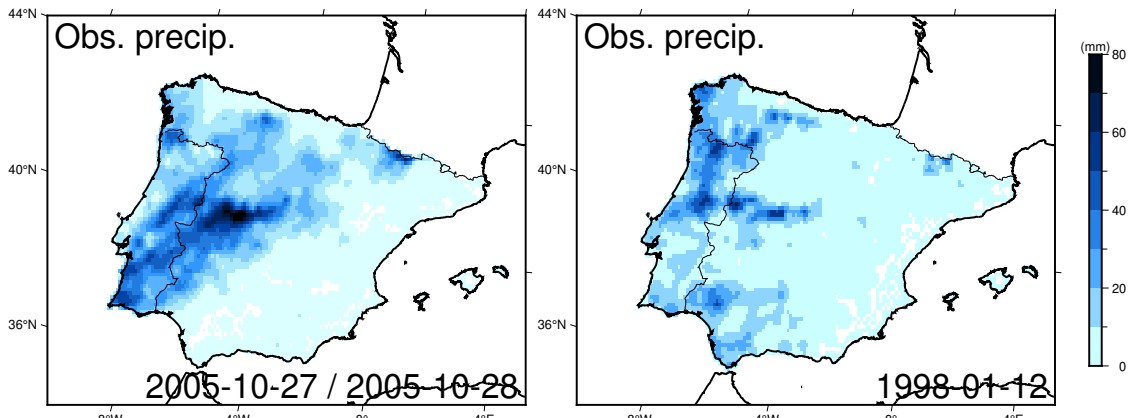

**Figure D1.** Observed accumulated precipitation during the case studies of (left) 27-28 October 2005 and (right) 12 January 1998. Precipitation data derived from Gutiérrez et al. (2019).

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
