# Peer review of "Sensitivity of atmospheric rivers to aerosol treatment in regional climate simulations: Insights from the AIRA identification algorithm"

_EGUsphere, 2023_

## Referee Comment (RC3)

Review of the paper: **Sensitivity of atmospheric rivers to aerosol treatment in regional climate simulations: Insights from the AIRA identification algorithm** by Eloisa Raluy-López, Juan Pedro Montávez, and Pedro Jiménez-Guerrero

This investigation aims to develop a robust approach to identify ARs (called AIRA algorithm) under the constraint of the regional domain provided by the area-limited simulations. By developing this approach, the authors are able to inspect the impact of online aerosols on this meteorological phenomenon. The key finding of this work is how aerosols are treated in simulations may influence both in ARs intensity and trajectory by radiative changes cooling or heating the air.

**General comments**

The paper is well structured, and the algorithm adopted is easy to understand thanks to the illustrative figures. Although several approaches have been developed to identify ARs, their innovation relies on overcoming the RCMs' limitations where most of the runs are focused over land, and this precludes capturing the long way over the ocean. The success of this approach will allow the use of RCMs to provide more accurate precipitation amounts than GCMs and to perform less computationally costly simulations such as online aerosol runs to understand ARs mechanisms. Then, I found this work a valuable advance to analyze the impacts of the AR's landfalling.

Under these arguments, I recommend accepting this work after addressing a minor revision detailed below.

1. Introduction. In line 55, the authors mention the lack of research about the impact of aerosols on ARs but they did not discuss the challenges nor mention previous works such as any Counterbalancing influences of aerosols and greenhouse gases on atmospheric rivers by Baek and Lora

2. Methods.

   How can the AIRA be sure that is detecting an AR and not the branch of a low system with a bigger enough radius? Does $\Delta\Theta < 25$ guarantees this fact? Maybe introducing SLP values will avoid this concern.

   In Table 1 the authors show the imposed parameters. To demonstrate the robustness of the approach some discussions about the sensitivity of these parameters are needed. For instance, how many percentages of ARs increase/decrease if the IVT threshold is modified?

To better contextualize your methodology, I missed a discussion comparing AIRA approach with other methodologies of other tracking approaches, For instance, a review can be found in: *Atmospheric River Tracking Method Intercomparison Project (ARTMIP): project goals and experimental design by Ruth et al.*

3. Results

Following the previous comment, some validation against observations (e.g. satellite images) and/or using the ARs inventory/catalogs is needed to be the coherence of your approach with the ARs already identified along the bibliography.

In line 196 the authors mention. "It was found that most of the ARs identified by AIRA also matched those identified by global-scale algorithms, as reported by Brands et al. (2017)." How many coincidences did you find? Did you find more 'real' ARs in BASE or in ARCI? Do you think that some discrepancies may be due to a different approach or the use of an RCM instead of a GCM?

In Line 224 the authors assert that the ARs explain the 30% of the precipitation, it is not clear what area did you use to obtain this value, and the Fig. 5 shows strong spatial variability to perform a spatial average. Furthermore, how accurate is the precipitation during these events? Is ARCI or BASE more representative of the observed precipitation?

In Line 232. Only 37 % of the coincidence of ARs between ARCI and BASE looks like a few percent. When the simulations are described there isn't any mention of nudging or re-initialization of initial conditions has been mentioned. What percentage of these discrepancies could be due to different treatments of aerosols or due to internal variability of the simulations?

When sea salt and dust clusters are analyzed (Fig. 7 and 10) It will be interesting to see mean ARs trajectories for each cluster (for instance superimposed with dotted lines).

In the analysis of the differences to better understand the thermodynamics and dynamics changes, it will be illustrative to analyze whether the IVT changes are more due to IWV or/and winds.

Throughout the work, I missed more analysis about the impacts of ARs on precipitation. I understand that may be the scope of future work. For the case studies will be interesting

to show the spatial distributions of the precipitation (accumulated during the whole event and/or hourly) for the three simulations; BASE, ARI, and ARCI. These will provide some insights about how the intensity and trajectory of ARs impact on the precipitation distributions. Furthermore, the authors found around 30% of ARs impact precipitation but this percentage will have spatial and temporal variability. For instance, as ARs have an interannual variability also their impact on precipitation will be significant. Finally, it will be interesting a further understand the low impact on precipitation of the ARs over Galicia, Is it less frequency of ARs, more precipitation due to cold fronts, or orographic arguments?

---

## Author Response (AR1)

**Authors' Response**

We would like to once again express our gratitude to the three referees for their valuable suggestions and insightful comments. Their thoughtful and thorough review of our work have greatly contributed to the substantial improvement of the manuscript. Referee comments are included in **Black**. Our detailed responses, previously posted during the open discussion, are highlighted in **Red**. The changes implemented in the revised manuscript are indicated in **Blue**.

**Response to Referee #1**

**General comments:**

This study is not the first that aim at detecting ARs in regional models. This should be mentioned. Some remarks are given in the special comments.

Thank you very much for your comment. It is something that was missing and the other referees also noted this issue. In the revised manuscript, we are going to put AIRA in the ARTMIP context and classification, including its main differences with the IDL ARDT (Ramos et al., 2016) and Brands ARDT (Brands et al., 2017) algorithms, which are the most similar to AIRA and also detect ARs over the Iberian Peninsula. As a preliminary observation, the main contrast is that both algorithms make use of spatial tracking, while AIRA never uses it, as it is intended to perform also in regions close to the domain edges. This is indeed the case in our study, with the detection lines located very near the limits of the spatial domain.

As stated in the answers to similar questions of Referees #2 and #3, we have included the following paragraph at the end of Section 2.2.2, following the valuable suggestions of all the referees:

"In the ARTMIP context (Shields et al., 2018), AIRA would be classified as a condition ARDT that imposes an absolute IVT threshold to determine if an AR could be present on the detection lines at a given time slice over the IP region. Then, all the consecutive potential AR time slices are gathered into potential AR intervals with a minimum time stitching T and the geometry requirements (width, direction) are imposed to the mean values of the intervals. Throughout this process, the trigonometric elements employed are derived from just two close points: the locations of maximum IVT over L1 and L2. No spatial tracking is required. This is the main difference between AIRA and other ARDTs that look at ARs over the IP or west Europe. For instance, the IDL ARDT (Ramos et al., 2016a) uses an IVT threshold (relative instead of absolute) to identify the arrival of a potential AR to a detection line. However, once the threshold is exceeded, this ARDT performs an east-west analysis to spatially determine the AR spine and impose a minimum length. A similar approach was employed by Lavers et al. (2012) and Brands et al. (2017). The innovation of the AIRA approach relies on overcoming the RCMs limitations where most of the runs are focused over land, and this may preclude capturing the long AR

structure over the ocean. AIRA was designed to work even in regions located very close to the domain borders, as it only employs data over two line grids."

The description of the algorithm should be improved and some deviation from existing ones should be explained.

As said right above, its deviation from the existing ones will be addressed in the revised manuscript.

The description of the algorithm has been improved thanks to the Specific Comments (see that section for specific changes) and thanks to the comments of the other referees. The paragraph shown above includes the comparison between AIRA and other regional ARDTs that look at ARs on the IP.

Unlike others, the AIRA algorithm detects ARs on two longitudes at 10 and 12°E and infers additional length and direction criteria by employing trigonometric functions. Though this is described briefly in the text, a figure sketch with a zoom on L1 L2 to draw the trigonometric elements used to derive the relevant parameters would be helpful, e.g. the direction and length scales.

Thank you for the suggestion. A visual representation of the trigonometric elements used in AIRA will make the algorithm explanation easier to follow. We are going to include it as a new figure in the revised manuscript.

The following Figure A1 has been included in the manuscript as an appendix and its mention has been added to the text:

"A diagram displaying the trigonometric elements utilized to calculate the aforementioned parameters is available in Fig. A1."

[Figure]

Figure A1. Sketch of the trigonometric elements used to derive the relevant parameters in AIRA. The blue shade represents an AR. L1 and L2 are the identification lines. φ1 and φ2 denote the latitudes of the maximum IVT registered on L1 and L2, respectively. φmin corresponds to the latitude of the farthest point from the AR spine whose IVT value over L1 still exceeds the IVT threshold. Δl is the distance between the identification lines expressed in degrees of longitude. These parameters are used to calculate the AR direction (d), the AR width over L1 (a) and the AR width (w).

As far as I understand the IVT threshold was calculated using all time stamps and not only those at 12:00 UTC (when moisture is at the higher end) as in for e.g. Lavers et al. (2013). Likely this may result in a lower threshold which should be discussed.

The IVT threshold was derived from a calculation using all time steps. This concern is also present in the Special Comments section, so we would like to address the answer there.

The BASE, ARI and ARCI experiments should be better described for those readers who are not specialists in aerosol modelling and those who are not familiar with the WRF-Chem model. What precisely is meant by semi-direct and direct effects on a physical basis? The interaction of aerosoles with radiation beyond the optical depth in ARI should be physically explained. The same would help for the interaction of cloud (micro-)physics. Are condensation-nuclei reduced due to precipitation for example?). If so, in which of the BASE, ARI, and ARCI experiments is this the case? In the current version only references to literature about the WRF model and it's coupling is given. A brief summary about coupling prognostic variables, input, and output etc would be helpful. This knowledge is essential for the understanding of the results.

Thank you for your comment. These three experiments were developed and have been used before by other members of our research group. Their complete description is similar (using ERA20-C data as input, instead of GCMs data) to the one depicted in

Jerez, S., Palacios-Peña, L., Gutiérrez, C., Jiménez-Guerrero, P., López-Romero, J. M., Pravia-Sarabia, E., and Montávez, J. P.: Sensitivity of surface solar radiation to aerosol–radiation and aerosol–cloud interactions over Europe in WRFv3.6.1 climatic runs with fully interactive aerosols, Geoscientific Model Development, 14, 1533–1551, https://doi.org/10.5194/gmd-14-1533-2021, 2021.

We referenced this work in line 85. However, your comment made us realise that it may be not so clear that this reference intention was to offer the reader a complete description of the simulations. Therefore, we have change it to make it more explicit:

"Three experiments were considered in this study, each of which included different aerosol interactions. The complete description of these three simulations can be found in Jerez et al. (2021). "

Furthermore, we are going to include a brief description of the three experiments, answering all your question (aerosol-radiation interactions, cloud microphysics, CCN reduced due to precipitation), in the revised manuscript. In addition, a brief summary of the model inputs-outputs will also be included.

What is meant by direct effects (radiation scattering, absorption and emission) and semi-direct effects (thermodynamical changes in the clouds induced by direct effects) of the aerosols has been explained in the specific comments section.

A short explanation of what is meant by direct, semi-direct and indirect aerosol effects was included in the introduction, as this knowledge allows a better understanding of the simulations:

"Aerosols, both natural and anthropogenic, interact with incoming solar radiation by absorption and scattering processes (direct aerosol effects). On a global scale, scattering has a net cooling effect on the surface (Jerez et al., 2021; Glassmeier et al., 2021). However, the regional impacts may differ significantly depending on the type of aerosol (Palacios-Peña et al., 2019; Palacios-Peña et al., 2020; Li et al., 2022). Additionally, direct effects can derive in thermodynamic alterations of cloud properties, leading to subsequent changes in radiative forcing (semi-direct effects (Hansen et al., 1997)). Moreover, aerosols interact with clouds, acting as Cloud Condensation Nuclei (CCN), affecting cloud albedo (Twomey effect (Twomey, 1977), first indirect effect) and cloud lifetime (Albrecht effect (Albrecht, 1989), second indirect effect), as well as precipitation (López-Romero et al., 2021; Sun and Zhao, 2021)."

Despite the answer above, we have decided to change the reference of Jerez et al. (2021) to López-Romero et al. (2021), as we have concluded that it explains the BASE, ARI and ARCI simulations better. We have improved the description of the physical set-up of the simulations, based on what can be found in López-Romero et al. (2021). Moreover, we included a brief mention to the use of nudging (as other referees have commented the lack of a nudging-related sentence).

The data description now reads as follows:

"[...] which were incorporated into the inner domain via boundary conditions as in Palacios-Peña et al. (2019). **Nudging was used for the outer domain in order to minimize the internal variability of the model. The boundary conditions for the outer domain were updated every 6 hours and the model outputs were recorded every hour.** The vertical domain comprised 29 non-uniform sigma levels with higher resolution near the surface, subsequently interpolated to pressure levels. The upper boundary was set at the 50 hPa level.

**The physics configuration included the Lin microphysics scheme (Lin et al., 1983), the Noah land surface layer (Tewari et al., 2004), the RRTM radiative scheme for both short- and longwave (Iacono et al., 2008), the Grell 3D ensemble cumulus scheme (Grell, 1993; Grell and Dévényi, 2002), and the University of Yonsei boundary layer scheme (Hong et al., 2006).**

Three experiments were considered in this study, each of which included different aerosol interactions. **The complete description of these three simulations can be found in López-Romero et al. (2021).** The BASE experiment served as a reference, [...]

In the ARI and ARCI experiments, aerosols were calculated in the WRF-Chem model using a coupled approach, where the model solved the aerosol dynamics online, allowing it to incorporate its own aerosols based on variables such as soil type, vegetation, and wind at each point of the domain (López-Romero et al., 2021). **The gas-phase chemical mechanism RACM-KPP (Stockwell et al., 1997; Geiger et al., 2003) used in the model was coupled to the GOCART aerosol module (Ginoux et al., 2001; Chin et al., 2002), which considers** five aerosol species: sulphates, mineral dust, sea salt, organic matter, and black carbon. **The Fast-J module (Fast et al., 2006) was used for photolysis and the Guenther scheme (Guenther et al., 2006) was employed for biogenic emissions. Anthropogenic emissions were derived from the Atmospheric Chemistry and Climate Model Intercomparison Project (ACCMIP) (Lamarque et al., 2010) and did not vary during the simulations. However, natural emissions are dependent on meteorological conditions and thus change over time (Jiménez-Guerrero et al., 2013).**

The WRF-Chem model facilitated converting the single-momentum Lin parameterization into a double-momentum one**, essential for the comprehension of aerosol indirect effects. This microphysics approach involves six species: water vapor, cloud water, rain, cloud ice, snow, and graupel (Ghan et al., 1997). The conversion of cloud droplets into rain droplets depends on the droplet number (Liu et al., 2005). The rates of droplet nucleation and evaporation represent aerosol activation and resuspension rates. Although the experiments did not consider ice nuclei based on forecasted aerosols, a prescribed ice nuclei distribution was used to include ice clouds. Radiation-**cloud interactions were included by connecting the number of simulated cloud droplets to the Goddard solar radiation scheme **(Chou and Suarez, 1999),** representing the first indirect effect **(i.e., increased droplet number due to increases in aerosols),** and to Lin's microphysics parameterisation, representing the second indirect effect **(i.e., decreased precipitation efficiency related to increases in aerosols).** Consequently, the

number of droplets affected both their mean radius and the optical depth of the cloud."

In the results sections the physical processes that lead to differences in the three experiments should be better and more verbosely explained to meet a broader readership which are not only atmospheric or aerosole researchers. For example, often a heating or cooling is proposed but as no corresponding temperature anomaly is shown this is hard to see.

Thank you for your comment. Similar questions are found in the Specific Comments section, so we would like to address the answer there. However, the main explanation is that thickness is directly and solely related to temperature in an atmospheric layer between two fixed pressure levels. We have added a little explanation (see the answer to the specific comment) to the revised manuscript to make it easier to follow the results.

Also the clustering procedure which is based on leading EOFs of salt and aerosoles is not sufficiently described. All this makes it difficult follow the results and final conclusions. More examples are given below.

We have computed the sea salt and dust anomalies for a reduced spatial domain and then we have treated these two variables (the anomalies) as a single vector/field (of double the length of each aerosol field individually). We have then performed a Principal Component Analysis (PCA), also known as an EOFs analysis, of said field over time (considering the 80 common events). We have retained as many PCs (EOFs) as needed to explain at least a 75 % of the total variance. Then, we have performed a hierarchical clustering (using the Ward method) over the PCA, and the centroids of the resulting clusters are shown in Figs. 7 and 10. Each cluster centroid consists on a dust field and a sea salt field.

This is also explained in the manuscript (lines 257-261): "Initially, an EOF analysis has been jointly performed for the sea salt and dust AOD (550 nm) standardised anomalies within the region bounded by -15° E and 4° E longitude and 33° N and 45° N latitude. The ARI and ARCI experiments used five and six EOFs respectively, explaining 75 % of each total variance. A clustering classification was then performed on these analyses, which separated the common cases into eight different groups. The centroid of each cluster was associated with two centre fields, one per considered aerosol."

The manuscript changes related to this issue are described in the Specific/Special Comments section.

**Special Comments**

line 7: "The analysis of common AR events showed that the differences between simulations were minimal in the most intense cases, and a negative correlation was found between mean direction and mean latitude differences.

please rephrase: what is meant? you have three sensitivity simulations. When the ARs are located more to the North in e.g. BASE, then the direction is more south in ARI and ARCI? Perhaps it's better to remove the second part of the sentence.

This sentence means that if an AR in ARI (ARCI) is located further North than in BASE (positive latitude differences), then the AR in ARI (ARCI) presents a more zonal direction (negative direction differences) and viceversa.

An example sentence has been included in the abstract:

"This implies that more zonal ARs in ARI or ARCI with respect to BASE could also be linked to northward deviations."

line 11 deviations from what? What precisely is meant by reinforcement and attenuation? is it the moisture transport (in most studies taken as a proxy for intensity) or precipitation?

Deviations refer to spatial differences with the reference simulation, BASE. When we talk about intensity (or intensity reinforcement/attenuation), it refers to the magnitude (modulus) of the IVT.

We've changed that line to make it as clear as possible:

"[...], inducing spatial deviations and IVT magnitude reinforcements/attenuations with respect to the BASE simulation depending on the aerosol configuration."

**Introduction**

line 33: what is meant by "anomalous"? Heavy precipitation above a certain threshold?

Yes, that's what is meant here. For more information, we recommend consulting the reference, where this statement was derived from.

Climate change is indeed assumed to impact on ARs. However, besides the important studies of Payne and Algarra, there also relevant studies with more focus on Europe and in particular the Iberian Peninsula. Please consider these to mention, like e.g.

Gröger, M., Dieterich, C., Dutheil, C., Meier, H. E. M., and Sein, D. V.: Atmospheric rivers in CMIP5 climate ensembles downscaled with a high-resolution regional climate model, Earth Syst. Dynam., 13, 613–631, https://doi.org/10.5194/esd-13-613-2022, 2022

Ramos, A. M., Tomé, R., Trigo, R. M., Liberato, M. L. R., and Pinto, J. G. (2016), Projected changes in atmospheric rivers affecting Europe in CMIP5 models, *Geophys. Res. Lett.*, 43, 9315–9323, doi:10.1002/2016GL070634.

Lavers, D. A., Allan, R. P., Villarini, G., Lloyd-Hughes, B., Brayshaw, D. J., and Wade, A. J.: Future changes in atmospheric rivers and their implications for winter flooding in Britain, Environ. Res. Lett., 8, 034010, https://doi.org/10.1088/1748-9326/8/3/034010, 2013.

Thank you very much for these references. We have found them very interesting and we are going to include them in the revised manuscript.

We have included them in the Introduction as examples of the following statement:

"The modification of ARs due to climate change is of great research interest (e.g., Lavers et al. (2013), Ramos et al. (2016b), Payne et al. (2020), Algarra et al. (2020), Gröger et al. (2022), O'Brien et al. (2022), Shields et al. (2023)). Some of these

authors suggest that an increased atmospheric moisture due to global warming will lead to an intensification of ARs activity, and to a potential enhancement of the AR-related precipitation."

line 46: "tracking its long 2D structure". Do you mean tracking its elongated 2D structure?

Yes, thank you for the comment. We have corrected it.

"Therefore, most of them consist of detecting the arrival of the potential AR and spatially tracking its elongated 2D structure until it is delimited for that fixed time. "

line 51: That's true. The effect of resolved spatial orography on the representation of AR over land was found most evident over the Iberian Peninsula (see e.g. aforementioned study by Gröger et al.).

Thank you, we have added this citation here too.

"In such cases, the use of regional climate models (RCMs) with higher resolution can provide a better understanding of ARs. This is the case of the IP (Gröger et al., 2022), characterised by a complex orography."

line 57-61: The interactive online coupling between aerosole modules and other climate compartments will represent feedbacks by aerosoles in a much more realistic way. May be this could be explained a bit more in the Introduction. Can you mention some feedbacks we neglect if we use only prescribed fields of aerosoles instead of simulated ones?

As said in the General Comments, we are going to include a more complete description of the three simulations in the Data section. There, we are going to compare the BASE experiment (prescribed aerosols) with the ARI and ARCI experiments (aerosols calculation fully coupled in the model). However, in lines 57-61, we are trying to motivate/highlight the potential of these differences to change the simulated ARs. We find your comment very relevant, so we are going to extend these lines and mention some feedbacks that would be neglected in a prescribed aerosol configuration, like the changes in the CCN concentration due to precipitation or the modification of the cloud droplets properties based on the aerosol (acting as CCN) characteristics.

We have added the following:

"RCMs typically introduce aerosol species and their concentrations in a prescribed manner (Forkel et al., 2015), neglecting changes in their concentration and interactions with radiation and cloud microphysics, **thus not taking into consideration some important feedback processes, like changes in the CCN concentration due to precipitation or the modification of cloud droplets properties based on the aerosol type acting as CCN.**"

**Methods**

line 75: "The WRF-Chem model (v.3.6.1) was used for the simulations, both in a decoupled configuration (WRF alone (Skamarock et al., 2008)) and in a fully coupled

configuration with atmospheric chemistry and pollutant transport to account for aerosol-radiation and aerosol-cloud interactions (Grell et al., 2005)"

What does fully coupled mean and how is the coupling precisely done in the three experiments? This is essential to understand the results in this study. The section could benefit from a brief description of the WRF-Chem and how aerosoles have direct and semi direct effects on the models physics.

Thank you for your comment. This brief description of the WRF-Chem and aerosols effects will be included in the revised manuscript as addressed on a previous answer. Fully coupled means that it is included as an active coupled component; i.e., the model chemistry (aerosols) is computed simultaneously and integrated into the dynamics of the WRF model. This contrasts with the stand-alone configuration, in which the results of both parts can be computed independently and the chemical part is not re-introduced into the model.

See the answer to the general comment related to the simulations for the specific changes implemented.

**section 2.1 Data**

line 81: "… encompass major dust emission areas". Which are these areas? The Sahara desert?

Yes, the Sahara desert and its surroundings are the main dust emission areas for the IP. An explicit mention has been added to the manuscript:

"[…] encompass major dust emission areas (the Sahara desert and its surroundings), which were incorporated into the inner domain via boundary conditions […]"

line 87: what is CCN and how does it interact with model physics?

CCN stands for *Cloud Condensation Nuclei.* Their concentration and nature alter the physical properties and amount of cloud droplets, thus changing the lifetime of the clouds and the thermodynamics of the atmospheric layer in which they are present. Further information about the microphysics scheme used in the simulations will be included in the revised manuscript, in the brief experiments description mentioned before.

We have added the CCN meaning to the first appearance of the acronym.

aerosol-radiation interactions: does radiation then alter the optical properties of Aerosoles and or the number of condensation nuclei?

Radiation generally does not alter the optical properties of the aerosols, but aerosols do alter radiation by means of scattering, absorption and emission processes (that is what is called *direct effects* of the aerosols). These processes can also induce thermodynamical changes in the clouds (*semi-direct* effects), altering the size of the droplets and/or their development.

line 131: "First, the magnitude and direction of the IVT are bi-linearly interpolated to the detection lines, L1 and L2, enabling the computation of the required variables. "

What variables are meant here? The sentence implies that IVT is calculated from specific moisture, u, and v as a first step and thereafter IVT is interpolated from the

models grid to L1 and L2. What variables do you mean here in addition to u,v, and q and for what are they necessary?

Thank you for your question. IVT is calculated only from q, u and v. When we said "required variables", we were refering to the derived variables that can be computed from the IVT vector and the geometry of the detection lines, such as the direction, the width or the IVT maximum in both lines at a given time step. These variables are required later in the algorithm to determine whether an AR has been identified.

We have changed the sentence to make it more clear:

"First, the IVT magnitude and direction are bi-linearly interpolated to the detection lines, L1 and L2, enabling the computation of the geometrical and physical variables required later in the algorithm."

Line 134: How is the threshold value determined? Is this threshold latitude dependent? Is it determined from climatological values like e.g. the 85th percentile as in other algorithms? L1 extents over a wide range of latitudes ranging from semi-arid climates to more wet conditions. Are the northern latitudes more represented in the threshold than those from the south?

The threshold value is an absolute value stablished by the user. It is not latitude dependent and it is not determined by computing percentiles, at least in the algorithm itself. However, we recommend computing them beforehand to decide the threshold. For instance, we have chosen an IVT threshold of 300 kg m-1 s-1, based on the 99th percentile value of the IVT on L1 (260 kg m$^{-1}$ s$^{-1}$). As for the L1 question, detection line 1 extents over a wide range of latitudes but we do not think that any of them are more represented than the others. In fact, this methodology is also applied by other ARDTs. In the figure below, we show the distribution of the mean impact latitude of the identified ARCI ARs (similar results were found for the other experiments), which turned to be more or less even.

[Figure]

Figure: Number of ARs versus their mean impact latitude in the ARCI simulation for the whole period.

Some sentences have been included to clarify that it is an absolute threshold. See the Response to Referee #2 (page 31 of this document) regarding this issue for specific changes.

Line 139. "...direction of AR...". If I interpret equation 4 right, wouldn't the term orientation not better than the term direction? Direction might be related more to the movement of the AR over time.

Thank you for the interesting remark. From our perspective, both terms (direction and orientation) would be correct in this case, because the ARs tend to move in the direction given by their orientation.

Line 165: How is "s" determined? Do ARs not move over time so that changes in their axis latitudinal position are not unusual? Please explain why this is necessary.

The parameter "s" is determined by the extention of the detection lines and prior knowledge of the AR behaviour in this area, where the occurrence of two consecutive ARs is not so rare. ARs move over time, thus changing the latitudinal position of their maximums (spine), but these changes are gradual due to the movement. If we detect a large shift in the position (almost the length of the detection line), we assume that an AR is passing by the South and another one is arriving to the North, or viceversa, of the dectection area, instead of consider both of them as part of the same AR event. Distinguishing these events as different ARs is the reason why this parameter is needed here and adopts such high value. However, it can be changed by the user.

line 169: ".. estimation of the AR length..". Do you mean AR duration here? The length scale isn't determined so far, is it?

The consecutive time steps mentioned before in that sentence correspond to the duration of the AR (or at least an estimation of it), but this duration allows the estimation of the AR length, knowing the wind velocity. That is what this sentence refers to. In this paragraph, our aim was to just present the algorithm, thus no specific correspondence between length and duration has been made yet for the studied area. In section 3.1 (AIRA implementation to our study domain), one can read the following: "Secondly, the minimum time duration for an interval to be classified as an AR was set at T = 10 h. Given that the mean wind speed associated with ARs in the study area is around 30 m s$^{-1}$, this minimum duration would indicate the occurrence of an AR of approximately 1,000 km in length."

**185 ff:**

Please explain how the value of the mean 90th percentile is calculated. Is it determined over all latitudinal points (i.e. m=22) at L1 and over the whole time period? Or do you calculate 22 90 percentiles an at the end average over the 22 points? Also contrary to other algorithms you take into account all day times while others include only time steps of 12:00 UTC time stamps (when moisture content is high due to solar heating). This is likely the cause why your value of 260 kg m-1s-1 seems a bit lower than in other studies (see e.g. Lavers and Villarini, 2013: https://agupubs.onlinelibrary.wiley.com/doi/full/10.1002/grl.50636)

Thank you for your question. First of all, we would like to clarify that there was a typo in this sentence, and we have already amended it in the revised manuscript. The

99th percentile was what was supposed to be written here. This percentile was just used to give us an idea of the IVT threshold and it was computed as the mean 99th percentile of the IVT field over L1 during all the considered time steps, i.e., for every given time, we computed the field percentile, and then we applied a time mean. With respect to your second remark, we used all the time steps, instead of only those with the 12:00 UTC time stamp. As you pointed out, this is likely why the 260 kg m-1 s-1 seems lower than the threshold obtained by percentile calculations in other studies. However, we were aware of this issue and thus we chose a higher IVT threshold, 300 kg m-1 s-1. In the revised manuscript, we will provide a brief discussion about this matter, referencing other studies.

Firstly, we have corrected the typo. Then, we have included a short discussion about the percentile value: "Firstly, the mean of the **99th** percentile of the IVT magnitude on L1 for all time steps resulted in a value close to 260 kg m$^{-1}$ s$^{-1}$ in all three experiments. **The computation of this value using only the data with the 12:00 h time stamp would have resulted in a higher IVT, as seen in Lavers and Villarini (2013) or Ramos et al. (2016a).**"

Is there any empirical evidence to support the limits for w (150 – 800 km)?

There are different ARs catalogs, and some of them even include a representation tool to see the identified ARs. At the beginning of our research, we explored these catalogs and came up with these limit values for the AR width, which seem reasonable. A minimum width of 150 km allows us to distinguish very thin structures that are not ARs. However, we want to clarify that these limits are just parameters of the algorithm and thus are adjustable by the user.

The spatial / temporal criteria listed in Table 1 seem to be more or less reasonable from theoretical/geometrical considerations, but ultimately lack empirical evidence. So it would supportive if sensitivity tests could be made to estimate the sensitivity of the thresholds on the AR frequency, duration and intensity. If this is too much effort, this should be at least discussed in terms of uncertainties associated with the algorithm.

Thank you for your comment. This remark was also mentioned by the other referees. Following your suggestions and those of the other two referees, we have performed an analysis of the sensitivity of the IVT threshold given a fixed minimum duration and the sensitivity of the duration threshold given a fixed IVT threshold. The results are exposed in the tables below and include the variation in the number of ARs in each simulation, the number of common ARs events, the percentage of common AR time steps and the mean intensity and mean duration of the identified ARs.

On one hand, a lower IVT threshold results in a decrease in the number of ARs but also in an increase of their duration, because two very close in time events could be identified as a single but longer event. On the other hand, increasing the IVT threshold over 300 kg m$^{-1}$ s$^{-1}$ reduces the mean duration of the ARs but has little impact on the number of ARs itself. For instance, the selection of an IVT threshold of 400 kg m$^{-1}$ s$^{-1}$ would have resulted in a decrease in the number of ARs in BASE, ARI and ARCI of 2.5%, 5.6% and 6.8%, respectively.

With respect to the sensitivity of the duration threshold, the results turned as expected. The higher the minimum duration imposed, the lower the number of ARs

identified that meet this condition. Furthermore, we also wanted to remark that the selected parameter (T=10h), gives rise to the highest percentage of common AR time steps, with 80 common events that have allowed us to perform our comparison study.

We have included the sensitivity analysis as a new subsection (3.1.1) inside the AIRA implementation and application discussion, containing both tables displayed below and the following paragraphs:

"Using a single ARDT can entail some limitations when studying ARs. ARTMIP has conclusively demonstrated that the thresholds selection constitutes the principal source of variability in AR metrics across different ARDTs, resulting in substantial variations in frequency, depending on the chosen criteria (Rutz et al., 2019). Among the different parameters, the IVT threshold was reported to be the main contributor to the uncertainty. To address this variability, an analysis of the sensitivity to the IVT threshold given a fixed minimum duration and the sensitivity to the duration threshold given a fixed $\Gamma$ was performed (Tables 2 and 3). The values of the remaining AIRA parameters were identical to those presented in Table 1.

Lowering the IVT threshold decreased the number of ARs but increased their duration due to the possibility of multiple closely timed events being identified as a single, longer event. Conversely, raising the IVT threshold above 300 kg m$^{-1}$ s$^{-1}$ resulted in a decrease in the mean duration of the ARs but had little effect on the number of ARs itself. For example, the selection of an IVT threshold of 400 kg m$^{-1}$ s$^{-1}$ would have led to a decrease in the number of identified ARs in BASE, ARI and ARCI of 2.5%, 5.6%, and 6.8%, respectively. As for the duration threshold, increasing the value of the minimum duration criteria resulted in a lower number of identified ARs."

Table 2. Sensitivity analysis to the IVT threshold, given a fixed minimum duration (T = 10 h), of the number of ARs identified in the three simulations, the number of common (COM) AR events, the percentage of common AR time-steps and the mean intensity and duration of the ARs of the three simulations.

| T = 10 h | $\Gamma$ (kg m$^{-1}$ s$^{-1}$) | | | | | | | | |
|---|---|---|---|---|---|---|---|---|---|
| | 200 | 225 | 250 | 275 | **300** | 325 | 350 | 375 | 400 |
| ARs BASE (#) | 194 | 212 | 230 | 236 | **244** | 245 | 252 | 244 | 238 |
| ARs ARI (#) | 166 | 195 | 210 | 217 | **248** | 254 | 247 | 230 | 234 |
| ARs ARCI (#) | 173 | 205 | 222 | 232 | **250** | 244 | 243 | 230 | 233 |
| ARs COM (#) | 39 | 54 | 63 | 73 | **80** | 92 | 94 | 86 | 91 |
| COM time-steps (%) | 24.79 | 28.51 | 32.11 | 33.65 | **37.16** | 40.54 | 38.91 | 38.11 | 40.65 |
| $\overline{IVT}$ BASE (kg m$^{-1}$ s$^{-1}$) | 344.42 | 380.58 | 407.61 | 435.15 | **469.20** | 495.67 | 523.24 | 549.25 | 579.26 |
| $\overline{IVT}$ ARI (kg m$^{-1}$ s$^{-1}$) | 345.14 | 373.10 | 407.43 | 440.88 | **465.47** | 491.42 | 520.35 | 551.15 | 589.44 |
| $\overline{IVT}$ ARCI (kg m$^{-1}$ s$^{-1}$) | 347.59 | 377.23 | 404.54 | 434.99 | **459.18** | 490.43 | 517.50 | 550.55 | 574.19 |
| $\overline{d}$ BASE (h) | 53.17 | 50.73 | 47.54 | 45.36 | **42.55** | 40.44 | 40.11 | 37.75 | 36.35 |
| $\overline{d}$ ARI (h) | 56.76 | 52.44 | 51.24 | 47.47 | **43.13** | 41.26 | 39.00 | 37.69 | 36.46 |
| $\overline{d}$ ARCI (h) | 56.61 | 53.45 | 48.71 | 46.72 | **43.79** | 43.10 | 41.82 | 40.03 | 36.76 |

Table 3. Sensitivity analysis to the minimum duration threshold, given a fixed IVT threshold ($\Gamma$ = 300 kg m$^{-1}$ s$^{-1}$), of the number of ARs identified in the three simulations, the number of common (COM) AR events and the percentage of common AR time-steps.

| $\Gamma$ = 300 kg m$^{-1}$ s$^{-1}$ | T (h) | | | | | | | | |
|---|---|---|---|---|---|---|---|---|---|
| | 4 | 6 | 8 | **10** | 12 | 14 | 16 | 20 | 24 |
| ARs BASE (#) | 267 | 262 | 253 | **244** | 233 | 222 | 209 | 183 | 162 |
| ARs ARI (#) | 261 | 259 | 254 | **248** | 232 | 225 | 212 | 193 | 170 |
| ARs ARCI (#) | 267 | 261 | 254 | **250** | 233 | 226 | 212 | 198 | 171 |
| ARs COM (#) | 86 | 85 | 84 | **80** | 74 | 69 | 65 | 58 | 50 |
| COM time-steps (%) | 35.73 | 35.83 | 35.94 | **37.16** | 36.19 | 36.12 | 35.97 | 36.41 | 36.75 |

**section 3.2.1**

Figure 5 fits very well with result from Gao et al. (Fig. 8) and Gröger et al., 2022 (Fig. 5d). They could be mentioned to support the validity of the new developed AIRA algorithm.

Gao et al.: https://journals.ametsoc.org/view/journals/clim/29/18/jcli-d-16-0088.1.xml

Thank you very much for the comment. We have added the references of both studies to the revised manuscript.

Bearing in mind also the relevant study of Baek and Lora (2021) mentioned by Referees #2 and #3, we have added the following sentence to the AR-related precipitation discussion:

"These results are similar to those obtained by Gao et al. (2016) and Gröger et al. (2022) at the regional-scale, and consistent with the findings of Baek and Lora (2021) for the IP at global-scale."

**section 3.3 Common events**

you may consider renaming the section, e.g. coherence of events or so

We kindly appreciate your comment. This is something that we have already discussed and we ended up choosing "Common AR events". We would like to keep the section name as it is, because the main idea is that we have analysed here the AR events shared by the three simulations, thus we think that the term "common", common to the three simulations, fits well in this case.

I would speculate that the different treatment of aerosoles will alter not only the precipitation pattern of AR related precipitation events but also alter systematically the mean precipitation rates. Could it be that the alteration seen in AR related P are similar to those in mean P? Implying that aerosoles impact similar mean and AR precipitation events.

Thank you very much, this is a very interesting question. Of course, the different aerosol treatments may affect not only the AR-related precipitation but also the non AR-related precipitation distributions. And they affect it indeed. However, if aerosols had affected both precipitation distributions exactly similarly, there wouldn't have been changes from simulation to simulation in the percentage of the total precipitation that could be related with ARs, as the alterations would have been similar and thus compensated in the computation of the percentage.

General precipitation changes due to aerosol effects, not distinguishing AR-related precipitation, were analyzed in López-Romero et al. (2021), using the same regional simulations (BASE, ARI and ARCI) as in this study. We have found it relevant to mention this in the revised manuscript:

"The overall rainfall changes due to aerosols effects using the same set of regional simulations (BASE, ARI and ARCI) were analyzed by López-Romero et al. (2021)."

**3.3.1. Analysis of differences**

What is the idea of eliminating non coherent AR intervals to elaborate the effect of aerosoles? I think here a more profound explanation for the strategy should be added. From a methodological point of view I would guess ARs penetrate into the EuroCordex model domain roughly at the same time and at the same position. Then, differences in precipitation, IVT intensity and frequency etc. would be attributed to the different treatment of aerosoles. Can you confirm this? Consequently, the non coherent AR time steps would be the result of the aerosole treatment which would neglected in this approach. Would it be wrong to calculate Fig. 6 without the eliminating step?

Thank you very much for this interesting insight. The idea of eliminating non coherent AR intervals was to do a one-to-one comparison between the ARs of the simulations. Therefore, we needed to have the same time steps to compute the differences and then compare the IVT intensity, mean trajectory, etc. For instance, you wouldn't be able to calculate Fig. 6 without the elimination step, because the number of identified ARs is not the same in all the simulations, thus e.g. AR #150 may not be the same AR in ARI and ARCI. This is why we have followed this methodology. However, there are other approaches, like the one you mentioned. It would consist on identifying each AR in the three simulations and comparing for instance their "arrival time". We computed something similar a while ago, during the first stages of our research. Not only did it make the analysis way more complex but also it led to some dead-ends, due to the impossibility of relating the results with the effects of aerosols, at least at the general scale pursued in our research. We concluded that this approach could be very interesting in a single case study but it fell out of the scope of this work, where general conclusions have been found and study cases were used as illustrative examples of these conclusions.

We have slightly changed the structure of the "Common AR events" introduction:

"To study the potential differences between the ARs of the three experiments, a one-to-one comparison of their coherent AR events was designed. Each coherent interval reproduces the same forecast period but with three different aerosol treatments. The common AR intervals have been identified by applying AIRA to the common time steps, eliminating coincident intervals with a duration of less than 10 hours or not

satisfying the other criteria to be deemed proper ARs. As a result, a total of 80 common AR intervals from the three experiments were obtained, and their characteristics were compiled. These common AR events represent only the 37 % of the time steps with ARs concerning the BASE total. This low percentage could be attributed to [...]."

Similarly, the conclusion sentence related with this matter now reads as follows:

"Although the number of identified ARs is comparable among the three simulations, their common AR events were only observed in 37 % of the time steps with AR."

line 245: "...The maximum IVT was obtained by averaging the maximum IVT magnitude of each AR event in the three experiments...". You mean intensity here?.

Thank you for the comment. Yes, we meant the maximum intensity of each AR, which corresponds to the maximum IVT magnitude of the AR, as explained in the second paragraph of section 3.2. However, we have modified this sentence to make it more clear:

"The mean maximum IVT was obtained by averaging the maximum IVT intensity of each AR event in the three experiments."

line 246: What is meant by spatial deviation. Does it refer to the deviations in latitude (Fig. 6 middle)? Please be consistent with the terms throughout the manuscript.

When we refer to 'spatial deviations,' we are addressing differences in latitude and direction; i.e., we are referring to all non intensity related differences. We have added this aclaration to the manuscript.

"The spatial deviations (latitude and direction differences) tended to zero, [...]"

line 247: what is meant with "the three magnitudes". A distinction between three magnitude categories was not done before.

Thank you for your observation. We meant "the three variables" shown in the figure. We have modified the sentence, which now reads as follows:

"The spatial deviations (latitude and direction differences) tended to zero, and the ARI-ARCI differences of the three considered variables (latitude, direction and mean IVT) became minimal in the most intense events."

line 249: Can you summarize Fig. 6 to explain what you aim to analyze with the EOF analysis. Are there systematic differences in the deviations to BASE in Fig. 6? At a first glance, it seems like noise (with the exception that most intense ARs seems to be consistent in the experiments). Also, it would be interesting to show at least the first or three leading EOFs for sea salt to get an impression where most variance is concentrated.

Thank you for your question. Fig. 6 shows the 80 common ARs events yet unclassified. More specifically, it shows the ARI-BASE (red) and ARCI-BASE (blue) differences in mean IVT, mean latitude and mean direction. As you have just commented, the differences seem like noise at a first glance (with the exception of the most intense AR events). Thus, the aim of the following EOF and clustering analysis was to shed light on these differences, gathering similar events and then

studying their relations with aerosols. The EOF analysis (or PCA) was primarily used to reduce the dimensionality of the study problem and to perform the clustering (hierarchical classification, see the answer to the related question in the General Comments section for more information) over it, to obtain the different groups of events which were subsequently studied individually.

With respect to the first or three leading EOFs, we also find it interesting and we are going to add it to the revised manuscript as an appendix, or maybe as supplementary material.

A sentence introducing the need of a further analysis of the differences has been added at the end of Section 3.3.1.:

"The absence of a clear general signal in the differences prompted the clustering analysis explained in the following section."

Figures B1 and B2 have been included as Appendix B, portraying the three leading EOFs obtained in the joint EOF analysis of dust and sea salt aerosols. Each EOF is thus associated with two fields, one per considered aerosol type. In addition, the following sentence was added to the text in Section 3.3.2:

"The ARI and ARCI experiments used five and six retained EOFs to explain at least a 75% of their total variance, respectively. **The three leading EOFs of each experiment are portrayed in Figs. B1 and B2.**"

[Figure]

Figure B1. Three leading EOFs of the joint analysis of dust and sea salt aerosols of the 80 common AR events in the ARI simulation. Each EOF is associated with two fields, one per considered aerosol: (top) dust and (bottom) sea salt. The percentage of variance explained by each EOF is shown in brackets.

[Figure]

Figure B2. Three leading EOFs of the joint analysis of dust and sea salt aerosols of the 80 common AR events in the ARCI simulation. Each EOF is associated with two fields, one per considered aerosol: (top) dust and (bottom) sea salt. The percentage of variance explained by each EOF is shown in brackets.

**section 3.3.2**

Please explain more verbose how the EOF analysis was performed, e.g. how were salt and aerosole anomalies calculated, what clustering algorithm was applied etc. Moreover, is there a seasonality in the aerosol fields (as you showed for AR incidents)? Could the clustering also explained by over-representations of certain seasons? Did the classification procedure require to choose the number of different clusters? If so, why were 8 classes chosen?

As previously mentioned in the General Comments section, we have computed the sea salt and dust standardized anomalies for a reduced spatial domain, treating these two variables as a single vector/field (of double the length of each aerosol field individually). We have then performed a Principal Component Analysis (PCA), also known as an EOFs analysis, of said field over time (considering the 80 common events). In fact, the PCA function in R is able to perform also the computation of the standardized anomalies, making the analysis straightforward. We have retained as many PCs (EOFs) as needed to explain at least a 75% of the total variance. Then, we have performed a hierarchical clustering (using the Ward method, which is the default method in the HCPC function in R) over the PCA. This classification procedure has an optimal number of resulting clusters (obtained by elbow diagrams), but one can choose a different number of clusters by looking at the tree diagram of the classification. That's how we made our decision (see Figure below). The centroids of the resulting clusters are shown in Figs. 7 and 10. Each cluster centroid consists on a dust field and a sea salt field.

With respect to the aerosols seasonality, we want to clarify that we have included in the analysis only the aerosol fields of the 80 common events, and these events are not evenly distributed along the year. Autumn common AR events are the most numerous. In all the 8 clusters of each experiment (ARI and ARCI), we have found AR events of very different seasons. For example, ARI cluster 2 gathers 1 January AR, 3 March ARs, 2 April ARs, 1 from May, 1 from June and 4 October ARs.

[Figure]

Figure: Tree diagrams of the hierarchical clustering classification of sea salt and dust aerosols jointly in the ARI (left) and ARCI (right) experiments.

The clustering method has been added and the explanation of the Principal Component Analysis has been improved. It now reads as follows:

"Initially, an EOF analysis (Principal Component Analysis) has been jointly performed for the sea salt and dust AOD (550 nm) standardized anomalies within the region bounded by -15° E and 4° E longitude and 33° N and 45° N latitude. The ARI and ARCI experiments used five and six retained EOFs to explain at least a 75 % of their total variance, respectively. The three leading EOFs of each experiment are portrayed in Figs. B1 and B2. A clustering classification following the Ward method (Ward, 1963) was then performed on these analyses, which separated the common cases into eight different groups in every experiment. The centroid of each cluster was associated with two centre fields, one per considered aerosol."

line 259/Figure 7: " …it was observed that an AR weakening occurs in clusters 2 and 3." Not clear at first reading what is meant. Figure 7 top (which I think this statement is related to) shows the IVT difference at the y-axis, in the caption it reads as "magnitude", and in the text it is termed weakening.

The magnitude of a vector is also called the modulus of a vector. IVT magnitude refers to the modulus of the IVT, the intensity of the AR. The mentioned statement is related to Figure 8 top, where the y-axis shows the IVT differences and the x-axis shows the different clusters. As you can see, the IVT differences between ARI and BASE in clusters 2 and 3 were mainly negative, thus the IVT of the ARs related to the

ARI simulation was generally lower than the IVT of the ARs related to BASE. Therefore, the ARs in ARI are weakened in comparison with BASE.

We have added the following clarification:

"[…], it was observed that an AR weakening (negative ARI-BASE IVT differences) occurs in clusters 2 and 3."

Figure 7 shows red points which are not explained.

Is it possible that you are referring to Figure 8? Figure 8 is a boxplot and the points represent the outliers. They are more present in the clusters with a higher number of members due to a higher variability between the members of those clusters. However, Referee #2 suggested the inclusion of a little explanation about what a boxplot shows (quartiles, median, outliers, etc.), so there will be a more profound insight in the revised manuscript.

The following explanation has been added to the manuscript:

"Figure 8 is a box and whiskers plot that shows the ARI-BASE differences of mean IVT, mean incidence latitude and mean IVT direction of the common AR events belonging to each ARI cluster. The box length represents the interquartile range (IQR) of the data, thus the bottom (Q1) and top (Q3) edges of the box correspond to the $25^{th}$ and $75^{th}$ percentiles, respectively. The line inside the box is the median, or $50^{th}$ percentile. The whiskers extend to 1.5 times the IQR. The outliers, data points that fall outside the whiskers range, are marked with dots."

line 262: what kind of frontal surface? that of a storm? An explanation at this first place what is meant by a thickness to non-specialists is lacking.

We strongly appreciate your observation. An explanation of the thickness of an atmospheric layer and its relation with temperature is clearly missing and it may cause a misconception of the present section for those readers who are not so familiar with this definition. Before further discussion of the thickness field and answering your first question, ARs are usually related to the frontal surfaces (cold fronts) of extratropical cyclones, as said previously in the Introduction section.

In meteorology, the thickness of an atmospheric layer refers to the vertical distance [m] between two pressure levels, which define the layer. The hypsometric equation (see equation below) represents the relation between the thickness (x) of an atmospheric layer and its mean virtual temperature ($T_v$). $R_d$ is the specific gas constant for dry air, $g_0$ is the standard gravitational acceleration, $P_1$ is the pressure of the inferior level and $P_2$ is the pressure of the superior level.

$$x = \frac{R_d T_v}{g_0} \ln\left(\frac{P_1}{P_2}\right)$$

Therefore, the thickness of the layer is directly and solely related (by a multiplicative constant) to its mean temperature given two fixed pressure levels (1000 and 850 hPa in this study). That is why we are able to talk about cooling or heating by analysing the thickness fields. The higher the thickness, the higher its mean temperature.

We have added an explanatory sentence and the citation to a well-known book of meteorology, so that non-specialists can follow the results presented in this manuscript while having access to a further and more detailed reading about this topic:

"ARs are commonly associated with a frontal surface, which can be identified by analyzing the thickness field. The thickness field of an atmospheric layer is directly and solely related to its mean virtual temperature given two fixed pressure levels, as depicted in the hypsometric equation (Stull, 2011)."

line 267: "In cluster 3, a wider cooling effect is present, but the more pronounced cooling in the south (over the north of Africa) leads to the observed weakening". Where is this cooling derived from? Figure 9 show thickness [m]. There is no information about temperature differences at this place.

Thank you again for your question. It is derived from the thickness differences given two fixed pressure levels. As explained before, in such cases, the thickness of the atmospheric layer is directly related to its mean temperature.

Accordingly I have to go back to Figure 7 where an elevated dust concentration in the region is visible. Shall I interpret this as proxy for cooling in this region (in the sense of dimming?). So far no explanation for the assumed cooling is given at the place of line 267. I get lost here...

Thank you for your comment. We are sorry for the confusion. Although there is not a complex explanation for the cooling, we should have made the statement in a more explicit way (and we have corrected it in the revised manuscript). The cooling is mainly due to the scattering and absorption of solar radiation, also known as direct effects or aerosol-radiation interactions. In a not so summarized way, the explanation would be as follows. In the BASE experiment, the AOD is set to zero, which means that radiation encounters a perfectly "clean" atmosphere. In contrast, the ARI experiment includes the on-line calculation of the aerosols optical properties (AOD) and their interactions with radiation are activated in the model. With that said, in the ARI experiment, part of the incident radiation is scattered and absorbed by dust aerosols, thus changing the mean temperature of the considered atmospheric layer (1000-850 hPa) with respect to the BASE experiment. These dust-related temperature changes are usually a warming of the atmospheric layer in which the aerosols are present and a cooling of the surface and its adjacent atmospheric layer.

The new explanation reads as follows:

"As observed in cluster 2, the inclusion of aerosol-radiation interactions (direct effects) of dust aerosols in the ARI experiment results in a cooling of the atmospheric layer. This cooling acting on the warmer zones of the domain derives in weaker thickness gradients when compared to BASE, simulation in which radiation encountered a perfectly clean atmosphere (prescribed AOD set to zero)."

line 279 ff: The ARCI-BASE comparison reads much better than the previous ARI-BASE comparison because physical explanations that appear plausible are given to the reader. This should be likewise provided for the ARI-BASE comparison. Saying this, most of the explanation is based on the interpreted aerosole effect on temperature which is not shown itself, though often it is argued with "cooling" or

"warming". Therefore it would help to show additional plots for temperature either instead of thickness or as supplementary material.

Thank you very much for the positive feedback about the ARCI-BASE comparison. Thanks to your previous comment, and as said in its answer, we are going to add an explicit explanation for the cooling found in the ARI-BASE comparison to make it more complete. With respect to the temperature plots, we refer again to the definition of thickness and its relation with the mean temperature of an atmospheric layer. Bearing this in mind, we do not find necessary to show temperature plots, because all our considerations can be derived and followed by means of the thickness plots. Furthermore, they include information of a whole layer instead of only representing a specific pressure level. However, if you still find it necessary to include some additional temperature plots (perhaps for two or three different pressure levels) after all the previous considerations, we could include them as supplementary material.

**3.3.3 Case studies**

line 309: "...-70.32 and 58.01 kg m−1 s−1..." over which area has this been averaged. Over the AR area? Model domain? Iberian Peninsula?

Thank you for your question. It is not a spatial average, but a time average. Each AR is characterized by its mean intensity (mean IVT modulus of the AR spine), among other variables. Thus, when we mention the IVT differences between the simulations, we are just comparing the mean IVT of that AR in the three experiments.

lines 309 to 333:

This paragraph reads very well as it provides a process-based discussion about the aerosol effects on ARs, involving a chain of interactions between temperature, clouds, droplets etc. The role of heating/cooling and temperature gradients is again highlighted and the reader may wonder if it would be possible to support this statement by a figure showing e.g. temperature anomalies.

Thank your for your feedback about this paragraph. Once again, we want to refer to the hypsometric equation and how it directly relates the mean temperature of an atmospheric layer with its thickness. For additional information, see the answers above.

In particular, the paragraphs (and already the previous ones) emphasize the cooling effect by aerosoles as well as a heating effect from more abundant droplets, prolonged cloud presence, and latent heat gain are discussed. However, it becomes not quite clear why the individual effects (cooling or heating) dominate in the respective cases. This could be more explained.

Thank you very much for your comment. We have shown the effects of each aerosol type (dust and sea salt) when only their interactions with radiation were implemented and when both aerosol-radiation and aerosol-cloud interactions were activated in the model. We have seen that microphysics effects tend to dominate over radiation effects in the ARCI simulation, not only compensating but also surpassing the radiative effects. This may be due to their relation with greater energy (heat and thus temperature) changes.

line 323-324. Isn' it rather a southwestward shift seen in Figure 15 ARCI-BASE?

Thank you for your remark. That sentence was only referring to the mean impact latitude of that AR over the detection line. That is why we just said "southward" shift/deviation, because the line has a fixed longitude. We have changed this sentence to make it more clear in the revised manuscript.

"In contrast, the indirect effects of dust aerosols in the ARCI experiment result in a heating effect, and a further southward latitude deviation of the AR trajectory."

**Conclusions**

Including an atmospheric chemistry and trajectory model yields likely the most realistic and physically consistent treatment of aersoles. But it is likely also the most expensive? If so can we derive from the experiments a statement whether or not the additional online coupling of an expensive chemistry/aerosole model is worth and/or in which cases? Can we expect systematic shifts in AR related precipitation and or moisture convergence which may be of importance on climate related time scales? Would the conclusions also hold for e.g. the U.K. which is further away from major dust aerosole sources?

Thank you for these interesting questions. Including the atmospheric chemistry and aerosol transport in the model is 4 to 8 times more expensive, according to the experts in our research group. On one hand, from a physical point of view, the more accurate representation of the physical processes leads to more realistic interactions between the model components and thus better and more realistic results. The higher computational cost can be worth in researches that aim to study these physical processes and/or the relative significance of each interaction. This was the case of the present work. On the other hand, some studies suggest that the differences obtained in the most expensive and most physically realistic runs are mainly relevant at the very local scale (distribution changes). Therefore, these very complex simulations would not represent a substantially better reference for operational use at the synoptic scale.

With regards to the shifts in AR-related precipitation, our findings reveal that they would be very case-dependent, influenced by the aerosol fields configuration present at that moment. In this research, the distinction between dust and sea salt aerosols effects was made. Therefore, the conclusions regarding sea salt aerosols would also hold for the U.K., even if the dust aerosol concentration is negligible.

**Response to Referee #2**

**General comments:**

1. The relationship between the dust, sea salt and the AR mechanisms needs to be more clearly and directly shown. The thickness diagnostics (and differences) for the composite "common" ARs are hard to interpret. I would recommend keeping these diagnostics for the case studies, but illustrating the connection between the ARs and aerosols cluster groups more directly, or in a more focused way. Ideas include using AR variables themselves,(IVT, IWV, or low level winds) for the "strengthening/weakening" component with the aerosols in lat/lon space, rather than box/whisker, and only showing the clusters that are significant. Or perhaps AR-spine centric averages vs aerosols cluster (highest density areas?)(and/or perhaps thickness) in scatter plots, to show this relationship. I think it is there, but at present it is a little unfocused. Also, significance needs to be shown in any difference plots.

Thank you for your suggestions. We have substantially improved the physical description of the processes that relate aerosols and the observed AR differences in the revised manuscript, especially for the ARI-BASE comparison, thanks to some comments of Referee #1.

In an early stage of our research, we also considered the analysis/representation of the IVT or IWV fields instead of the thickness diagnostics to analyse the clusters. We have a collection with all the ARs IVT representations available. However, as we have discussed later in the Specific Comments section, representing e.g. the IVT fields of all the members of a cluster (or averaging them) made quite difficult the extraction of any conclusions. The natural variability of ARs, with their diverse trajectories, locations, width, etc. obscured/hided the patterns in the differences of the group, thus complicating the relation with aerosols effects. After trying many approaches, we came up with the present methodology. Furthermore, our physical discussion is mainly based on temperature changes due to aerosols effects, and the thickness fields show these changes. However, thanks to some of your specific suggestions, we have added the trajectories of each AR belonging to a cluster and their mean trajectory to the thickness plots of the most relevant clusters (Fig. 9 and 12). For instance, you can find the resulting plot of the ARI clusters 2-3 in the answer to the "case studies" specific comment. Each thin arrow represents an AR. It is located on its mean latitude with its mean direction and the length of the arrow is proportional to its mean intensity (IVT). The thicker arrow represents the mean characteristics of the ARs belonging to the cluster. We sincerely hope that the revisions we have implemented address your concerns.

In response to your comment regarding the significance of the differences, we will include statistical significance to the greatest extent possible, where it is applicable. Given the small sample size of some clusters, we are aware that statistical analyses may have limitations. In cases like this, it may be more reasonable to focus on providing a qualitative description of the observed differences. That being said, we would like to once again express our gratitude for your valuable feedback.

The implemented changes to the thickness plots are explained in the Specific Comments section. In addition to the arrows representing ARs, we have included

significance to the difference plots (Figs. 9 and 12, found below), highlighting the significant points with a 90 % confidence level. Following a similar criteria, we have added the p-values of the differences shown in Figs. 8 and 11. These significance analyses have been performed only on the clusters with at least five members, so as a small sample size does not preclude obtaining meaningful results. We would like to refer to the answers in the Specific Comments section for specific modifications.

2. AR and ARDT uncertainty needs to be addressed. AIRA needs to be put into context of published ARDTs, and specifically, regional-specific algorithms that cover the Iberian Peninsula (e.g. IDL/Ramos, Lavers, Brands). Given the IDL code uses transects and also a Lagrarangian framework, this is the most similar type of code). ARTMIP (https://www.cgd.ucar.edu/projects/artmip/algorithms will have the reference list for the above mentioned ARDTs) has robustly shown that threshold choice is the largest source of AR metrics variability across ARDTs with dramatic differences in frequency, for example, depending on how this is chosen. See specific comments for details on suggestions on how to address this issue.

We strongly appreciate your comments and suggestions here. It is something that was missing and the other referees also noted this issue. In the revised manuscript, we are going to put AIRA in the ARTMIP context and classification, including its main differences with the IDL/Ramos, Lavers and Brands ARDT algorithms, which are the most similar to AIRA and also detect ARs over the Iberian Peninsula. As a preliminary observation, the main contrast is that these algorithms make use of spatial tracking, while AIRA never uses it, as it is intended to perform also in regions close to the domain edges. This is indeed the case in our study, with the detection lines located very near the limits of the spatial domain.

With respect to the AR and ARDT uncertainty, we have re-ran AIRA multiple times with different IVT and duration thresholds to assess the sensitivity to the thresholds choice. For information about the results, we would like to refer to the Specific Comments section.

We have included a comparison between AIRA and the mentioned ARDTs (page 1 of this document), and a sensitivity analysis to the chosen thresholds was performed (page 13). See the Specific Comments section and the answers to Referee #1 for further discussion.

3. Referencing needs to be improved and representative of the recent AR literature.

Thank you for your remark. We have substantially improved the referencing of this work in the revised manuscript thanks to not only your valuable suggestions but also the recommendations of the other two referees.

See the Specific Comments section for specific improvements in the referencing.

**Specific comments:**

Line 21: In the midlatitudes, this is indeed the case, but not necessarily for high latitude ARs. I recommend amending this statement with "in the midlatitudes".

Thank you very much for your comment. We have corrected it:

"The number of ARs in the midlatitudes increases during autumn and winter months, as extratropical cyclones are more frequent during these seasons (Gimeno et al., 2014)."

Lines 24 and 25: There are many many references that could fit this statement, I recommend adding an "e.g.," to your citation list, or add a few more references.

Thank you for the remark. Many other references could have been used here, so we have added "e.g." to the revised manuscript, as the included references were just some examples of researches about ARs in those regions.

"From the beginning, the West Coast of the United States (e.g., Lorente-Plazas et al. (2018), Guan et al. (2012)) and the Pacific Ocean (e.g., Ralph et al. (2004, 2011)) have been the most studied regions."

Lines 28,32,34: Again, there are quite a few that could be listed here, so "e.g." should be used. I am surprised not to see any Lavers references as this group was among the first to discuss North Atlantic ARs.

We appreciate again your remark. We have corrected it by using "e.g." in lines 28-34 and we have also included two references to the work of Lavers and Villarini in lines 33-34:

"[...] The modification of ARs due to climate change is of great research interest (e.g., Lavers et al. (2013), Ramos et al. (2016b), Payne et al. (2020), Algarra et al. (2020), Gröger et al. (2022), O'Brien et al. (2022), Shields et al. (2023)). Some of these authors suggest that an increased atmospheric moisture due to global warming will lead to an intensification of ARs activity, and to a potential enhancement of the AR-related precipitation. Another topic of great interest is the influence of ARs on the Arctic Sea ice, as they have been related with a slowing of the ice seasonal recovery (e.g., Zhang et al. (2023)). Western Europe has been the focus of several studies in the last decade. These studies have demonstrated a connection between ARs and their Mediterranean variant (Lorente-Plazas et al., 2020) with some of the heaviest rainfall recorded in the Iberian Peninsula (IP) (e.g., Lavers and Villarini (2013, 2015), Trigo et al. (2015), Eiras-Barca et al. (2018)). [...]"

Paragraph Line 36: I appreciate the author's discussion here, but there are some major gaps in the literature review. ARTMIP has had a number of workshops, plus 5 major group/overview papers, and many contributed papers. All discuss the issues of defining and detecting ARs, and the philosophy of using an ARDT (AR detection tool) that is appropriate for the science question asked. In addition to referencing the workshop report (or instead of), please read and cite the following papers. (Note: the climate change papers, O'Brien and Shields/Payne, would be good additions to the climate change literature review sentences, with the Rutz and Collow papers for reanalysis).

Shields, C. A., Rutz, J. J., Leung, L.-Y., Ralph, F. M., Wehner, M., Kawzenuk, B., Lora, J. M., McClenny, E., Osborne, T., Payne, A. E., Ullrich, P., Gershunov, A., Goldenson, N., Guan, B., Qian, Y., Ramos, A. M., Sarangi, C., Sellars, S., Gorodetskaya, I., Kashinath, K., Kurlin, V., Mahoney, K., Muszynski, G., Pierce, R., Subramanian, A. C., Tome, R., Waliser, D., Walton, D., Wick, G., Wilson, A., Lavers, D., Prabhat, Collow, A., Krishnan, H., Magnusdottir, G., and Nguyen, P.: Atmospheric River Tracking Method

Intercomparison Project (ARTMIP): project goals and experimental design, Geosci. Model Dev., 11, 2455-2474, https://doi.org/10.5194/gmd-11-2455-2018, 2018.

Rutz, J.J, Shields, C.A., Lora, J.M, Payne, A.E., Guan, B., Ullrich, P., O'Brien, T., Leung, L.-Y., Ralph, F.M., Wehner, M., Brands, S., Collow, A., Goldenson, N., Gorodetskaya, I., Griffith, H., Hagos, S., Kashinath, K., Kawzenuk, B., Krishnan, H., Kurlin, V., Lavers, D., Magnusdottir, G., Mahoney, K., McClenny, E., Muszynski, G., Nguyen, P.D., Prabhat, Qian, Y., Ramos, A.M., Sarangi, C., Sellars, S., Shulgina, T., Tome, R., Waliser, D., Walton, D., Wick, G., Wilson, A., Viale, M.: The Atmospheric River Tracking Method Intercomparison Project (ARTMIP): Quantifying Uncertainties in Atmospheric River Climatology, Journal of Geophysical Research-Atmospheres , https://doi.org/10.1029/2019JD030936, 2019.

O'Brien, Travis Allen and Wehner, Michael F and Payne, Ashley E. and Shields, Christine A and Rutz, Jonathan J. and Leung, L. Ruby and Ralph, F. Martin and Marquardt Collow, Allison B. and Guan, Bin and Lora, Juan Manuel and et al., (2022) Increases in Future AR Count and Size: Overview of the ARTMIP Tier 2 CMIP5/6 Experiment. JGR-A https://agupubs.onlinelibrary.wiley.com/doi/10.1029/2021JD036013.

Collow, A.B., Shields, C.A., Guan, B., Kim, S., Lora, J.M., McClenny, E.E., Nardi, K., Payne, A., Reid, K., Shearer, E. J. , Tome, R., Wille, J.D., Ramos, A.M., Gorodetskaya, I.V., Leung, L.R., O'Brien, T.A., Ralph, F.M., Rutz, J. Ullirich, P.A., Wehner, M., (2022) An Overview of ARTMIP's Tier 2 Reanalysis Intercomparison: Uncertainty in the Detection of Atmospheric Rivers and their Associated Precipitation, Journal of Geophysical Research, Atmospheres, https://agupubs.onlinelibrary.wiley.com/doi/10.1029/2021JD036155.

Shields, C. A., Payne, A. E., Shearer, E. J., Wehner, M. F., O'Brien, T. A., Rutz, J. J., Leung, L.R., Ralph, F. M., Collow, A. B. M., Ullrich, P. A. Ullrich, Dong, Q., Gershunov, A., Griffith, H., Guan, B., Lora, J. M., Lu, M., McClenny, E., Nardi, K. M., Pan, M., Qian, Y., Ramos, A. M. Ramos, Shulgina, T., Viale, M., Sarangi, C., Tomé, R., Zarzycki, C. (2023). Future atmospheric rivers and impacts on precipitation: Overview of the ARTMIP Tier 2 high-resolution global warming experiment. Geophysical Research Letters, 50, e2022GL102091. https://doi.org/10.1029/2022GL102091

More details on ARTMIP here: https://www.cgd.ucar.edu/projects/artmip

We kindly appreciate all your suggestions and recommended references. We have improved and extended this discussion in the revised manuscript. Furthermore, some of these references were also good additions to other parts of the text, as you just mentioned.

Said paragraph now reads as follows:

"The importance of ARs has given rise to numerous identification algorithms (also known as Atmospheric River Detection Tools, ARDTs) with a wide range of methodologies and conclusions. This diversity is, among others, due to the ongoing need of establishing a robust AR definition (Gimeno et al., 2021) and to the vast variety of questions that these ARDTs were developed to answer. The Atmospheric River Tracking Method Intercomparison Project (ARTMIP, Shields et al. (2018)) aims to quantify the uncertainties in AR climatology based on detection algorithms alone

and to provide guidance on the most appropriate algorithm for a given science question or study region. ARTMIP also states the need of creating a common software infrastructure and classifying ARDTs to understand the broad uncertainty in AR detection results. The outcomes of ARTMIP Tier 1 phase addressed these topics and were summarized in Rutz et al. (2019). It was found that threshold values were the main contributors to AR uncertainty. For instance, an IVT magnitude greater than 250 kg m$^{-1}$ s$^{-1}$ and a length over 2,000 km would be considered an AR according to some algorithms (Zhu and Newell, 1998) but not to others. Percentiles of the IVT or IWV fields, typically the 85$^{th}$ or 90$^{th}$ percentile, have also been utilized (Lavers et al., 2012). ARTMIP Tier 2 conducted several AR detection sensitivity analyses to reanalysis products, such as MERRA-2 or ERA5 (Collow et al., 2022), and under climate change scenarios (O'Brien et al., 2022; Shields et al., 2023), including their impacts on AR-related precipitation. They found that the ARDT selection is the main contributor to the uncertainty in projected AR frequency. Therefore, climate change studies should consider using more than a single ARDT and assessing their uncertainties. The Third ARTMIP Workshop (O'Brien et al., 2020) contemplates the existence of different "flavors" of ARs, although most tracking methods have not considered this possibility yet. Future AR researches would also be able to apply machine-learning techniques easily."

In addition, some of the references were also included into the climate change sentences:

"The modification of ARs due to climate change is of great research interest (e.g., Lavers et al. (2013), Ramos et al. (2016b), Payne et al. (2020), Algarra et al. (2020), Gröger et al. (2022), O'Brien et al. (2022), Shields et al. (2023)). Some of these authors suggest that an increased atmospheric moisture due to global warming will lead to an intensification of ARs activity, and to a potential enhancement of the AR-related precipitation."

Line 48: The statement that GCM's "may not accurately represent their (AR) behavior" is a bit misleading. Most GCMs (and ESMs) are able to simulate the synoptics, bulk numbers, duration, etc. realistically. I recommend amending this statement specifically to AR-precipitation, given it is the precipitation piece that does better with high resolution (citations are needed here, there are quite a few out there now for high resolution global/earth system models, and ARs).

Thank you for your comment. We have amended it in the revised manuscript.

"As ARs interact with orography on a regional scale, GCMs can represent ARs but may not accurately reproduce AR-related precipitation (Lorente-Plazas et al., 2018). "

Line 51: I am not sure I understand why a timeslice approach doesn't work for limited area models? Many timeslice ARDTs work well within a limited area domain (see the ARDT list on the ARTMIP webpage, some of these are both timeslice and regional). I agree with the authors that regional ARDTs tend to do a better job because localized considerations are made for regional-specific that would not otherwise be considered in globals (for example, for IP, the complex topography and the North Atlantic storm track climatology). If this is the intent of the authors, I recommend using this as motivation for the newly developed ARDT for the IP, rather

than timeslice vs lagrangian approach. If I misunderstood, please make this statement more clear.

Many ARDTs work well within a limited area domain (regional domain) if it is big enough to perform the spatial tracking (mainly over the ocean) usually required to determine the length of the AR. In our case, the detection lines were very close to the limits of the study domain, as you can see on Figure 1 (red box, inner domain). Therefore, we introduced a duration-length relation to estimate the length of the AR, allowing us to work with smaller regions and thus reducing the time to perform computationally costly simulations such as online aerosol runs to understand ARs mechanisms. As referee #3 commented, the innovation of AIRA relies on overcoming the RCMs limitations where most of the runs are focused over land, and this precludes capturing the long way over the ocean. This was the motivation to develop this new regional ARDT, not only the higher resolution (which is also an advantage that plays an important role in the study of AR behavior and AR-related precipitation at the local scale). The statement in line 51 has been corrected, specifying the cases in which spatial tracking given a fixed time step method is not suitable (not enough domain to perform the tracking). We are going to include this motivation as clear as possible in the revised manuscript and we want to thank you again for the interesting questions.

The statement of line 51 now reads as follows: "Nevertheless, it should be taken into account that the spatial tracking given a fixed time step method may not be suitable for data obtained from RCMs whose spatial limits are very close to the detection area. This is the case for most of the RCM runs, as they are primarily land-focused."

The first sentences of Section 2.2 presented a similar issue as line 51, thus it has also been amended: "The identification of ARs on a global scale may not apply to regional climate simulations due to the limited spatial domain. Consequently, it would be impossible to determine the complete length of an AR if the regional domain is not sufficiently wide to track the AR structure for a fixed time. Many ARDTs employ this method (e.g., Brands et al. (2017), Ramos et al. (2016))."

Introduction general comment: I am surprised there is no mention of the Calwater experiment. Although this was focused on the western U.S., it was an important and groundbreaking study to look at aerosols with observations and AR. Here is a citation from CalWater that uses the same model as this study, i.e. WRF-Chem.

Naeger, A. R. (2018). Impact of dust aerosols on precipitation associated with atmospheric rivers using WRF-Chem simulations. Results in Physics, 10, 217-221, https://www.sciencedirect.com/science/article/pii/S2211379717318223

Thank you very much for this comment, we have added a brief mention to this study in the Introduction section of the revised manuscript.

"Another relevant research, conducted by Naeger (2018), explored the impact of long-range transported dust aerosols on the precipitation related with a specific AR over the western United States. "

Paragraph at line 74: It might be useful to readers familiar with climate models, but not WRF forecast systems, to add a sentence or two explaining how lateral boundary conditions nudge the model back to the "observations". This is important for when

you describe your common ARs periods later, it makes sense to use common periods given each simulation is reproducing the same forecast period, but just with different aerosol treatments. If I am misunderstanding the design, please clarify.

Thank you for your comment. Other referees have requested a brief explanation about the model set up and a more profound explanation about the experiments. Although the complete physical set up description of the three simulations is the same as in the reference included in line 85 (Jerez, S., Palacios-Peña, L., Gutiérrez, C., Jiménez-Guerrero, P., López-Romero, J. M., Pravia-Sarabia, E., and Montávez, J. P.: Sensitivity of surface solar radiation to aerosol–radiation and aerosol–cloud interactions over Europe in WRFv3.6.1 climatic runs with fully interactive aerosols, Geoscientific Model Development, 14, 1533–1551, https://doi.org/10.5194/gmd-14-1533-2021, 2021), we are going to include a brief description in the revised manuscript.

In answer to your question, boundary conditions from the reanalysis ERA20-C were updated every 6 h to the outer domain. Although nudging was applied to the outer domain, neither nudging nor re-initialization of initial conditions have been used in the target (inner) domain. We were interested in allowing the model to run "freely" in this domain once the initial conditions had been established, in order to see how the different aerosol treatments affected the simulations. We will address these comments in the brief explanation of the experiments in the revised manuscript.

See the answers to Referee #1 for general additions to the simulations description. With respect to nudging, we have included the following sentences:

"Nudging was used for the outer domain in order to minimize the internal variability of the model. The boundary conditions for the outer domain were updated every 6 hours and the model outputs were recorded every hour."

Line 88: Just checking how "online" is meant here, as an active coupled component and not stand-alone simulation?

Thank you for your question. Yes, that's exactly what is meant here. We have added a little explanation to that sentence: "In the ARI experiment, aerosols were treated online, introduced as an active fully coupled component, and the aerosol-radiation interactions were activated in the model".

Line 108: I think this a Lagrangian approach, i.e. tracking rather than timeslice, given Figure 2? I am not sure I understand why a regional ARDT can't track an AR? This approach is similar to the IDL ARDT (an ARTMIP contributor, Ramos et al., 2016). I think it would be helpful to add what aspects of AR science that AIRA addresses that the IDL does not. Or, how it compares to IDL, especially given both of these ARDT look at Iberian ARs.

Ramos, A. M., Nieto, R., Tomé, R., Gimeno, L., Trigo, R. M., Liberato, M. L. R., and Lavers, D. A.: Atmospheric rivers moisture sources from a Lagrangian perspective, Earth Syst. Dynam., 7, 371–384, https://doi.org/10.5194/esd-7-371-2016, 2016

Thank you for your question. AIRA never uses spatial tracking, because the detection lines are so close to the domain limits that it would not be possible to do it. This is the main difference with the IDL Ramos approach, because it performs the tracking to estimate the length of the AR, but we have introduced a duration-length correspondence (given an estimation of the wind speed of ARs in the studied area).

However, we are going to include a comparison between the IDL ARDT and Brands ARDT with respect to AIRA in the revised manuscript, as suggested by your second general comment and by the other referees. With respect to the second question here, ARDTs can track ARs if the regional area they are working on is wide enough to perform the spatial tracking. It was not the case of our region. We would like to refer to the answer given to the specific comment about line 51.

Following the valuable suggestions of the three referees, we have included a paragraph at the end of Section 2.2.2 including this comparison. This paragraph can be found in the answer to the first general comment of Referee #1 (page 1 of this document).

Line 134 and Paragraph at Line 185: From Table 1 and paragraph at Line 185, I think this is an absolute threshold, used for all simulations and does not change with the respective simulated climatologies? If so, please state that an absolute threshold is used for all simulations in the initial description, and point to the application for further explanation.

You are right, it is an absolute threshold established by the user and we have chosen to use the same value for all three simulations. Following your suggestion, we will state in this section that it is an absolute threshold used for all simulations, and we will refer to the AIRA implementation section for more information.

We have clarified that the IVT threshold is an absolute threshold after line 134: "This filter applies a threshold value Γ to the IVT magnitude. Γ is an absolute threshold established by the user. Section 3.1 contains specific information about the AIRA implementation in this study".

We have stated that the same parameters were used for the three simulations in the second paragraph of Section 3.1: "Before implementing the algorithm, it is necessary to determine the values of the parameters involved. The same values were used in the application of AIRA for the three simulations (Table 1). "

Line 196: Which ARDT catalogues/datasets were compared? The Brands ARDT contributions to ARTMIP are regional algorithms.

As answered to Referee #3, by the time this research was conducted, there was a website mentioned in Brands et al. (2017) with their *Atmospheric Rivers Archive* available: http://www.meteo.unican.es/atmospheric-rivers. This catalogue documented all the ARs detected by their algorithm using ERA-20C data and we compared our results with it (see figure below for some qualitative examples). Unfortunately, the page was shutdown. To answer Referee #3 questions, we have contacted the authors and they have provided us a database with all the information through a Zenodo repository (https://doi.org/10.5281/zenodo.8010794), although the representation tool is not available anymore. In the revised manuscript, we are going to assess the coincidences to the fullest extent possible.

A paragraph was added to the text at the end of Section 3.1. We would like to refer to the answer to Referee #3 (page 46 of this document) for more information about the changes performed in the manuscript related with this matter.

[Figure]

Figure: ARs identified the 1992-12-18 (left) and 1998-03-03 (right) by AIRA (top) and Brands ARDT (bottom, Brands et al., 2017). In the top images, green, red and blue contours/shades represent the ARs of the BASE, ARI and ARCI simulations, respectively.

Line 203: This is consistent with ARTMIP findings as October being the month with the maximum frequency for these latitudes (Rutz et al. 2019, Fig 13).

Thank you very much for your remark. We have added this citation to the manuscript:

"Notably, the highest number of ARs is detected in October, with at least 30 ARs identified in all three simulations (Fig. 4 (top)). This result is consistent with the findings of Rutz et al. (2019)"

Line 207: The mean intensity values are somewhat "baked in" to the values given the application of an absolute threshold.

Thank you for your comment. The lower the IVT threshold, the lower the mean intensity of the identified ARs, and viceversa. However, to identify ARs we have to set an IVT threshold, either absolute or relative. For precise information about how the mean intensity of the ARs in each simulation changes with the IVT threshold, see the answer to the last question, where we try to assess the variability of AIRA.

Figure 5: I noticed is that the AR metrics presented in this paper do not agree with other published results that look at aerosols, ARs, and climate, (Baek et al., 2021)

where the Baek shows very little change over the Iberian Peninsula in the thermodynamic/precipitation and more of a change with the dynamics. There could be many reasons, including model resolution, aerosol treatment, ARDT, but this should be discussed or addressed in some way.

Baek, S.H., Lora, J.M. Counterbalancing influences of aerosols and greenhouse gases on atmospheric rivers. Nat. Clim. Chang. 11, 958–965 (2021). https://doi-org.cuucar.idm.oclc.org/10.1038/s41558-021-01166-8

Thank you for your comment. Referee #3 has suggested mentioning this paper in the Introduction section, although it seems relevant to include it also during the AR-related precipitation discussion. As a preliminary comparison, our approach in Fig. 5 is similar to Extended Data Fig. 2 (% AR Precip Relative to Total Precip) of said paper, and we even use the same metrics (a percentage). For the historical period of their study (1920-2005), the authors have obtained an AR-related precipitation between 20 and 40% of the total accumulated precipitation over the North Atlantic coast of the IP. These results are consistent with our outcomes (around a 30% of maximum percentage over this region) and with those obtained by Gao et al. (2016) and Gröger et al. (2022), as Referee #1 has pointed out. Thanks to the higher resolution of regional data, we could perform a local-scale analysis of the distribution of this AR-related precipitation percentage over the IP. It allowed us, e.g., to highlight a lower percentage over the Northwest due to a higher amount of non AR-related precipitation. In our study, instead of comparing a historical period and a future period, we analysed the changes in three simulations of the same period due to different aerosols treatments: prescribed (BASE), only direct and semi-direct effects included (ARI) and all aerosol-radiation-cloud interactions activated in the model (ARCI). Furthermore, as depicted in Fig. 5, the greatest percentage differences were observed in the ARCI-BASE comparison over the Southwest, showing an increasing of approximately 5%, which is not an exceptionally large difference.

We have included a brief mention to this study in the Introduction section (see the answer to Referee #3) and to the AR-related precipitation discussion:

"These results are similar to those obtained by Gao et al. (2016) and Gröger et al. (2022) at the regional-scale, and consistent with the findings of Baek and Lora (2021) for the IP at global-scale."

Line 238: I am not convinced that 80 AR clusters is enough to overcome natural variability, could you add some discussion on the robustness of only using 80? Have you considered playing with your threshold to increase your sample size? Would the results be the same if you used a fixed-relative threshold, based on the "base" climatology? And/or a simple relative climatology unique to each of your experiments (base, ari, aric?) This would increase your sample size and also test uncertainty in your AR definition. (One thing that ARTMIP has shown is that the moisture threshold value is by far the biggest influence on AR frequency, and quite significantly so).

We kindly appreciate these interesting comments. We have played with the IVT threshold to see its influence on the number of ARs, their mean intensity and duration, the number of common AR events that would result and the percentage of AR steps shared by the three simulations. You can find a table displaying the results

in the answer to your last question. Bearing that in mind, the 80 common events employed seem like a reasonable approach to extract conclusions, as we have clustered them based on their aerosol configurations. Increasing the sample size could have increased the number of members in every cluster, but the conclusions would have been similar.

The chosen threshold (300 kg m-1 s-1) is an absolute value (already discussed) that was derived from the computation of the 99th percentile (there was a typo in the manuscript that read "90th" instead of "99th", but it has been corrected) of the IVT over L1 in the BASE simulation, which yielded a time mean value of around 260 kg m-1 s-1. However, this percentile showed quite similar values (between 250 and 270 kg m-1 s-1) in ARI and ARCI, so the results could have been similar if relative thresholds were used.

Figure 6: I am not sure if this figure adds much to the manuscript as currently described. Their differences don't seem significant by eye (?) How are they important? If they are not, then maybe omit this figure.

As we have answered to Referee #1, Fig. 6 shows the 80 common ARs events yet unclassified. More specifically, it shows the ARI-BASE (red) and ARCI-BASE (blue) differences in mean IVT, mean latitude and mean direction. As you have just pointed out, the differences seem like noise at a first glance (with the exception of the most intense AR events). Thus, the aim of this figure was to motivate and illustrate the need of the following EOF and clustering analysis to shed light on these differences, gathering similar events and then studying their relations with aerosols.

As previously answered to Referee #1, a sentence introducing the need of a further analysis of the differences has been added at the end of Section 3.3.1.:

"The absence of a clear general signal in the differences prompted the clustering analysis explained in the following section."

Figure 8: Add an explanation for the box and whisker styled plots: mean, median, quantiles? What is the color scheme showing? As clusters 2 and 3 are primarily discussed, perhaps only show these instead of all the clusters? It will be more focused.

You are absolutely right, an explanation of the box and whisker plots is missing and it may lead to some difficulties when interpreting the displayed results. For instance, one of the referees posed a question regarding what the red points (outliers) were, because we had not mentioned them in the text. This explanation is going to be included in the revised manuscript.

The color scheme is just showing the ARI clusters/boxes in different shades of red and the ARCI clusters/boxes in different shades of blue, because red and blue colors represent these simulations along the work. It is just an aesthetic decision.

We have focused on clusters 2 and 3 because they were the ones that presented the biggest differences. However, we have discussed whether showing the rest of the clusters and we have concluded that it may be interesting to show how there is not a so clear signal in their differences.

As previously answered to Referee #1, the following explanation has been added to the manuscript:

"Figure 8 is a box and whiskers plot that shows the ARI-BASE differences of mean IVT, mean incidence latitude and mean IVT direction of the common AR events belonging to each ARI cluster. The box length represents the interquartile range (IQR) of the data, thus the bottom (Q1) and top (Q3) edges of the box correspond to the 25th and 75th percentiles, respectively. The line inside the box is the median, or 50th percentile. The whiskers extend to 1.5 times the IQR. The outliers, data points that fall outside the whiskers range, are marked with dots."

Figure 11: Same comment as Figure 8, as well as only showing the significant clusters.

The response here is the same as the one to the previous comment, so we would like to refer to the answer there.

Thickness field diagnostic : Have you considered showing low level winds and/or IWV instead of the frontal boundaries via thickness field for these composite plots? I would think that IWV might be a better diagnostic to show ARs, given it is the moisture stream that makes the AR unique, and not all ARs are associated with the warm conveyor belt? To show strengthening/weakening of the thickness fields, the gradient value (i.e., anomalies ahead - behind the front might be more intuitive than the difference plots which are hard to interpret. I like the thickness plots for the case studies, which help to highlight the relationship between the AR and the strengthening/weakening of the frontal boundaries, but for the composites, they are hard to interpret. If difference plots are continued to be used, then significance should be added.

Thank you again for your comment. As previously explained, we have attempted to explore other variables but the uniqueness of the ARs obscured the underlying patterns. We would like to refer to the answer to the general comment 1 for a longer discussion about this matter and statistical significance, and to the answer to the "case studies" suggestion for the changes implemented in the thickness diagnostic plots.

We have added significance to the thickness plots, where white dots highlight statistically significant differences with a 90 % confidence level. The statistical analyses have been carried out only for the clusters with at least 5 members. Clusters with a lower number preclude a meaningful analysis due to an insufficient sample size. See Figs. 9 and 12 below.

Figures 8,11: There is a lot of information packed into these figures, but not a lot of explanation in the text. Consider adding more description and inference with these figures to make your points.

Thank you for your comment. We will take it into consideration during the revision of the manuscript. Thanks to your previous suggestions and those of the other referees, the description and discussion of these two figures will be substantially improved and extended to make it as clear and complete as possible.

We have extended the description of Figs. 8 and 11, explaining what is represented in a box and whiskers plot, as mentioned in a previous comment. Furthermore, the pvalues of the clusters with at least 5 members have been added into the plots. Updated Fig. 8 can be found below this answer, to illustrate the changes.

An explanatory sentence about why we have studied ARI cluster 3, despite the fact that it only gathers two events and thus no significance analysis was suitable, has been also included to the text:

"Cluster 3, comprising only two AR events, precludes conducting a meaningful statistical significance analysis due to the insufficient sample size. However, cluster 3 could be interpreted as particularly intense dust events of the same nature as in cluster 2."

Similarly, in the case of ARCI cluster 8:

"A meaningful statistical analysis of cluster 8 is not viable with only three cases. However, it gathers the most intense sea salt events, whose effects can be explained as in cluster 7. "

Figure 8. ARI-BASE differences of the mean IVT magnitude, mean incidence latitude and mean IVT direction of the common AR intervals grouped by the eight ARI sea salt and dust cluster groups. The number of events belonging to each cluster is indicated in grey. The p-values of the clusters with at least 5 members are included in black ("*": p ≤ 0.20, "**": p ≤ 0.10, "***": p ≤ 0.05).

Figure 13: Contour labels need to be a bit bigger, it is hard to see them even after zooming in.

Thank you very much for the observation. We have made the contour labels bigger in Figures 13-18.

Case studies: I really like the figures with the dust and IVT overlays as this shows the displacements of the ARs. I would recommend trying to do something similar with the composites to help illustrate your conclusions that the aerosol locations and magnitudes impact intensity and location of the ARs. The case studies show this, but the current figures 8-12 aren't as convincing.

Thank you very much for your comment. Following also the suggestion of Referee #3, we have added the trajectories of each AR belonging to a cluster and their mean trajectory, but instead of performing this approach to all the clusters and representing it on Figs. 7 and 10, we have focused on the most relevant clusters (discussed in the manuscript), and we have added the representation of the trajectories to Figs. 9 and 12, where the thickness fields are shown. For instance, you can find the resulting representation of the ARI clusters 2-3 in the figure below this paragraph. Each thin arrow represents an AR. It is located on its mean latitude with its mean direction and the length of the arrow is proportional to its mean intensity. The thicker arrow represents the mean characteristics of the ARs belonging to the cluster.

As answered to Referee #3, Figs. 9 and 12 (shown below) have been updated to include the representation of the ARs as arrows. Their caption has been extended and a similar mention was included in the text:

"The mean thickness fields between 1,000 and 850 hPa of the events belonging to ARI clusters 2 and 3 are represented in Fig. 9 for ARI and BASE experiments. The same time steps are included in the representations of both experiments. Each thin arrow represents an AR event, located on its mean latitude and oriented accordingly to its mean direction. The length of the arrow is proportional to its mean IVT. The thickest arrow depicts the mean characteristics of all the ARs belonging to a cluster."

[Figure]

Figure 9. ARI and BASE mean thickness fields of the atmospheric layer between 1,000 and 850 hPa of the common AR events belonging to clusters 2 and 3 in the ARI simulation and ARI-BASE thickness differences. The same time steps are included in the representations of both experiments. **Each thin arrow represents an AR event in (red) ARI or (black) BASE, located on its mean latitude and oriented accordingly to its mean direction. The length of the arrow is proportional to its mean IVT. The thickest arrow represents the mean characteristics of the cluster.** White dots highlight statistically significant differences with a 90 % confidence level.

[Figure]

Figure 12. ARCI and BASE mean thickness fields of the atmospheric layer between 1,000 and 850 hPa of the common AR events belonging to clusters 2, 6, 7 and 8 in the ARCI simulation and ARCI-BASE thickness differences. The same time steps are included in the representations of both experiments. **Each thin arrow represents an AR event in (blue) ARCI or (black) BASE, located on its mean latitude with its mean direction. The length of the arrow is proportional to its mean IVT. The thickest arrow represents the mean characteristics of the cluster.** White dots highlight statistically significant differences with a 90 % confidence level.

Line 342: This was not explained or motivated convincingly and AR uncertainty (that is, the uncertainty in AR metrics due to ARDT alone) is not addressed in the

manuscript. This should be done given that AR frequency is highly sensitive to thresholding values. Suggested ways to address this: (1) Uncertainty can be discussed in the text addressing the limitations of using one ARDT, (2) For extra robustness and my recommendation, repeat the AR analysis by running the AIRA ARDT using different threshold values to both increase the sample size and attempt to bound ARDT uncertainty, (3) More work, but useful could be to compare AIRA with other ARTMIP ARDTs. Other ARDT catalogues for MERRA2 and ERA5 available, in addition to source data so AIRA could be run for a sample period for direct comparison. Data available at https://www.earthsystemgrid.org/dataset/ucar.cgd.artmip.html. Comparing to other regional ARDTs such as the IDL (Ramos), or the Brands ARDTs are highly recommended, especially if there are plans to use AIRA for other applications, including climate change where more than one ARDT is typically needed (O'Brien et al., 2022).

Thank you for all your comments. We have changed line 342 to specify that some of them are not suitable if the domain is so limited that spatial tracking can not be performed.

Referee #3 also mentioned the need of a discussion about the sensitivity to the threshold parameters. We have followed some of your suggestions. First, we have discussed in the text the limitations of using only one ARDT. Second, we have performed an analysis of the sensitivity to the IVT threshold given a fixed minimum duration and the sensitivity to the duration threshold given a fixed IVT threshold. The results are exposed in Tables 2 and 3 and include the variation in the number of ARs in each simulation, the number of common ARs events, the percentage of common AR time steps and the mean intensity and mean duration of the identified ARs.

On one hand, a lower IVT threshold results in a decrease in the number of ARs but also in an increase of their duration, because two very close in time events could be identified as a single but longer event. On the other hand, increasing the IVT threshold over 300 kg m$^{-1}$ s$^{-1}$ reduces the mean duration of the ARs but has little impact on the number of ARs itself. For instance, the selection of an IVT threshold of 400 kg m$^{-1}$ s$^{-1}$ would have resulted in a decrease in the number of ARs in BASE, ARI and ARCI of 2.5 %, 5.6 % and 6.8 %, respectively.

With respect to the sensitivity of the duration threshold, the results turned as expected. The higher the minimum duration imposed, the lower the number of ARs identified that meet this condition. Furthermore, we also wanted to remark that the selected parameter (T=10h), gives rise to the highest percentage of common AR time steps, with 80 common events that have allowed us to perform our comparison study.

The statement of line 342 has been changed: "A number of AR identification algorithms are available. However, many of them may not be suited for use with regional land-focused models, whose spatial limits are very close to the detection area, and thus preclude capturing the AR structure over the ocean. To address this issue, a novel regional scale AR identification algorithm, called AIRA, has been developed […]".

We have included the sensitivity analysis as a new subsection (3.1.1) inside the AIRA implementation and application discussion, containing both tables mentioned above. The included paragraphs and tables were previously shown in the response to Referee #1 (page 13 of this document).

**Response to Referee #3**

**General comments:**

The paper is well structured, and the algorithm adopted is easy to understand thanks to the illustrative figures. Although several approaches have been developed to identify ARs, their innovation relies on overcoming the RCMs' limitations where most of the runs are focused over land, and this precludes capturing the long way over the ocean. The success of this approach will allow the use of RCMs to provide more accurate precipitation amounts than GCMs and to perform less computationally costly simulations such as online aerosol runs to understand ARs mechanisms. Then, I found this work a valuable advance to analyze the impacts of the AR's landfalling.

Under these arguments, I recommend accepting this work after addressing a minor revision detailed below.

We sincerely appreciate your thoughtful and positive feedback on our research. Your comments on the algorithm, as well as on the innovation in addressing RCMs limitations, are of great value for our team. Your recommendation to accept the work after addressing a minor revision is encouraging. We have addressed your suggested revisions in the specific comments section to ensure the quality of the manuscript. Thank you for considering our work a valuable advance in the field.

**Specific comments:**

**Introduction**

In line 55, the authors mention the lack of research about the impact of aerosols on ARs but they did not discuss the challenges nor mention previous works such as Counterbalancing influences of aerosols and Greenhouse gases on atmospheric Rivers by Baek and Lora.

Thank you for the remark. We are going to include this discussion in the Introduction section.

The beginning of that paragraph now reads as follows:

"Several researchers have investigated the role of ARs and similar structures in the global transport of atmospheric aerosols (Chakraborty et al., 2021). However, the isolated impact of these aerosols and their variability on the formation, characteristics and behavior of ARs has received less attention. **One of the most important studies concerning this issue at global scale was carried out by Baek and Lora (2021). It uncovered opposite influences of industrial aerosols, which weakened ARs, and greenhouse gases, which strengthened them.** Another relevant research, conducted by Naeger (2018), explored the impact of long-range transported dust aerosols on the precipitation related with a specific AR over the western United States. RCMs typically introduce aerosol species [...]"

The main challenges that one has to face when carrying out a research as ours are: (1) the need of regional climate simulations with different levels of interactions

between aerosols, radiation and cloud microphysics, and (2) the use of an ARDT suitable for their domain. This is motivated in the last paragraphs of the Introduction. They have not been modified.

**Methods**

How can the AIRA be sure that is detecting an AR and not the branch of a low system with a bigger enough radius? Does ΔΘ < 25 guarantees this fact? Maybe introducing SLP values will avoid this concern.

Thank you for your question. Yes, the maximum direction difference allows us to ensure that moisture transport is taking place in the direction of the AR, distinguishing it from the branch of a low system. With respect to the slp suggestion, we kindly appreciate it. However, although it may provide some extra information which would also allow us to distinguish low systems from ARs, the currently presented method (AIRA) is sufficiently capable of performing this distinction.

In Table 1 the authors show the imposed parameters. To demonstrate the robustness of the approach some discussions about the sensitivity of these parameters are needed. For instance, how many percentages of ARs increase/decrease if the IVT threshold is modified?

Thank you for your comment. This remark was also mentioned by the other referees. Following your suggestions and those of the other two referees, we have performed an analysis of the sensitivity to the IVT threshold given a fixed minimum duration and the sensitivity to the duration threshold given a fixed IVT threshold. The results are exposed in Tables 2 and 3 and include the variation in the number of ARs in each simulation, the number of common ARs events, the percentage of common AR time steps and the mean intensity and mean duration of the identified ARs.

On one hand, a lower IVT threshold results in a decrease in the number of ARs but also in an increase of their duration, because two very close in time events could be identified as a single but longer event. On the other hand, increasing the IVT threshold over 300 kg m$^{-1}$ s$^{-1}$ reduces the mean duration of the ARs but has little impact on the number of ARs itself. For instance, the selection of an IVT threshold of 400 kg m$^{-1}$ s$^{-1}$ would have resulted in a decrease in the number of ARs in BASE, ARI and ARCI of 2.5 %, 5.6 % and 6.8 %, respectively.

With respect to the sensitivity of the duration threshold, the results turned as expected. The higher the minimum duration imposed, the lower the number of ARs identified that meet this condition. Furthermore, we also wanted to remark that the selected parameter (T=10h), gives rise to the highest percentage of common AR time steps, with 80 common events that have allowed us to perform our comparison study.

We have included the sensitivity analysis as a new subsection (3.1.1) inside the AIRA implementation and application discussion. The included paragraphs and tables were previously shown in the response to Referee #1 (page 13 of this document).

To better contextualize your methodology, I missed a discussion comparing the AIRA approach with other methodologies of other tracking approaches, For instance, a

review can be found in: Atmospheric River Tracking Method Intercomparison Project (ARTMIP): Project Goals and Experimental Design by Ruth et al.

Thank you for your comment. Other referees also noted this issue. In the revised manuscript, we are going to put AIRA in the ARTMIP context and classification, including its main differences with the IDL ARDT (Ramos et al., 2016) and Brands ARDT (Brands et al., 2017) algorithms, which are the most similar to AIRA and also detect ARs over the Iberian Peninsula. As a preliminary observation, the main contrast is that both algorithms make use of spatial tracking, while AIRA never uses it, as it is intended to perform also in regions close to the domain edges. This is indeed the case in our study, with the detection lines located very near the limits of the spatial domain.

As previously mentioned in the responses to Referees #1 and #2, we have included a comparison paragraph at the end of Section 2.2.2. This discussion was already shown on page 1 of this document.

**Results**

Following the previous comment, some validation against observations (e.g. satellite images) and/or using the ARs inventory/catalogs is needed to be the coherence of your approach with the ARs already identified along the bibliography.

In line 196 the authors mention. "It was found that most of the ARs identified by AIRA also matched those identified by global-scale algorithms, as reported by Brands et al. (2017)." How many coincidences did you find? Did you find more 'real' ARs in BASE or in ARCI? Do you think that some discrepancies may be due to a different approach or the use of an RCM instead of a GCM?

By the time this research was conducted, there was a website mentioned in Brands et al. (2017) with their *Atmospheric Rivers Archive* available: http://www.meteo.unican.es/atmospheric-rivers. This catalogue documented all ARs detected by their algorithm using ERA-20C data and we compared our results with it (see figure below for some qualitative examples). Unfortunately, the page was shutdown. To answer your questions, we have contacted the authors and they have provided us a database with all the information through a Zenodo repository (https://doi.org/10.5281/zenodo.8010794), although the representation tool is not available anymore. In the revised manuscript, we are going to assess the coincidences to the fullest extent possible to answer your first two questions.

With respect to your last comment, we think that the discrepancies could be mainly due to differences in the methodology approach, like the IVT threshold or the shorter detection line used by the Brands ARDT to study W Iberia region. In addition, we have considered the same line to study ARs on the southwest of the IP, while Brands ARDT employed a different line for S Iberia. Furthermore, aerosol effects can cause spatial deviations, as seen in this research, potentially pushing ARs out of the study area and lowering the number of coincidences. A more detailed discussion of these differences will be included in the revised manuscript.

[Figure]

Figure: ARs identified the 1992-12-18 (left) and 1998-03-03 (right) by AIRA (top) and Brands ARDT (bottom, Brands et al., 2017). In the top images, green, red and blue contours/shades represent the ARs of the BASE, ARI and ARCI simulations, respectively.

The following paragraph was added to the text at the end of Section 3.1:

"It was found that most of the ARs identified by AIRA also matched those identified by global-scale algorithms. AIRA's outcomes were compared against the results of Brands et al. (2017) ARDT for ERA20-C data over W Iberia region (Brands, 2023). Specifically, the daily JFMOND performance of both algorithms, i.e., whether an AR was present over western Iberia during a JFMOND day, displayed similar results in 82.1 %, 81.6 % and 80.9 % of the total days for BASE, ARI and ARCI, respectively. Discrepancies could be mainly due to differences in the identification approach. Brands Method 0 employed the 95[th] percentile to detect the AR arrival and the 85[th] percentile to perform the spatial tracking of the AR structure, imposing a minimum AR length of 2000 km. In addition, its detection region for W Iberia did not extend to the most southern latitudes of the IP, as they were considered as a different region. Furthermore, aerosol effects may cause spatial deviations, potentially pushing ARs out of the identification area and lowering the number of coincidences from simulation to simulation."

In Line 224 the authors assert that the ARs explain the 30% of the precipitation, it is not clear what area did you use to obtain this value, and the Fig. 5 shows strong spatial variability to perform a spatial average.

ARs don't explain the same percentage of total precipitation in every cell of the domain, as can be seen in Fig. 5. We have stated that "In all three simulations, it is apparent that the maximum percentage of total precipitation attributable to the presence of ARs is close to 30 % and occurs along the western Iberian coast, which is the impact zone of the ARs". This means that ARs could explain up to 30 % of the precipitation of a given location/grid cell and we have also shown the locations in which this maximum takes place. Nevertheless, 30 % constitutes a maximum, thus the percentage of total precipitation related to ARs decreases in the rest of the study domain, especially in the points located far from the impact zone of the ARs. Furthermore, these percentages were calculated for the whole period, but as you have mentioned in the last paragraph of your specific comments, these percentages will have temporal variability, due to the interannual variability of ARs, thus changing the precipitation.

Furthermore, how accurate is the precipitation during these events? Is ARCI or BASE more representative of the observed precipitation?

This is a very interesting question. Let's use the observed precipitation of the Iberia database. Considering the AR-related precipitation of each simulation and the observed precipitation of those same days (different from simulation to simulation), we can plot the Taylor diagram shown below. The correlation coefficient of the three simulations is higher than 0.85 and ARCI presents a lower standard deviation than the rest. The three simulations represent quite accurately their related observed precipitation.

[Figure]

Figure: Normalized Taylor diagram of the AR-related precipitation in BASE (black), ARI (red) and ARCI (blue) with respect to the observed precipitation of the set of AR days of each experiment.

In Line 232. Only 37 % of the coincidence of ARs between ARCI and BASE looks like a few percent. When the simulations are described there isn't any mention of nudging or re-initialization of initial conditions has been mentioned. What percentage of these discrepancies could be due to different treatments of aerosols or due to internal variability of the simulations?

Thank you for your question. We have already mentioned some plausible explanations to this few percentage in the manuscript: "Only 37% of the time steps with ARs coincide among all three simulations concerning the BASE total. This low percentage could be attributed to weak events and the temporal limitations of the identified ARs, where the IVT threshold is exceeded in some simulations but not in others. Furthermore, aerosol effects can cause spatial deviations, as seen in the following sections, potentially pushing ARs out of the study area, decreasing the time steps with AR on the detection lines in some experiments, and thus lowering the coincidence percentage."

Although nudging was applied to the outer domain, neither nudging nor re-initialization of initial conditions have been used in the target (inner) domain. We were interested in allowing the model to run "freely" in this domain once the initial conditions had been established, in order to see how the different aerosol treatments affected the simulations. Nudging can reduce the internal variability of the model but it would have prevented us from obtaining the desired conditions in the simulations. Another referee requested to include a more detailed explanation of the simulations design, so we will address these comments there.

To determine the exact percentage of the discrepancies that could be due to the internal variability of the model, a deeper and more complex study should be done. It would require repeating the simulations to address their variability. This is an interesting question that falls out of the scope of this research.

We have included the following sentences related to nudging, as stated previously in the Response to Referee Comments #2:

"Nudging was used for the outer domain in order to minimize the internal variability of the model. The boundary conditions for the outer domain were updated every 6 hours and the model outputs were recorded every hour."

When sea salt and dust clusters are analyzed (Fig. 7 and 10) It will be interesting to see mean ARs trajectories for each cluster (for instance superimposed with dotted lines).

Thank you very much for your comment. Following also the suggestion of Referee #2, we have added the trajectories of each AR belonging to a cluster and their mean trajectory, but instead of performing this approach to all the clusters and representing it on Fig. 7 and 10, we have focused on the most relevant clusters (discussed in the manuscript), and we have added the representation of the trajectories to Fig. 9 and 12, where the thickness fields are shown. For instance, you can find the resulting representation of the ARI clusters 2-3 in the figure below this paragraph. Each thin arrow represents an AR. It is located on its mean latitude with its mean direction and the length of the arrow is proportional to its mean intensity. The thicker arrow represents the mean characteristics of the ARs belonging to the cluster.

As commented in the Response to Referee Comments #2, Figs. 9 and 12 have been updated to include the representation of the ARs as arrows. Their caption has been extended and a similar mention was included in the text. These changes were already shown on pages 38-40 of this document.

In the analysis of the differences to better understand the thermodynamics and dynamics changes, it will be illustrative to analyze whether the IVT changes are more due to IWV or winds.

In the analysis of the differences, we have found that direct, semi-direct and indirect aerosol effects play an important role in ARs behavior and characteristics. These effects were translated into temperature differences that give rise to changes in the thermodynamic properties of the clouds, as discussed along the manuscript. These thermodynamic/temperature changes trigger the thickness field differences and thus are the origin of the dynamic changes (weakening-strengthening of the thickness field gradient and winds).

Throughout the work, I missed more analysis about the impacts of ARs on precipitation. I understand that may be the scope of future work.

Thank you for your remark. You are right, a more in-depth study about the impacts of aerosols on the precipitation related to ARs is intended as the main topic of future works. However, we strongly appreciate your precipitation-related comments above and we have added some calculations/representations to the manuscript, especially to the case studies, with the aim of making this work more complete.

Said additions can be found in the answer to the following comment.

For the case studies will be interesting to show the spatial distributions of the precipitation (accumulated during the whole event and/or hourly) for the three simulations; BASE, ARI, and ARCI. These will provide some insights about how the intensity and trajectory of ARs impact on the precipitation distributions.

As said right above, we have included this to the case studies, following your valuable suggestion. The new results show that the precipitation distributions of the involved days are quite different from simulation to simulation, being the ARCI distribution the most similar to the observed precipitation.

We have included Figs. 15 and 18, depicting the total accumulated precipitation in BASE, ARI and ARCI during the AR events considered in the case studies section. Furthermore, Appendix D has been incorporated into the manuscript, containing the observed accumulated precipitation of these events.

With respect to the 2005 event, the following comment has been added to the text: "Furthermore, Fig. 15 displays the total accumulated precipitation distribution of this event. BASE and ARI present a similar magnitude, while the ARCI experiment exhibits a notably higher amount of precipitation on the west coast of the IP. The recorded accumulated rainfall can be found in Fig. D1 (left)."

As for the 1998 event, we have added: "As a result of this shift, the ARCI simulation displays the highest values of accumulated precipitation over land (Fig. 18), which aligns with the observed data for this event (Fig. D1 (right))."

[Figure]

Figure 15. Common AR event of the 27 October 2005. Total accumulated precipitation during the entire event (2 days) in the three simulations (top) and precipitation differences (bottom). Black, red and blue contours represent BASE, ARI and ARCI ARs at 22:00 h on October 27, respectively (400 and 600 kg m$^{-1}$ s$^{-1}$ IVT levels).

[Figure]

Figure 18. Common AR event of the 12 January 1998. Total accumulated precipitation during the entire event (1 day) in the three simulations (top) and precipitation differences (bottom). Black, red and blue contours represent BASE, ARI and ARCI ARs at 09:00 h, respectively (400 and 600 kg m$^{-1}$ s$^{-1}$ IVT levels).

[Figure]

Figure D1. Observed accumulated precipitation during the case studies of (left) 27-28 October 2005 and (right) 12 January 1998. Precipitation data derived from Gutiérrez et al. (2019).

Furthermore, the authors found around 30% of ARs impact precipitation but this percentage will have spatial and temporal variability. For instance, as ARs have an interannual variability also their impact on precipitation will be significant.

Thank you for the remark. We have calculated the percentage of the total accumulated precipitation that could be related to the presence of ARs in the whole period. However, due to the interannual variability of ARs, this general percentage is supposed to change from year to year if we perform the calculation yearly. In addition, the spatial distribution may also be dependent of this interannual variability.

Finally, it will be interesting a further understand the low impact on precipitation of the ARs over Galicia, Is it less frequency of ARs, more precipitation due to cold fronts, or orographic arguments?

This was already slightly discussed in the manuscript: "In Galicia, located in the northwest region of the IP, this percentage is slightly lower owing to the greater amount of precipitation that is not associated with ARs". In fact, ARs discharge significantly more precipitation over the Galicia region than over the rest of the study domain. However, the precipitation related with other phenomena, like cold fronts, is even greater thus deriving in a lower percentage of AR-related precipitation in the area.

The explanatory sentence has been modified: "In Galicia, located in the northwest region of the IP, this percentage is slightly lower owing to the greater amount of precipitation that is associated with other phenomena, like cold fronts."

---

## Referee Report (RR1)

The authors have verbosely and comprehensively answered to all 3 reviewers comments. This resulted major revisions that that significantly improved the manuscript:

1. The study is now set in the context of the ARTMIP framework thereby direcvtly addressing a big readership already in the introduction. Also the relavant literature is now mentioned in the context of regional AR detection and the specific advancement of the introduced algorithm is clearly outlined.

2. The description of the algorithm (which is always challenging) has has been likewise improved. A new schematic figure has been added in the appendix that greatly facilitates the readability of the text passage.

3. A sufficient description of the aerosol physics in the WRF model is now added. And at many places additional explanations have been made where required.

4. The description of experiments has significantly improved.

5. The uncertainties, sensitivity and potential limitations of the algorithm are now well explained and discussed in section (e.g. section 3.1.1). The greatly increases the trust in this method.

6. Figs 15 and 18 were added to show the differences to demonstrate the aerosol effect on accumalated AR precipitation in addition the thickness plots.

7. Concerning the issue of internal model noise and thus significance of the results (raised by reviewers 2 and 3) the authors have addressed this by implementing a quick significance test to the figures highlighting the differences in layer thickness (Figure 9 &12). This is here completely sufficient. To make this really robust would require expensive ensemble runs (e.g. Ho-Hagemann et al, 2020) to assess the full internal random short term noise. However, this is not necessary within the scope of the current study which introduces a new detection algorithm.

Ho-Hagemann, H.T.M.; Hagemann, S.; Grayek, S.; Petrik, R.; Rockel, B.; Staneva, J.; Feser, F.; Schrum, C. Internal Model Variability of the Regional Coupled System Model GCOAST-AHOI. *Atmosphere* **2020**, *11*, 227. https://doi.org/10.3390/atmos11030227Ho-Hagemann, H.T.M.; Hagemann, S.; Grayek, S.; Petrik, R.; Rockel, B.; Staneva, J.; Feser, F.; Schrum, C. Internal Model Variability of the Regional Coupled System Model GCOAST-AHOI. *Atmosphere* **2020**, *11*, 227. https://doi.org/10.3390/atmos11030227

The above 7 major changes greatly improved the manuscript. Thus, from my point of view, I can fully recommend the publication of the revised version.

**Some final rather cosmetic comments the authors may consider:**

From the reply to rev #1:
**The threshold value is an absolute value stablished by the user. It is not latitude dependent and it is not determined by computing percentiles, at least in the algorithm itself. However, we recommend computing them beforehand to decide the threshold. For instance, we have chosen an IVT threshold of 300 kg m-1 s-1, based on the 99th percentile value of the IVT on L1 (260 kg m-1 s-1). As for the L1 question, detection line 1 extents over a wide range of latitudes but we do not think that any of them are more represented than the others. In fact, this methodology is also applied by other ARDTs. In the figure below, we show the distribution of the mean impact latitude of the identified ARCI ARs (similar results were found for the other experiments), which turned to be more or less even.**

I agree there is at least no significant increase with the higher latitudes up to 44°N. However, what would be if L1 would extend up to 55 ot 65 °N. Would you recommend then the use of latitude dependent values to detect Ars impacting the UK or Norway? If so, you may consider mentioning this.

**Line 64: Nevertheless, it should be taken into account that the spatial tracking given a fixed time step method may not be suitable for data obtained from RCMs whose spatial limits are very close to the detection area. This is the case for most of the RCM runs, as they are primarily land-focused."**

That's true. However, not the limited size of domain may be problematic but the also fact that ARs loose moisture after landfalling which makes so that mapping over land methods with fixed time stepping deliver very uncertain results.

In my point of view Appendix D1 could be omitted. It is used in the main text to identify which of the aerosole treatment experiment is closest to observations. But a general statement about this would likely require more than two cases studies.

---

## Author Response (AR2)

**Authors' response to final comments**

First of all, we want to express our gratitude to the topic editor and Referee #1 for their thoughtful revision of the latest version of the manuscript. This document presents a point-by-point reply to their final suggestions. Reviewer comments are shown in **black**. The authors response is included in **red**. The implemented changes are marked in **blue**.

**Some final rather cosmetic comments the authors may consider:**

From the reply to rev #1:

***The threshold value is an absolute value established by the user. It is not latitude dependent and it is not determined by computing percentiles, at least in the algorithm itself. However, we recommend computing them beforehand to decide the threshold. For instance, we have chosen an IVT threshold of 300 kg m-1 s-1, based on the 99th percentile value of the IVT on L1 (260 kg m-1 s1). As for the L1 question, detection line 1 extends over a wide range of latitudes but we do not think that any of them are more represented than the others. In fact, this methodology is also applied by other ARDTs. In the figure below, we show the distribution of the mean impact latitude of the identified ARCI ARs (similar results were found for the other experiments), which turned to be more or less even.***

I agree there is at least no significant increase with the higher latitudes up to 44°N. However, what would be if L1 would extend up to 55 to 65 °N. Would you recommend then the use of latitude dependent values to detect Ars impacting the UK or Norway? If so, you may consider mentioning this.

This is a very interesting question. In Rutz et al. (2019), Fig. 6 (shown below) illustrates the AR frequency of different ARTMIP methods for selected transects depending on the latitude. Along the European West Coast (from 35 ºN to 62 ºN), some methods diverge for latitudes higher than 44 ºN, especially *absolute* methods, like AIRA. This greater AR frequency was explained by a higher climatological value of IVT at these latitudes. In addition, this study attributed the dramatic jump in AR frequency around 45 ºN to the potential combination of climatology (placement of the storm track) and the greater number of coastal transect points at higher latitudes.

[Figure]

*Figure. AR frequency of ARTMIP methods for selected transects along the European West Coast. Image extracted from Rutz et al. (2019) Figure 6.*

Bearing these results in mind, we recommend calculating IVT percentiles before choosing the established absolute threshold. Furthermore, in response to your question, it may be advisable to use multiple smaller pairs of identification lines (sub-regions) if the region of interest comprises a wide range of high latitudes or if it includes the 44-45 ºN point, to mitigate the aforementioned divergence.

Rutz, J. J., Shields, C. A., Lora, J. M., Payne, A. E., Guan, B., Ullrich, P., O'Brien, T., Leung, L. R., Ralph, F. M., Wehner, M., Brands, S., Collow, A., Goldenson, N., Gorodetskaya, I., Griffith, H., Kashinath, K., Kawzenuk, B., Krishnan, H., Kurlin, V., Lavers, D., Magnusdottir, G., Mahoney, K., McClenny, E., Muszynski, G., Nguyen, P. D., Prabhat, M., Qian, Y., Ramos, A. M., Sarangi, C., Sellars, S., Shulgina, T., Tome, R., Waliser, D., Walton, D., Wick, G., Wilson, A. M., and Viale, M.: The Atmospheric River Tracking Method Intercomparison Project (ARTMIP): Quantifying Uncertainties in Atmospheric River Climatology, Journal of Geophysical Research: Atmospheres, 124, 13 777–13 802, https://doi.org/https://doi.org/10.1029/2019JD030936, 2019.

We have included this recommendation in Section 3.1:

"The identification lines employed in this study span a wide range of latitudes. However, no over-representation of the highest latitudes was observed. In order to study northern regions, like the UK coast, the use of smaller sub-regions may be advisable to mitigate the potentially over-increased frequency of ARs when applying absolute ARDTs at latitudes higher than 45º N (Rutz et al., 2019). Furthermore, the computation of IVT percentiles before establishing $\Gamma$ is highly recommended."

**Line 64:** *Nevertheless, it should be taken into account that the spatial tracking given a fixed time step method may not be suitable for data obtained from RCMs whose spatial limits are very close to the detection area. This is the case for most of the RCM runs, as they are primarily land-focused.*

That's true. However, not the limited size of domain may be problematic but also the fact that ARs loose moisture after landfalling which makes so that mapping over land methods with fixed time stepping deliver very uncertain results.

Thank you for your comment. It's true that once ARs make landfall they start to loose moisture and thus this is another difficulty that has to be faced when tracking ARs over land. AIRA could be useful in such cases, adjusting the IVT threshold. However, we have located the identification lines (L1 and L2) employed in this study close to the IP but over the ocean, in order to avoid said problem.

In my point of view Appendix D1 could be omitted. It is used in the main text to identify which of the aerosol treatment experiment is closest to observations. But a general statement about this would likely require more than two cases studies.

Thank you for your suggestion. You are right. The aim of Appendix D was just to show the observed precipitation distribution during the two case studies to deepen their understanding, following some suggestions of Referee #3. We did not intend to extract general conclusions. However, taking into consideration that no other relation between simulated and observed precipitation is mentioned during the manuscript, this appendix may seem out of place. Therefore, we have agreed to omit it in the final version.

We have omitted Appendix D and removed the sentences related to it:

"Furthermore, Fig. 15 displays the total accumulated precipitation distribution of this event. BASE and ARI present a similar magnitude, while the ARCI experiment exhibits a notably higher amount

of precipitation on the west coast of the IP. "

"The southward displacement of the sea salt distribution in the ARCI experiment coincides with the deviation of the AR trajectory. As a result of this shift, the ARCI simulation displays the highest values of accumulated precipitation over land (Fig. 18)."